


# Evaluation of CESM1 (WACCM) free-running and specified-dynamics atmospheric composition simulations using global multi-species satellite data records

Lucien Froidevaux[1], Douglas E. Kinnison[2], Ray Wang[3], John Anderson[4], and Ryan A. Fuller[1]

[1] Jet Propulsion Laboratory, California Institute of Technology, Pasadena, CA, USA
[2] National Center for Atmospheric Research, Boulder, CO, USA
[3] School of Earth and Atmospheric Sciences, Georgia Institute of Technology, Atlanta, GA, USA
[4] School of Science, Hampton University, Hampton, VA, USA

*Correspondence to*: L. Froidevaux (lucienf@jpl.nasa.gov)

## Abstract

We evaluate the recently delivered Community Earth System Model version 1 (CESM1) Whole Atmosphere Community Climate Model (WACCM) using satellite-derived global composition datasets, focusing on the stratosphere. The simulations include free-running (FR-WACCM) and specified-dynamics (SD-WACCM) versions of the model. Model evaluations are made using global monthly zonal mean time series obtained by the Aura Microwave Limb Sounder (MLS), as well as longer-term global data records compiled by the Global Ozone Chemistry and Related Trace gas Data Records for the Stratosphere (GOZCARDS) project. A recent update (version 2.20) to the original GOZCARDS merged ozone ($O_3$) data set is used. We discuss upper atmospheric climatology and zonal mean variability using $O_3$, hydrogen chloride (HCl), nitrous oxide ($N_2O$), nitric acid ($HNO_3$), and water vapor ($H_2O$) data. There are a few significant model/data mean biases, such as for lower stratospheric $O_3$, for which the models overestimate the mean observed values and seasonal amplitudes. Another clear difference occurs for $HNO_3$ during recurring winter periods of strong $HNO_3$ enhancements at high latitudes; this stems from the known omission of ion chemistry relating to particle precipitation effects, in the global models used here. In the lower stratosphere at high southern latitudes, the variations in polar winter/spring composition observed by MLS are generally well matched by SD-WACCM, the main exception being for the early winter rate of decrease in HCl, which is too slow in the model. In general, the latitude/pressure distributions of annual and semi-annual oscillation amplitudes derived from the MLS data are properly captured by the corresponding model values. Nevertheless, detailed aspects of the interactions between the quasi-biennial, annual, and semi-annual ozone variations in the upper stratosphere are not as well represented by FR-WACCM as by SD-WACCM.

One of the evaluation diagnostics we use represents the closeness of fit between the model/data anomaly time series, and we also consider the correlation coefficients. Not surprisingly, SD-WACCM, which is driven by realistic dynamics, generally matches observed deseasonalized anomalies better than FR-WACCM does. Other results indicate that the root mean square variability is sometimes found to be significantly smaller in FR-WACCM than in SD-WACCM and the observations. Most notably, FR-WACCM underestimates the observed interannual variability for $H_2O$ by ~30%, typically, and by as much as a factor of two in some regions; this has some implications for the time needed to detect small trends.

We have derived trends using a multivariate linear regression (MLR) model, and there is a robust signal in both MLS observations and WACCM of an upper stratospheric $O_3$ increase from 2005 to 2014 by ~0.2-0.4%/yr (± 0.2%/yr, 2σ), depending on which broad latitude bin (tropics or mid-latitudes) is considered. In the lower stratosphere, while some decreases are indicated for 1998-2014 (based on merged GOZCARDS $O_3$), we find near-zero or positive trends when using MLS $O_3$ data alone for 2005-2014, albeit with no robust statistical significance. SD-WACCM results track such positive tendencies (albeit with no statistical significance). For $H_2O$, the most statistically significant trend result for 2005-2014 is an upper stratospheric increase, peaking at slightly more than 0.5%/yr in the lower mesosphere, in fairly close agreement with SD-WACCM trends, but with smaller values in FR-WACCM. For HCl, while the lower stratospheric vertical gradients of MLS trends are duplicated to some extent by SD-WACCM, the model trends (decreases) are always on the low side of the data trends. There is little model-based indication (in SD-WACCM) of a significantly positive HCl trend derived from the MLS tropical series at 68 hPa; this deserves further study. For $N_2O$, the MLS-derived trends (for 2005-2012) point to negative trends (of up to about -1%/yr) in the NH mid-latitudes and positive trends (of up to about +3%/yr) in the SH mid-latitudes, in good agreement with the asymmetry that exists



in SD-WACCM trend results. The small observed positive $N_2O$ trends of ~0.2%/yr in the 100 to 30 hPa tropical region are also
consistent with model results (SD-WACCM in particular), which in turn are very close to the known rate of increase in
tropospheric $N_2O$. In the case of $HNO_3$, MLS-derived lower stratospheric trend differences (for 2005-2014) between
hemispheres are opposite in sign to those from $N_2O$ and in reasonable agreement with both WACCM results, despite large error
bars.
6       The data sets and tools discussed here for the evaluation of the models could be expanded to additional comparisons of
species not included here, as well as to model intercomparisons using a variety of CCMs, keeping in mind that there are different
parameterizations and approaches for both free-running and specified-dynamics simulations.
## 1   Introduction
State-of-the art chemistry climate models (CCMs) are known to reproduce the main features of stratospheric climatology and
change, although there has always been some variability between the models (e.g., Waugh and Eyring, 2008; SPARC, 2010;
Dhomse et al., 2018). Free-running CCMs are used to make long-term simulations of atmospheric composition, as well as
predictions of future changes, driven by a large set of photochemical reactions, as well as time-dependent boundary conditions
for surface concentrations of greenhouse gases and ozone depleting substances (ODSs), sea surface temperatures and sea ice
concentrations, 11-year solar variability, sulfate aerosol surface area density, as well as tropospheric ozone and aerosol precursor
emissions. In more recent years, modeling groups have implemented "specified-dynamics" versions that are constrained to
meteorological fields (e.g., surface pressure, temperature, and horizontal and meridional winds). Our main purpose here is to
evaluate these two types of model runs from CESM1 WACCM, using multi-species satellite-derived global composition
datasets; we will refer to these two types as FR-WACCM, for the free-running model version, and SD-WACCM, for the
specified dynamics version. The model simulations were based on scenarios defined by the Chemistry Climate Model Initiative
(CCMI) (Eyring et al., 2013; Morgenstern et al., 2017). Our evaluation focus is on monthly zonal mean time series from the
models versus satellite-derived global data sets. The main stratospheric (and mesospheric) time series used here are from Aura
MLS products (version 4.2 data) and from the Global OZone Chemistry And Related trace gas Data records for the Stratosphere
(GOZCARDS), which include MLS data from late 2004 onward. The GOZCARDS data records include merged multi-satellite
data files for $O_3$, $H_2O$, and HCl, and Aura MLS-derived data files for $HNO_3$ and $N_2O$; these 5 species are used for the model
evaluations herein. We also focus, in part, on the Aura MLS high quality data sets for 2005-2014 (with 2014 being the last year
of WACCM runs considered here). The regular and nearly uninterrupted daily global coverage of the MLS day and night
measurements leads to minimal sampling-related biases, both for climatological comparisons (see Toohey et al., 2013) and trend-
related studies (Millan et al., 2016). This data set also has a well characterized set of error bars (see Livesey et al., 2018, for the
latest update to the data quality documentation); however, we also note that there are some caveats to take into account regarding
long-term stability for some of the MLS species.
In terms of the model/data comparisons, we will analyze the climatological mean state and "goodness of fit" issues, as well as
variability. While one has the expectation that, in general, better fits to the data would be obtained for a specified-dynamics run
than for a less dynamically constrained run (FR-WACCM), one needs to demonstrate this quantitatively with diagnostics that can
provide enough differentiation between models that are sometimes found to track each other closely. There have been essentially
no published trend studies using the Aura MLS data set by itself. However, this data set now covers a sufficiently long time
period that it becomes useful to investigate such trends, as the analyses deal with one homogeneous data set, while removing the
potential issues associated with data merging prior to 2005, whether related to poorer sampling, or to uncertainties in the bias
removal between various data sets. On the downside, a shorter time series will also lead to larger uncertainties in the derived
trends.



We first provide (in Sect. 2) an overview of the global stratospheric data sets used for these comparisons. Brief descriptions
of FR-WACCM and SD-WACCM are provided in Sect. 3. Climatological comparisons between the models and the Aura MLS
data sets are provided in Sect. 4, in order to assess, for example, whether any obvious biases exist; these comparisons include an
overview of the main short-term variations, namely the annual oscillation (AO) and semi-annual oscillation (SAO). More
detailed comparisons of deseasonalized anomaly time series are provided in Sect. 5.1, where we evaluate how well the two
model versions fit the data sets, both in terms of closeness of fits and variability. Finally, trend comparisons are investigated in
Sect. 5.2.
**2    Data sets**
**2.1    Aura MLS**
The Microwave Limb Sounder (MLS) is one of four instruments on NASA's Aura satellite, launched on July 15, 2004. The
MLS antenna scans the atmospheric limb as Aura orbits the Earth in a near-polar sun-synchronous orbit, and the instrument
measures thermal emission (day and night) in narrow spectral channels, via microwave radiometers operating at frequencies near
118, 190, 240, and 640 GHz, as well as a 2.5 THz module to measure OH. MLS (see Waters et al., 2006) has been providing a
variety of daily vertical stratospheric temperature and composition profiles (~3500 profiles per day per product), with some
measurements extending down to the upper tropospheric region, and some into the upper mesosphere or higher. For more
information and access to the MLS data, the reader is referred to  http://disc.sci.gsfc.nasa.gov/Aura/data-holdings/MLS; the
current data version is labeled 4.2$x$ (with $x$ varying between 0 and 3, depending on the date).  Data users interested in MLS data
quality and characterization, estimated errors, and related information, should consult Livesey et al. (2018), the latest update to
the MLS data quality document (available from the MLS website at http://mls.jpl.nasa.gov ).
**2.2    GOZCARDS**
The data considered here for longer-term model evaluation analyses is from GOZCARDS, a data record that was created
using high quality satellite-based Level 2 data as "source" data sets, which were merged together into global monthly zonal mean
records for $O_3$, $H_2O$, and HCl, going back in time before the (2004) launch of Aura.  Readers are referred to the GOZCARDS
description and highlights provided by Froidevaux et al. (2015). In brief, for $O_3$, the original GOZCARDS version 1.01 (v1.01)
data record starts in 1979 with solar occultation measurements from the first Stratospheric Aerosol and Gas Experiment (SAGE
I), and continues with data from SAGE II, the Halogen Occultation Experiment (HALOE), the Upper Atmosphere Research
Satellite (UARS) MLS, the Atmospheric Chemistry Experiment Fourier Transform Spectrometer (ACE-FTS, using solar
occultation), and Aura MLS. Basically, the overlap time periods for different data sets are used to calculate offsets between zonal
mean time series in 10°-wide latitude bins at each pressure level, and the data sets are adjusted to a reference value (SAGE II
mean values, for $O_3$, or an average of satellite measurements for $H_2O$ and HCl); monthly standard deviations are also provided,
along with other diagnostic quantities. GOZCARDS data extensions past 2012 were created simply by adding more recent MLS
data, appropriately adjusted to account for zonal mean differences between versions, once the MLS $O_3$ v2.2 data became
unavailable (for 2013 onward); the latest ACE-FTS data version was not included. GOZCARDS $O_3$ (v1.01) has been used for
past $O_3$ trend assessments and in comparisons to other data records (e.g., WMO, 2014; Nair et al., 2015; Tummon et al., 2015;
Harris et al., 2015; Ball et al., 2017, 2018).
For the GOZCARDS ozone data discussed here, unless otherwise noted, we use GOZCARDS v2.20, a recent improvement





and update to the original version. The ozone v2.20 data set was provided for an updated assessment of stratospheric ozone by
Steinbrecht et al. (2017), as well as for the assessment activities of the Long-term Ozone Trends and Uncertainties in the
Stratosphere (LOTUS) project and in preparation for the latest international report on the state of the ozone layer, led by the
World Meteorological Organization (WMO). GOZCARDS ozone data updates are also used as part of the yearly "State of the
Climate" stratospheric ozone-related summaries, produced for the Bulletin of the American Meteorological Society (BAMS). For
GOZCARDS $O_3$ v2.20, the stratospheric retrieval pressure grid is twice as fine as for v1.01; there are now 12 regularly-spaced
levels per decade change in log of pressure. The UARS MLS $O_3$ data were not included in v2.20, since these retrievals are not
readily available on the finer vertical grid (although approximations such as interpolation could be used); also, there is no easy
provision of UARS MLS retrieval uncertainties on a finer grid. The most significant change for the new merged ozone is the
effect of using the updated and more robust version 7 data from SAGE II (Damadeo et al., 2013). Version 7 uses National
Aeronautics and Space Administration (NASA) Global Modeling and Assimilation Office (GMAO) Modern-Era Retrospective
Analysis for Research and Applications (MERRA) temperature (T) profile data (Rienecker et al., 2011) in the retrievals, and
these values (rather than T from the National Centers for Environmental Prediction (NCEP)) also have a significant impact on
the conversion of SAGE II $O_3$ from its native density/altitude grid to the GOZCARDS mixing ratio/pressure grid. Also, Aura
MLS v4.2 $O_3$ data are now used (instead of MLS v2.2); HALOE v19 $O_3$ profiles are included, after interpolation to the finer
pressure grid before merging. As a result of these improvements in GOZCARDS $O_3$, we have observed closer agreement and
larger correlation coefficients between the Stratospheric Water and Ozone Satellite Homogenized (SWOOSH) data record (Davis
et al., 2016) and GOZCARDS v2.20 $O_3$ time series than between SWOOSH and GOZCARDS v1.01 (SWOOSH $O_3$ also uses
SAGE II v7 ozone data); more details regarding the impact of GOZCARDS v2.20 $O_3$ on trends are provided in Sect. 5 and in the
Appendix.
**3   WACCM (CESM1) description and simulations**
WACCM (CESM1) is a chemistry climate model of the Earth's atmosphere, from the surface to the lower thermosphere
(Garcia et al., 2007; Kinnison et al., 2007; Marsh et al., 2013; Garcia et al., 2017). WACCM is a superset of the Community
Atmosphere Model, version 4 (CAM4), and includes all of the physical parameterizations of CAM4 (Neale et al., 2013) and a
finite volume dynamical core (Lin, 2004) for the tracer advection. The horizontal resolution is 1.9˚ latitude x 2.5˚ longitude. The
vertical resolution in the lower stratosphere ranges from 1.2 km near the tropopause to ~2 km near the stratopause; in the
mesosphere and thermosphere the vertical resolution is ~3km. Simulations used here are based on the guidelines from the
International Global Atmospheric Chemistry / Stratosphere-troposphere Processes And their role in Climate (IGAC/SPARC)
Chemistry Climate Model Initiative (CCMI) (Morgenstern et al., 2017). Improvements in CESM1 (WACCM) for CCMI include
a modification to the orographic gravity wave forcing, which reduced the cold bias in Antarctic polar temperatures (Garcia et al.,
2017; Calvo et al., 2017) and updates to the stratospheric heterogeneous chemistry, which improved the representation of polar
ozone depletion (Wegner et al., 2013; Solomon et al., 2015). In this work, there are two CCMI scenarios, spanning the 1990-
2014 period. The first scenario follows the CCMI REF-C1 definition and three ensemble members were completed; this falls
under the "free-running" scenario. We note that all the analyses herein are based on an average of these three simulations. We
have checked that the three representations' departures from the average are small enough not to require separate comparisons
for each case, when pursuing average or root mean square (RMS) differences versus observations, in comparison to differences
using the 2[nd] model scenario (see below); this is also true for the model/data comparisons of RMS variability. This first model
scenario includes forcing from greenhouse gases ($CH_4$, $N_2O$, and $CO_2$), organic halogens, volcanic aerosol surface area density





and heating, and 11-year solar cycle variability. The sea surface temperatures are based on observations and the quasi-biennial
oscillation (QBO) is nudged to observed monthly mean tropical winds over 86-4 hPa, as described in Matthes et al. (2010).
The second scenario is based on the CCMI REF-C1SD scenario and includes all the forcings of REF-C1, except for additional
external QBO nudging. This scenario uses the specified dynamics (SD) option in WACCM (Lamarque et al., 2012). Here,
temperature, zonal and meridional winds, and surface pressure are used to drive the physical parameterization controlling
boundary layer exchanges, advective and convective transport, and the hydrological cycle. The meteorological analyses are taken
from MERRA and the nudging approach is described in Kunz et al. (2011). The QBO circulation is inherent in the MERRA
meteorological fields and is therefore synchronized with that in the "real" atmosphere. The horizontal resolution is the same as
the REF-C1 version and the vertical resolution follows the MERRA reanalysis (from ~1 km resolution near the tropopause to
about 2 km near the stratopause). The model meteorological fields are nudged from the surface to 50 km; above 60 km, these
fields are fully interactive, with a linear transition in between. Both WACCM versions used here contain an identical
representation of tropospheric and stratospheric chemistry (Kinnison et al., 2007; Tilmes et al., 2016). The species included in
this mechanism are contained within the Ox, NOx, HOx, ClOx, and BrOx chemical families, along with $CH_4$ and its degradation
products. In addition, 20 primary non-methane hydrocarbons and related oxygenated organic compounds are represented, along
with their surface emissions. In total there are 183 species and 472 chemical reactions; this includes 17 heterogeneous reactions
on multiple aerosol types (i.e., sulfate, nitric acid trihydrate, and water-ice). For this work, the CESM1 (WACCM) REF-C1 and
REF-C1SD simulations will generally be referred to as FR-WACCM and SD-WACCM, respectively. While the runs were
originally designed to stop at the end of 2010, for this work, the forcing inputs have been extended through 2014.

## 4    Climatological comparisons and biases

We first describe some of the major climatological features for the various stratospheric species mentioned in the
Introduction. We focus here on the main differences between average model trace gas abundances in the FR-WACCM and SD-
WACCM runs and the corresponding abundances from Aura MLS for 2005 through 2014; this includes a sub-section on annual
and semi-annual variations. Further analyses of interannual variations and trends are discussed in Sect. 5.

### 4.1    Average abundances

We provide climatological latitude/pressure contour plots in the Supplement (Fig. S1 for $O_3$ and $H_2O$, and Fig. S2 for HCl,
$HNO_3$, and $N_2O$), showing Aura MLS and WACCM mixing ratio distributions, averaged over 2005 through 2014. Since such
plots do not easily allow one to quantify the largest areas of model/data disagreement, we show in Fig. 1 (left column, top two
panels) the percent differences between WACCM and the MLS climatologies; a positive value means that, on average, for 2005-
2014, the model values exceed the data values. Also, we show in the right column (top two panels) the absolute value of the
average model/data difference divided by systematic error estimates ($2\sigma$ values) for MLS ozone. These error estimates have been
provided in past MLS validation and error characterization work as tabulated "typical" (global average) profiles as a function of
pressure; the latest update of such error estimates for version 4 MLS data is provided by the MLS team in Livesey et al. (2018).
The vertical profile of estimated systematic errors for MLS ozone is given in the Supplement (Fig. S3). Note that the MLS team
can provide further systematic error details for MLS data users (e.g., for pressure levels not listed in the standard Tables). Past
validation references for MLS ozone include Jiang et al. (2007), Froidevaux et al. (2008a), and Livesey et al. (2008), as well as
the more recent work covering many satellite (and other) instruments by Hubert et al. (2016). The original MLS data validation



work for $H_2O$ is from Read et al. (2007) and Lambert et al. (2007), who also described $N_2O$ validation, whereas $HNO_3$ validation
work was discussed by Santee et al. (2007).
The two bottom panels of Fig. 1 show a comparison between the two model runs (with percent differences on the left and the
ratio of the absolute differences versus data on the right). The main conclusion from Fig. 1 is that most of the model ozone
climatology falls within about 5 to 10% of the data climatology, except in the upper troposphere and lower stratosphere (UTLS),
where the SD-WACCM $O_3$ values are even larger than those from FR-WACCM (see bottom left panel). Our work focuses on the
stratosphere, but both FR-WACCM and SD-WACCM average $O_3$ values at low latitudes are lower than the observed mean
values from 215 to 261 hPa (MLS retrieval pressure levels that lie in the upper troposphere at these latitudes). While the right
column of Fig. 1 indicates that the difference to error ratio does not show a very significant systematic difference in this region,
there are known MLS positive biases versus tropical ozonesonde data, and this could account for at least part of the apparent
model low bias, at least for the 215 hPa level; between this level and the tropopause (near 100 hPa), the SD-WACCM model
values appear to be biased somewhat high. On the other hand, $O_3$ mid-latitude values from 100 to 215 hPa are biased high in SD-
WACCM in particular, with a difference to systematic error ratio larger than 2 to 3 in most of this region; otherwise, Fig. 1
shows that these ratios are typically less than 1 to 1.5. An illustration of the more significant differences is given in Fig. 2, for the
$O_3$ data and model time series at 215 hPa for 50°N-60°N (these abundances represent lower stratospheric values). This shows
that both FR-WACCM and SD-WACCM average values are larger than the data in this region, and more so for SD-WACCM. In
relation to this model overestimate of ozone, Imai et al. (2013) have shown that the SD-WACCM $O_3$ values are also larger than
those from the Superconducting Submillimeter-Wave Limb-Emission Sounder (SMILES) in the lowest portion of the
stratosphere (near 18-20 km). In Fig. 2, we also observe that the model ozone annual amplitude (AO) is larger than the observed
amplitude; we discuss the stratospheric AO more generally in the next section. If anything, MLS $O_3$ at mid-latitudes is biased
slightly high (by roughly 5%) with respect to a multi-instrument mean ozone field based on a large number of monthly zonal
mean satellite data sets from the Stratosphere-Troposphere Processes And their Role in Climate (SPARC) Data Initiative (DI), as
discussed by Tegtmeier et al. (2013), who also showed that the MLS $O_3$ seasonal cycle at mid-latitudes and 200 hPa is in very
good agreement with the multi-instrument mean. The limb emission measurements from Aura MLS for the species discussed
here provide generally strongly peaked vertical averaging kernels with a resolution of 2.5-4 km (see Livesey et al., 2018, for
sample plots of these kernels). We have confirmed that smoothing the model profiles using the MLS averaging kernels (and a
priori information) gives very little change (less than a few % for the Fig. 2 example, and even less at higher altitudes) in $O_3$
abundances and seasonal cycles, and we see no real need to use such a smoothing for the model/data comparisons herein; the
significant model/data differences in Figs. 1 and 2 are thus not caused by this sort of issue.
Figure 3 is the $H_2O$ analog of Fig. 1, for pressures reaching up to 0.01 hPa, since the Aura MLS $H_2O$ retrievals cover both the
stratosphere and the mesosphere; again, a typical profile of MLS systematic error estimates (Livesey et al., 2018) has been used
in deriving the results shown in the top two right panels, which test for the existence of a significant bias between model and
observations. We note that MLS $H_2O$ version 3 stratospheric data generally exhibit a slight high bias (of a few to 5%) versus
multi-instrument mean values, with a somewhat larger positive bias (of ~10%) in the lower mesosphere (Hegglin et al., 2013).
Such biases are within the expected measurement systematic errors; MLS version 4 stratospheric $H_2O$ data show essentially no
systematic change in comparison to version 3 (Livesey et al., 2018). FR-WACCM and SD-WACCM $H_2O$ mean values are on the
low side (by ~5-15%) relative to MLS $H_2O$ in the upper stratosphere and in most of the mesosphere (see Fig. 3), implying that
the models are in good agreement with the SPARC DI multi-instrument mean $H_2O$. There is independent evidence that MLS
$H_2O$ has a dry bias near the hygropause (at the low end of the vertical range shown in Fig. 3, where the bottom level is 150 hPa);
this has been known for some time (see Read et al., 2007; Vömel et al., 2007). This is also consistent with the existence of a





model high bias relative to MLS near 150 hPa, assuming that the models come close to representing the $H_2O$ climatology in this
region. In terms of the significance of the biases, the right top two panels of Fig. 3 indicate that the $H_2O$ model/data comparisons
are generally in agreement within the estimated ($2\sigma$) systematic errors, and that this level of agreement is slightly better in the
case of SD-WACCM.
For HCl, the climatological comparisons of Fig. 4 show that both models exhibit a small (5-10%) low bias versus MLS HCl
in much of the stratosphere, with a stronger negative model bias in the tropical region between 100 and 150 hPa. The model/data
relative biases in stratospheric HCl are generally within the MLS HCl systematic errors (see Fig. 4, top two panels at right). MLS
HCl is slightly on the high side of the multi-instrument mean climatological results provided in the SPARC DI report (SPARC,
2017). The small negative model bias in the upper stratosphere could also arise from the lack of a sufficiently pronounced
decrease in upper stratospheric MLS HCl, as a result of the interruption in the main MLS HCl target band (band 13) data after
early 2006 (see Livesey et al., 2018). There is also a known strong positive systematic bias in MLS HCl at 150 hPa in the tropics
(see Froidevaux et al., 2008b), so model underestimates in this region are not a sign of model weakness. We also note that both
models exhibit a systematic difference versus HCl observations in the lower stratosphere (with larger differences for SD-
WACCM), as well as a downward sloping pattern (equator to pole) in the southern hemisphere (SH), and smaller mean
differences (for SD-WACCM) in the northern hemisphere (NH).
Figure 5 provides average comparisons for $N_2O$ (a good dynamical tracer). While the mean lower stratospheric SH $N_2O$
values are larger for SD-WACCM than for FR-WACCM (bottom left panel of Fig. 5), the mean absolute fit for SD-WACCM
versus MLS is not significantly better. The most significant climatological differences with respect to the error bars (top two
right panels in Fig. 5) are in the upper stratosphere at low latitudes; in this region, SD-WACCM agrees somewhat better with
MLS. However, this is also where the mean abundances decline rapidly with height towards the limits of the MLS sensitivity.
Not too surprisingly, this is also where the SPARC DI results for $N_2O$ show the largest scatter in terms of percent differences
(often exceeding 10-20%, see SPARC, 2017).
Finally, the climatological comparisons for $HNO_3$ (Fig. 6) reveal very few areas of mean model/data disagreements
significantly outside the systematic uncertainties. However, there is a general model underestimation, especially in the polar
upper stratosphere. We will see later that there are large recurring model/data differences in the upper stratosphere during certain
months. At high latitudes in the lower portion of the stratosphere, the models tend to overestimate the data. The upper
troposphere is where the satellite-based $HNO_3$ data have been validated the least, but there is some evidence for a high MLS bias
in this region, based on SPARC DI results (see SPARC, 2017). While this might explain, at least qualitatively, why the models
are observed to underestimate MLS tropical UT $HNO_3$ (see Fig. 6), more work is needed to better evaluate $HNO_3$ from models
and satellite-derived data sets in this particular region.
It is also interesting to point out modeled and observed seasonal changes in the polar lower stratosphere over Antarctica.
Figure 7 depicts average seasonal changes over the 70°S-80°S region at 46 hPa for 2005-2014, using Aura MLS observations
and model values for comparison. We wish to emphasize the slope of the early winter decline in HCl, and we see that the model
HCl values do not decline as fast as indicated by the data, even though SD-WACCM tracks the interannual variability better than
FR-WACCM does (see Fig. S4 for the relevant time series from 2005 through 2014). Many of the options and uncertainties
regarding lower stratospheric heterogeneous chemistry modeling for SD-WACCM at high latitudes in the polar winter/spring
have been discussed by Solomon et al. (2015), who have pointed, for one specific year (2011), most of the features depicted in
Fig. 7. For our broader time period, we see that the HCl rate of decline from May to July (dominated by nighttime conditions) is
slower in both the SD-WACCM and FR-WACCM results than the corresponding mean HCl rate of change from MLS (top left
panel of Fig. 7). Grooß et al. (2018) have recently discussed this HCl model/data discrepancy for dark polar vortex conditions,



including the potential impact of numerical diffusion issues in the Eulerian models, in comparison to their simulations using the Chemical Lagrangian Model of the Stratosphere (CLAMS), which shows even larger HCl discrepancies. These authors discuss some possible mechanisms and uncertainties, and they argue that additional decomposition of condensed-phase $HNO_3$ might play a role, possibly via galactic cosmic ray impacts. Since this rapid decline in HCl occurs during polar night, these authors point out that this early HCl issue does not lead to much difference in polar ozone loss rates, which only become significant during sunlit conditions (early spring). Figure 7 confirms that, on average, the SD-WACCM $O_3$ decline and rise match the data well. We also note that FR-WACCM shows smaller-than-observed declines in $HNO_3$ and $H_2O$, whereas SD-WACCM matches these observations much better. The temperature panel (bottom center) gives a possible reason for these differences, as T from FR-WACCM is higher by a few degrees during the coldest phase than T from SD-WACCM (MERRA-based), and larger than the MLS-derived values. This would lead to less irreversible denitrification and dehydration. The general nature of the results shown in Fig. 7 is similar at other lower stratospheric pressures (and for latitudes poleward of 80°S), although there is some variability in the magnitude of the differences. Over the Arctic region (not shown here), temperature-related differences are not as systematic or as large as over Antarctica, but similar model/data differences in the early winter rate of decline in HCl exist there also (as mentioned by Grooß et al., 2018).

As an addendum regarding the evaluation of models in comparison to data sets, we provide in Appendix (A1) the results of a model grading approach that has been used in the past (e.g., Douglass et al., 1999, Waugh and Eyring, 2008). We find (see Appendix A1) that this grading method often leads to low grades (see Fig. A1), if applied using systematic uncertainty estimates from the MLS data characterization work (Livesey et al., 2018). The model results, as good as they are in many respects, cannot always match the data closely enough, at least based on such grades (although the grading formulation could also be reconsidered). As mentioned in the Appendix, the multiplicative error factor (see Equation A1) can be increased (e.g., from 2 to 4) to force these grades (see Fig. A2) to span a more useful range (in the plots). Indeed, we observe similarities between Fig. A2 (top two panels) and Fig. 1 (middle panels): poorer SD-WACCM performance for pressures ≥ 100 hPa, better mid-latitude results near 30-40 hPa, and poorer performance again near 3 to 5 hPa. Also, we observe the poorest $H_2O$ grades near 1 hPa in Fig. A2 (bottom panels), which is similar to the poorer performance in the middle panel of Fig. 3, while the best stratospheric $H_2O$ grades are found near 10-20 hPa, which matches the best performance (smallest values) in Fig. 3 (top two panels at right). In the case of merged data records, we note that it is generally more difficult to estimate systematic errors. The GOZCARDS data analyses led to conservative (i.e. possibly somewhat pessimistic) systematic error estimates as a function of latitude and pressure (see Froidevaux et al., 2015) that are significantly larger than the systematic error estimates for MLS data only. There are other methods that could lead to useful error estimates, through the use of multi-satellite data sets and the spread between these (see SPARC, 2017), for some species at least. Moreover, when one considers relative variations such as anomaly time series (as done in a subsequent section), it becomes even less clear how to best assign uncertainties in the context of "error-weighted" grades; some data records may also drift with respect to others, or with respect to ground-based data, so that the actual errors will change with time (and possibly with location as well), in a difficult to determine way. We do not pursue this more traditional grading approach further here, especially as the two models we are comparing often lie fairly close together. For multi-model comparisons (which is outside the scope of this work), one could consider how to best apply grading methods such as the one in Appendix A1, along with other diagnostics such as those discussed here; in the end, the most important aspect of such analyses possibly lies in the relative values of the grades or diagnostics for different models.



## 4.2    Annual and semi-annual cycles

Figure 8 displays the amplitudes of annual and semi-annual variations for MLS ozone and the corresponding FR-WACCM and SD-WACCM runs for 2005-2014. We obtained these results from a simple regression fit to the monthly mean time series in each latitude/pressure bin. The primary time dependence of the fitted function is given by additive sine and cosine terms (with 12-month and 6-month periods), in addition to constant and linear trend terms; the AO and SAO amplitudes are given by the square root of the sum of the squares of the corresponding fitted coefficients. We see from Fig. 8 that the overall data and model patterns of AO and SAO variability are quite similar. The ozone AO amplitudes peak (in ppmv) at mid- to upper stratospheric levels, with high latitude variations also observed as a result of the effects of winter/spring polar chemistry and dynamics; the lower stratospheric peak AO amplitudes are more prominent over the southern polar regions, where stronger $O_3$ depletion occurs on a seasonal basis. These MLS $O_3$ AO patterns are very similar to those obtained by Schoeberl et al. (2008), using a much shorter time period (Sep. 2004 to Dec.2006); the same holds for other species ($H_2O$ and $HCl$) considered in that work and here. The observed SAO amplitude for $O_3$ exhibits strong peaks in the upper stratosphere, both in the tropics and at high latitudes. The anti-correlation between $O_3$ and temperature as a result of temperature-dependent photochemical production and loss terms for $O_3$ has long been known to cause most of the $O_3$ variability in the upper stratosphere (see Perliski et al., 1989, in relation to the AO and the SAO). The AO and SAO amplitudes obtained in that and other past studies (e.g., see Ray et al., 1994) are very similar to the amplitude patterns shown here. If we look more closely (and based on AO amplitude ratio plots not shown here), there are often $O_3$ AO amplitudes 20-80% larger than those derived from MLS data for both WACCM runs in the lower stratosphere (from 50 hPa at low latitudes to 215 hPa at high latitudes); such a model overestimate of the AO amplitude was shown in the time series example of Fig. 2. More generally, outside of this region, we observe somewhat closer fits to the MLS AO amplitudes for the SD-WACCM version, but both models track the data and each other well, with AO amplitudes typically within ~25% of the MLS AO amplitudes. For the Antarctic lower stratosphere, the timing and magnitude of the seasonal recovery after the ozone hole plays a role, and in this respect, we have observed that SD-WACCM generally fits the MLS data better than FR-WACCM does. In the tropical upper stratosphere, where the SAO is larger than the AO (see Fig. 8), the SD-WACCM results match the observed SAO amplitudes slightly better than those from FR-WACCM. Despite the existence of a few model/data differences, these AO and SAO amplitude comparisons, coupled with our examination of model/data amplitude ratio plots as well as the time series (which include the phase information), do not elicit major concerns regarding the model characterization of the primary processes expected to govern these modes of $O_3$ variability. The study of dynamical forcing mechanisms in relation to such modes continues to be an active area of research (e.g., see Ern et al., 2015, for a discussion of wave-driving and the SAO). Also, Smith et al. (2017) have recently shown that analyses of Aura MLS geopotential height data lead to derived tropical zonal mean winds that agree well with those derived from Sounding of the Atmosphere using Broadband Emission Radiometry (SABER) geopotential heights, and with direct wind data, thus enhancing our knowledge of tropical atmospheric dynamics (including the SAO and QBO).

For $H_2O$, a similar overview of the AO and SAO amplitudes is given in Fig. 9, which covers the vertical range from 100 to 0.01 hPa. A peak in these amplitudes resulting from the seasonal downward transport before the winter, followed by wintertime dehydration, is observed in the lower stratospheric southern polar region; we note that the SD-WACCM results match this feature better than the FR-WACCM simulation does. Other interesting features include the southern hemisphere's upper stratospheric AO peak in the extra-tropical region. This has been seen by many satellite-based measurements, as discussed in the comparisons by Lossow et al. (2017a). Lossow et al. (2017b) have explained this "island" feature in more detail, with the help of model simulations and analyses; they argue that vertical advection tied to the upper branch of the Brewer-Dobson circulation largely explains the seasonal highs (lows), via downwelling (upwelling). They also show that an AO maximum is observed as





well in other species in roughly the same region, including in $N_2O$ MIPAS data; we confirm this behavior (see also Fig. S5) from the $N_2O$ AO amplitude feature observed in MLS data, as well as in the WACCM runs (and more so in SD-WACCM). The derived AO and SAO amplitude patterns in $H_2O$ from Lossow et al. (2017a) are consistent with what we show here; this includes the peak values in the upper stratosphere and mesosphere, attributed to the combined effects of photochemistry and vertical transport. For ozone, a dominant feature in the SAO amplitude exists in the tropical upper stratosphere; see Lossow et al. (2017a) for a brief review of past work explaining such dynamically-driven features for $H_2O$. While there is generally a good level of model/data agreement in the main $H_2O$ AO and SAO patterns, both WACCM comparisons tend to underestimate observed AO and SAO amplitudes in the lower stratosphere and overestimate AO amplitudes in the SH mesosphere, while slightly underestimating the mesospheric SAO amplitudes in both polar regions. The largest amplitude differences reach a factor of two, in places, for the lower stratospheric model underestimates, which cannot be caused by slight model underestimates of average MLS $H_2O$ (as seen in Fig. 3); however, we note that the lower stratosphere is also the region where AO and SAO amplitudes are smallest (typically < 0.1 ppmv).

Turning briefly to the other species discussed here, for $N_2O$, we have already mentioned the existence of the upper stratospheric AO peak (the "island" feature described by Lossow et al., 2017b) in the southern hemispheric extra-tropical region. This is similar to the $H_2O$ AO amplitude maximum feature, but the $N_2O$ seasonal variations are anti-correlated with $H_2O$, as demonstrated by Lossow et al. (2017b), using MIPAS data (and as is also apparent in the MLS time series, not shown here). Furthermore, we observe in Fig. S5 a somewhat better match in the AO and SAO $N_2O$ amplitude patterns for SD-WACCM than for FR-WACCM versus MLS, in particular for tropical to southern mid-latitudes. We note that this is also manifested in better time series fits for SD-WACCM, besides the closer match in average values; as mentioned in the Introduction, this is what one would generally expect for the two model versions. We saw also in Fig. 5 that FR-WACCM overestimates the average values of upper stratospheric tropical $N_2O$. For HCl and $HNO_3$, the AO and SAO amplitudes are dominated by the variations at high latitudes (see Figs. S6 and S7). The model HCl upper stratospheric AO and SAO amplitudes match up fairly well with the observed amplitudes, despite the aforementioned issues relating to MLS upper stratospheric HCl trends. The NH lower stratospheric polar variations are somewhat underestimated by the models, and more so by FR-WACCM run, which (based on time series plots) exhibits larger values at the low end and smaller values at the high end of the HCl summer to winter cycle. For $HNO_3$, the main AO and SAO model features follow the MLS patterns, although there is a model underestimation of the amplitudes in the upper stratospheric polar regions, because these models do not properly capture the observed recurrences of enhanced $HNO_3$, as mentioned earlier (see also Sec. 4).

## 5    Time series comparisons

### 5.1    Anomaly time series: fits and variability

#### 5.1.1    Fits

We wish to evaluate which of the two WACCM models (free-running or specified dynamics) provides a better match, or fit, to the temporal variations in observed deseasonalized anomalies. Again, we would expect SD-WACCM to generally fit these anomalies better than FR-WACCM. We analyze the model and data deseasonalized anomaly time series, obtained by differencing each month's zonal mean value from the long-term averages for the same month. We then calculate a diagnostic of model fit to the data, by using the RMS differences between these deseasonalized model and data series, and normalize by dividing this quantity by the RMS of the data anomalies themselves. Thus, a diagnostic value that is much less than unity means





that the match to the time series is much smaller than the typical variability; this also implies a good fit to the observed
anomalies. In the Appendix (A2), we provide the mathematical expression for this "RMS difference diagnostic". A better model
fit to the observed anomaly series will be represented by a smaller "RMS difference diagnostic" value. We also calculate the
standard (Pearson) correlation coefficients, R, between model and data anomalies, and we use $R^2$ as another measure of
"goodness of fit" for the models. The first diagnostic is unitless and does not depart too much from the 0 to 1 range; $R^2$ is limited
to the 0 to 1 range, with larger values indicating a higher degree of linear correlation. We also use these two diagnostics together,
by calculating the ratio of $R^2$ over the RMS difference diagnostic to obtain a "combined diagnostic". This diagnostic could have
a large value (and a good model result) from both a large $R^2$ value (in the numerator), meaning a high correlation with observed
anomalies, and a small RMS difference (in the denominator), implying a good fit to these observations. This also tends to
amplify the differences between two model comparisons to the same data series. An ideal model fit would correlate tightly in
time to observed oscillations, but also exhibit the right magnitude for these variations by "hugging" the anomaly series without
being off in magnitude. Indeed, two model series could have oscillations in phase with data variations and thus the same $R^2$
values, but with different amplitudes (and different overall fits); conversely, two model series could have different $R^2$ values
versus observations, if one is more out of phase than the other, but they could still produce similar RMS difference fits.
In Fig. 10, we display latitude/pressure contour plots of the above diagnostics for FR-WACCM and SD-WACCM $O_3$
anomalies in relation to MLS anomalies for 2005-2014. We immediately see from the top two panels that the SD-WACCM RMS
difference diagnostic values (in the 0.2 to 1 range in the stratosphere) are smaller than those from FR-WACCM (with typical
values between 0.8 and 1.2). Values of $R^2$ (middle panels) show that SD-WACCM also correlates very well with the
observations from 2005-2014, with values typically between 0.7 and 0.95 in most of the stratosphere, and somewhat poorer
correlations in the UTLS. The FR-WACCM ozone series tend to correlate fairly well with the data at low to mid-latitudes for
pressure levels between about 70 and 7 hPa, which is also where the FR-WACCM RMS difference diagnostic shows better
performance than in other regions, and (as an explanation) where the dynamics are nudged (to tropical winds) in a similar way as
for SD-WACCM. However, FR-WACCM shows poor performance (almost zero correlation) at high latitudes and in all of the
uppermost stratosphere.
Figure 11 shows sample $O_3$ series for the upper stratosphere (at 2.2 hPa) for 0°-10°N as well as 40°N-50°N. Although the
tropical series show good correlations overall for both models versus data, the details of observed semi-annual peaks and the
interplay between the AO, SAO, and the quasi-biennial oscillation (QBO) are better followed by the SD-WACCM curve. The
differences in $O_3$ amplitude and phase are more clearly displayed in the bottom (left) panel, which shows the deseasonalized
anomalies. Diagnostic values provided in this panel for both models distinctly show that SD-WACCM performs better than FR-
WACCM here, with a much larger $R^2$ value and a smaller RMS difference diagnostic value, and hence, a much better (larger)
combined diagnostic value. The same comments apply to the right two panels of Fig. 11, which showcase the NH mid-latitudes
at 2.2 hPa. In the high latitude lower stratosphere, the poorer FR-WACCM results in Fig. 10 are generally caused by time series
that are observed to be less in-phase with the polar winter/spring variations, as well as by more departures in the magnitude of
such variations; we will return later to sample polar time series for ozone and other species. The bottom two panels in
Fig. 10 amplify the differences between the models, with values of the combined diagnostic below 1 in most regions for FR-
WACCM, but values between 1 and 3 for the whole stratosphere in the SD-WACCM case. The more realistic dynamics in SD-
WACCM, coupled with the same chemistry as FR-WACCM, allow for better SD-WACCM fits to the data, as shown
quantitatively in Figs. 10 and 11. These plots also point to poor results for both models in the upper troposphere, albeit with
somewhat better results for SD-WACCM, although this is not the focus of this paper.



We now turn to Fig. 12 for a description of the same diagnostics as above, but for model/data $H_2O$ comparisons, and a top
pressure level at 0.01 hPa. We observe, again, that the diagnostics of fit are usually much better for SD-WACCM, which yields
$R^2$ values of 0.6 to 0.9 and RMS difference diagnostic values below 1 for most of the stratosphere and lower mesosphere, and
therefore better combined diagnostic results than FR-WACCM as well. The SD-WACCM diagnostics themselves worsen in the
upper mesosphere and in the high latitude regions near 215 hPa. In the upper mesosphere, the better diagnostic results for SD-
WACCM are seen in the time series (not shown here) as a better match versus the MLS $H_2O$ series anomalies in terms of the
interannual variability at all latitudes, as well as some seasonal peaks at high latitudes; we interpret this as the result of a better
dynamical representation of the mesosphere for SD-WACCM. We note that the high-quality representation of mesospheric
composition by SD-WACCM is also demonstrated in comparisons to measurements of CO profiles above Kiruna, Sweden, by
MLS and the Kiruna Microwave Radiometer (Ryan et al., 2018). For $H_2O$ near 200 hPa, poor fits at high latitudes occur where
MLS is known to have a significant dry bias versus sonde and Atmospheric Infrared Sounder (AIRS) data, as discussed
previously in MLS data documentation (Livesey et al., 2018), as well as by Vömel et al. (2007) and Davis et al. (2016). MLS
$H_2O$ is low by a factor of several here versus the WACCM runs (which show values of 10-60 ppmv). The data variability may
also be affected by the dry bias retrieval issues at the lowest altitudes for these high latitude regions, where observed anomalies
are more poorly tracked by the models; a planned future update to the MLS $H_2O$ retrievals might help to mitigate this
discrepancy.
Figure 13 displays results similar to Fig. 10 but for stratospheric HCl. SD-WACCM HCl results versus MLS are also
generally superior to those from FR-WACCM and show high correlations ($R^2 > 0.7$) in most of the lower stratosphere, with
somewhat poorer results in the upper stratosphere, where the RMS difference fits as well as the combined diagnostic are poor for
both models. We trace the upper stratospheric issue back to a known data problem in this region, where MLS HCl trends are too
flat and the models depart more from the observations (down to about 10 hPa), notably in the RMS difference diagnostic. Poorer
correlations are observed for FR-WACCM at high latitudes, which we will return to later for the lower stratosphere, but even
fairly subtle differences in the timing (phase) can lead to significantly poorer correlations. The poorer fits at 100 hPa in the deep
tropics are related to a large underestimate of the mean data, which may be caused, at least in part, by an MLS high bias
(Froidevaux et al., 2008b), but more so, for $R^2$, by some out-of-phase variability as well. It will be difficult to resolve this issue
until more realistic (lower) MLS HCl values are obtained, and as the phasing may also change with new retrievals.
For the dynamical tracer $N_2O$, we also observe in Fig. 14 (showing model comparisons to the 68 to 1 hPa observations from
the 190 GHz MLS $N_2O$ band for 2005-2014) that SD-WACCM fits the data better than FR-WACCM, in both the $R^2$ and the
RMS difference categories. FR-WACCM exhibits poor results in the upper stratosphere and at high latitudes in the lower
stratosphere. Both models exhibit poorer RMS fits and poorer correlations in the tropical lower stratosphere. Partly, this appears
to be caused by a model underestimation of the MLS $N_2O$ variability in this region, with some QBO phasing differences as well.
$HNO_3$ results (see Fig. 15) show, again, better fits to the stratospheric MLS data from SD-WACCM than from FR-WACCM,
and poor performance from FR-WACCM at high latitudes. Both models do poorly in the upper stratosphere, and Fig. 16
illustrates the magnitude of this discrepancy in the region (3.2 hPa and 70°S-80°S) where it reaches its maximum, in terms of
mixing ratio values. Since its launch, MLS has been observing very large values of $HNO_3$ in the upper stratosphere, mostly in the
polar regions during winter. The WACCM runs used here do not have the right chemistry in the mesosphere and upper
stratosphere to adequately represent such variations; implementation of the many necessary missing chemical reactions has not
made its way into most CCMs. The solution is believed to be tied to ion cluster chemistry during energetic particle precipitation
(EPP) events, which includes large solar proton events (SPEs) as well as more regular auroral-type activity. The study of upper
stratospheric NOx enhancements tied to auroral activity and other EPP events has a long history based on other satellite





measurements and modeling (e.g., Kawa et al., 1995; Callis and Lambeth, 1998; Siskind et al., 2000; Orsolini et al., 2005;
Randall et al., 2007; Reddmann et al., 2010). Yearly upper stratospheric enhancements in ground-based microwave retrievals of
$HNO_3$ profiles over Antarctica were discussed by de Zafra et al. (1997) and de Zafra and Smyshlaev (2001). Direct high altitude
EPP effects enhance NOx, which can propagate downward in polar winter and increase stratospheric NOx and $HNO_3$ via this
indirect effect and conversion of $N_2O_5$ on ion water clusters (Böhringer et al., 1983). Large polar enhancements in upper
stratospheric $HNO_3$ (and other species) were observed by the Michelson Interferometer for Passive Atmospheric Sounding
(MIPAS) after significant SPE activity in 2003, as presented in several papers (Orsolini et al., 2005; von Clarmann et al., 2005;
Lopez-Puertas et al., 2005; Stiller et al., 2005). More complex modeling using modified chemistry and transport-related effects
(e.g., Jackman et al., 2008; Funke et al., 2011; Verronen et al., 2011; Kvissel et al., 2012; Andersson et al., 2016) has produced
EPP-induced enhancements in high latitude $HNO_3$, with related improvements in model/data comparisons into the mesosphere,
and in comparisons of other species. Regarding the low latitude upper stratospheric $HNO_3$ comparisons, the poorer model fits
(even for SD-WACCM) seem to be caused at least in part by more noisy and variable MLS data, under low $HNO_3$ conditions.
Finally, tropical MLS $HNO_3$ data at 147 and 215 hPa are not fit well by either model, as the data exhibit significant seasonal
oscillations between 0.2 and 0.5 ppbv (with larger amplitudes occurring at 147 hPa), whereas model values are smaller than 0.1
ppbv. There have been very few tropical UT validation comparisons for $HNO_3$ (see Santee et al., 2007), but in situ $HNO_3$ data
from an airborne chemical ionization mass spectrometer have indicated that UT $HNO_3$ tropical mixing ratios are mostly below
0.1-0.2 ppbv (Popp et al., 2007, 2009).
For a more in-depth look at the lower stratosphere over Antarctica, Fig. 17 displays anomaly time series comparisons for $O_3$
and temperature at 68 hPa and 70°S-80°S, along with the diagnostic quantities that we are using. We note (top panel) the poorer
$O_3$ correlations for FR-WACCM ($R^2 = 0.22$) than for SD-WACCM ($R^2 = 0.89$), as well as the poorer RMS difference values
(0.89 for FR-WACCM versus 0.33 for SD-WACCM), with correspondingly poorer results in the FR-WACCM combined
diagnostic (0.25) versus SD-WACCM (2.67). Similar differences in the diagnostics are obtained in the bottom panel for
temperature (T) anomalies, showing excellent agreement between SD-WACCM and retrieved temperature anomaly time series
from MLS, both for the $R^2$ and RMS fit diagnostics. One should not expect the free-running model, even if it has certain tropical
QBO-related constraints that mimic those from the SD version, to perform as well in terms of predicted high latitude
temperatures as the SD-WACCM run, driven by realistic (MERRA) meteorological winds, as well as temperatures, which match
up closely to the observed temperatures from MLS. These plots also show that springtime anomalies dominate the variability,
with warmer than usual springs (during October in particular), such as 2012 and 2013, leading to more positive ozone anomalies,
i.e. less ozone depletion; conversely, years (2006, 2008, 2010, 2011) with colder than usual springtime conditions are correlated
with negative ozone anomalies and more depleted conditions. Other factors (besides local mean temperatures) can significantly
influence interannual variability and longer-term ozone loss over Antarctica; this includes the strength of the vortex, total
chlorine abundances, the phase of the QBO, tropospheric wave driving, the timing of warming events, and the impact of aerosols
(e.g., Scaife et al, 2005; Parrondo et al., 2014; Strahan et al., 2015; Langematz et al., 2016; Solomon et al., 2016). We saw in
Sect. 4 that there are generally good comparisons between SD-WACCM and MLS variations over the Antarctic region during
polar winter/spring, except for the rate of HCl decline during early winter; also, poorer results are obtained by FR-WACCM. We
also find (not too surprisingly) that interannual differences in lower stratospheric chemical evolution over Antarctica are not as
faithfully reproduced by FR-WACCM as by SD-WACCM, although we do not show more related details here.



### 5.1.2  Variability

Given our expectations that SD-WACCM would match up better than FR-WACCM to the observed time series of multiple species, and having demonstrated this in the previous section, we turn to what should be a more fair comparison between the two sets of model results, namely the variability aspect. We calculate the ratio of model to data interannual variability, as obtained from the root mean square values of deseasonalized monthly anomaly time series, expressed as a percent of climatological (full period) means; a simple linear trend is first subtracted from the time series, so that the variability comparisons remove any significant trend differences. We do this for the MLS data considered previously (starting in 2005), but also for longer-term time series, using the GOZCARDS data records. The models are sampled monthly following the monthly sampling of the data sets (but not at the daily sampling level of detail); sampling plays a role for the longer-term (merged) GOZCARDS data, which are comprised of some unevenly-sampled occultation data records (depending on latitude and pressure). Figure 18 compares the ozone variability ratios for models versus data using as a reference the MLS 2005-2014 data (top two panels), and the GOZCARDS merged ozone (1992-2003) data (bottom two panels). To first order, we observe similar patterns for both time periods. The SD-WACCM variability is generally within 10-20% of the data variability (ratio values between 0.8 and 1.2). The FR-WACCM variability is generally somewhat smaller than the data variability in the polar regions and in the upper stratosphere. In a recent study, Bandoro et al. (2017) also found that the free-running version of WACCM displays somewhat smaller ozone variability in the upper stratosphere, both for shorter-term and longer-term variabilities, than the observed variability, based on the merged SWOOSH $O_3$ data record. In the lower stratosphere at low to mid-latitudes, we observe that FR-WACCM exhibits slightly larger variability than the data, whereas SD-WACCM shows slightly smaller variability than the data; Bandoro et al. (2017) found that FR-WACCM slightly overestimates the decadal variability in this region. If a free-running model exhibits significant differences versus observed variability, this has some implications regarding predictions of trend detection feasability, as free-running models are used for such predictions; indeed, trend error bars increase if the variability increases. We also see in Fig. 18 that at high latitudes, FR-WACCM tends to underestimate the actual variability, which means that interannual swings in lower stratospheric ozone, in particular, are more muted in the model, whereas this is less of an issue for SD-WACCM, with its more realistic representation of the dynamics; as an example, refer to Fig. 16, for model and data anomalies at 68 hPa and 70°S-80°S.

A similar overview of model/data variability ratios is provided for $H_2O$ in Fig. 19, which covers both the mesosphere and the stratosphere. In this case, while the variability from SD-WACCM is somewhat closer than that from FR-WACCM to the data variability during both time periods, the tendency for both models is to underestimate the observed variability, with FR-WACCM exhibiting a stronger underestimate in the upper stratosphere and mesosphere. A variability underestimate for FR-WACCM implies that any trend detection in the future will require more years of data in the real atmosphere, if $H_2O$ continues to have larger variability than model expectations. Also, this will translate into smaller estimated uncertainties in the model-derived trends in comparison to the observations (as we will see in the next section on trends). The FR-WACCM underestimate of the variability is sometimes by as much as a factor of two, although it is more typically by ~30% (see Fig. 19). For a time series with RMS variability about the fit represented by $\sigma_t$, the number of years needed to statistically detect a trend is proportional to $\sigma_t^{2/3}$ (Weatherhead et al., 1998), and thus, an increase of $\sigma_t$ by factors of 1.3, 1.5, and 2.0, for example, will lead to an increase in the number of years for trend detection by factors close to 1.2, 1.3, and 1.6, respectively. In the tropical lower stratospheric case, $H_2O$ and temperature values and anomalies for 1992-2014 are shown for 100 hPa and 10ºS-20ºS in Fig. 20. Again, we note the smaller-than-observed variability in the model $H_2O$ oscillations, with SD-WACCM tracking the data better. This correlates with the temperature series, where smaller variability is seen in FR-WACCM, in comparison to SD-WACCM (which follows the MERRA temperatures); we also note that FR-WACCM temperatures are somewhat larger (by ~1K on average) than SD-



WACCM temperatures in this region. It is well known that stratospheric entry level $H_2O$ is governed by temperatures near the
tropopause "cold trap"; the monthly average variations shown here are similar to what has been shown in past $H_2O$ work (e.g.,
Randel et al., 2004, 2006; Randel and Jensen, 2013). Brinkop et al. (2016) used model runs from both free-running and nudged
simulations to analyze the impacts of different constraints, including sea surface temperatures (SST) and meteorological fields,
on "sudden" drops in water vapor; they found that several of these factors play a role in the $H_2O$ variations, including the timing
of ENSO and SST variability, the phasing with the QBO, cold point temperatures, as well as the correct dynamical model state.
Many other analyses of the relation between entry level $H_2O$, tropopause temperatures, transport, and convection have been
carried out previously (e.g., Holton and Gettelman, 2001; Jensen and Pfister, 2004; Fueglistaler and Haynes, 2005; Rosenlof and
Reid, 2008; Read et al., 2008; Schoeberl et al., 2013). Our point here is that the WACCM $H_2O$ anomaly series underestimate the
observed variability. We note that this model underestimate exists if we calculate relative variability using a maximum minus
minimum range from yearly average anomalies rather than monthly averages; the resulting variability ratio patterns (not shown)
are similar, overall, to the contour plots of Fig. 19. We provide a global view of lower stratospheric variability differences
(models versus data) in the anomaly time series comparisons at 83 hPa for all latitude bins in Fig. S8. This also shows that the
observed interannual changes in $H_2O$ are clearly better followed by the SD-WACCM time series than by FR-WACCM,
including the drop in $H_2O$ after 2011 (see Urban et al., 2014). While lower stratospheric $H_2O$ variability is underestimated by
SD-WACCM by ~20%, the actual correlation between SD-WACCM and the observed anomalies is very good (as was shown in
Fig. 12).
For HCl, we show the (detrended) variability ratios in Fig. 21. The observed HCl variability is fairly well matched (within
~20%) by both models in the MLS time period, with an edge given to SD-WACCM. The observed variability is often
underestimated (by ~30%) by both FR-WACCM and SD-WACCM in the earlier period (1992-2003). We believe that the
HALOE sampling plays a role in this, i.e. even if we limit the model comparison (as we do here) to just the same months as
when HALOE observations occurred, incomplete sampling in latitude and time can lead to differences versus a fully sampled
model (see Toohey et al., 2013), and more so in the polar regions where the HCl variability is large. In the upper stratosphere, the
variability ratios are comparable to or somewhat smaller than those in the middle stratosphere, and there is a 20-30%
underestimate of the observed variability, which is based on HALOE HCl observations for the 1992-2003 period. For 2005-
2014, SD-WACCM actually matches the upper stratospheric MLS variability fairly well, although these variability values are
small. There have been difficulties in fully understanding (or modeling) observed upper stratospheric HCl variations before the
declining phase that started after about 2000 (Waugh et al., 2001); see also Sect. 5.
To complete this discussion of variability, we show the ratios of model to data variability for stratospheric $HNO_3$ (2005-2014)
and $N_2O$ (2005-2012) in Fig. 22. We already discussed the issues with missing model chemistry for upper stratospheric $HNO_3$
variability, as well as the low signal-to-noise issue for $HNO_3$ data at low latitudes in this region (see bottom two panels, showing
low variability ratios there). There is reasonably good agreement in the $HNO_3$ variability between SD-WACCM and MLS for the
lower to mid-stratosphere, while FR-WACCM generally overestimates the $HNO_3$ variability in this region. Fig. 22 shows $N_2O$
results extending down to 100 hPa. Here, the MLS N2O-640 data (from the 640 GHz radiometer) for 2005-2012 are used; these
retrievals were curtailed in the first half of 2013 as a result of degradation in the 640 GHz radiometer signal chain. Based on
results shown in SPARC (2017), there appears to be good agreement in the tropical interannual variability comparisons for $N_2O$
at 100 hPa between MLS and other satellite-derived results. The lower stratospheric $N_2O$ time series behave more smoothly at
low latitudes in the models than in the observations. The interannual variability in the MLS $N_2O$ measurements there is
somewhat smaller than the standard deviations in monthly mean $N_2O$ values (of order 20-30 ppbv). The MLS $N_2O$ measurement
noise itself for a monthly zonal mean (made up of about 5000-6000 profiles) should be less than 1 ppbv. Smoothing the model in



the vertical domain to better match the MLS vertical resolution would not lead to a better fit to the observed variability.
However, we should keep in mind that the MLS-derived $N_2O$ variability is a small percentage (< 3%) of the monthly zonal mean
$N_2O$ abundances. In summary, SD-WACCM shows some underestimate of the observed lower stratospheric tropical variability
for all the species considered here, except for $HNO_3$; FR-WACCM does so also for 3 species ($H_2O$, HCl, and $N_2O$). It may be
that some of the larger variability in the measurements arises from effects not tied just to MLS radiance noise issues, or from
variability caused by the proximity to the tropopause for measurements with finite vertical resolution; WACCM could also be
genuinely underestimating the actual atmospheric variability near the tropopause (for unknown reasons).
**5.2   Trends**
In this section, we discuss how the WACCM runs compare to stratospheric observations when it comes
to trends, from fairly short-term trends (from the Aura MLS time period) to longer-term trends based on comparisons with $O_3$,
$H_2O$, and HCl GOZCARDS data records (see Sect. 2 and Froidevaux et al., 2015). Trend analyses have their own complexities
in terms of analysis methods and uncertainty estimates. For example, $O_3$ trend assessments have had to deal with trend estimates
from different long-term data records, each with its own characteristics (Tummon et al., 2015; WMO, 2014; Harris et al., 2015;
Steinbrecht et al., 2017; Ball et al., 2017, 2018). This kind of analysis is especially difficult when investigating trends from time
series with high variability compared to the size of the change over time, which is an issue for the lower stratosphere in
particular. Global modeling efforts have also led to improved characterizations of the expected long-term impact of different
forcings, including the combined and separate impacts on ozone profiles of changes in halogen source gases and greenhouse
gases (WMO, 2014).
Here, we focus mostly on whether we obtain significant differences between trend results from WACCM and from
observations, given the application of the same analysis methods for the three sets of time series, for ozone and for other species.
We have applied multiple (or multivariate) linear regression (MLR) to the time series of deseasonalized anomalies from the data,
FR-WACCM, and SD-WACCM. In the Appendix (A3), we provide more details regarding the regression model, which includes
commonly used additive functional terms, namely a linear trend and a constant term, cosine and sine functions with annual and
semi-annual periodicities, as well as functions describing well known variations arising from the QBO and the El Nino southern
oscillation (ENSO); the same functions are applied to fit the model anomaly time series as well.  Examples of observational time
series from merged ozone observations for 1998 through 2014 are provided in Fig. A3, along with the fits to the series and the
linear components (trends). Given the use of fairly short-term time series here (e.g., Aura MLS data alone), we have not included
a solar cycle component in the fits, as it can be highly correlated with a linear trend, and more than one 11-yr cycle would be
useful to better enable a separation of this signal. We also discuss our methodology for trend error evaluations in the Appendix
(A3). We use a block bootstrap method, like the approach of Bourassa et al. (2014) for their trend analyses of ozone from the
OSIRIS retrievals. We use random resampling of the residuals in yearly blocks (with 20,000 samples) in this Monte Carlo
approach to estimating errors, and we display the trend error bars as $2\sigma$ values (which is very close to the 95% bounds on the
distribution of linear trend results). Such calculations often lead to significantly larger error bars than more standard methods,
which neglect the autocorrelation of residuals. Again, we use the same regression model fits to extract trends from both
WACCM and observed time series; error bar calculations are also applied the same way to all the time series.
For ozone, we give an overview in Fig. 23 of deseasonalized anomaly time series (expressed as a percent of long-term means)
for 3 latitude bins (northern mid-latitudes, tropics, and southern mid-latitudes) and 2 pressure levels (3.2 hPa for upper
stratosphere, 68 hPa for lower stratosphere). The series were first deseasonalized in 10° latitude bins and then averaged. The
GOZCARDS data record used here (version 2.20) is an update to the original (version 1.01) record (Froidevaux et al., 2015), as



mentioned in Sect. 2. Fig. 23 shows generally good agreement between the various time series, although if one looks carefully,
SD-WACCM is generally closer to the observational time series than FR-WACCM is, as one might expect from previous
considerations of goodness of fit and variability; also, percent variability is larger in the lower stratosphere than in the upper
stratosphere (note the y-axis range difference for these regions), thus rendering trend detection more difficult at lower altitudes.
We compare in Fig. 24 the ozone profile trend results from MLS data alone, from the period 2005 through 2014, to those from
FR-WACCM and SD-WACCM, for the 3 aforementioned latitude bins. We show the error bars as $2\sigma$ estimates, as we find this
to be an easy way to visualize if there are really significant differences between models and data, or indeed, between the models.
Figure 24 shows that there is a robust indication from the MLS data that the upper stratospheric ozone values have been on the
upswing in the past decade, at a rate of about 0.2 to 0.4%/yr, depending on latitude region, with $2\sigma$ uncertainties of ~0.2%/yr.
While the $2\sigma$ error bars (obtained using the bootstrap method mentioned earlier) are fairly large, there are several independent
latitude regions and pressure levels with similar results, and thus, this positive trend is a robust near-global upper stratospheric
result. These results are broadly consistent with $O_3$ trends obtained by Steinbrecht et al. (2017), who use MLS as part of the
longer-term merged data records, although they studied a longer time period (2000-2016). All things being equal, the errors in
these trends should diminish as more years of data are added to the MLS $O_3$ record, which, for the middle and upper stratosphere,
has been characterized as "very stable", namely within 0.1 to 0.2%/yr versus sonde and lidar network ozone data (Hubert et al.,
2016); it currently seems difficult to quantify "absolute stability" to much better than this, especially in the lower stratosphere. In
the lower stratosphere, trend results tend to be closer to zero, with larger variability and error bars (in %/yr), and unambiguous
detection of post-1997 ozone trends in this region remains elusive (WMO, 2014; Harris et al., 2015). The 2005-2014 trends in
Fig. 24 show good broad agreement between model and data, with a tendency for SD-WACCM to agree better than FR-
WACCM with MLS, albeit not significantly so, given the size of the error bars, especially in the lower stratosphere; the lower
stratospheric tropical results from FR-WACCM are negative, in contrast to both the observational result as well as SD-WACCM,
but with large overlapping error bars (given the variability and fairly short time period). In the tropical upper stratosphere, both
models exhibit a somewhat more positive trend than observed for this period, although, again, these trend differences are not
statistically significant. Nevertheless, as a function of latitude (in 10°-wide bins) and pressure, the 2005-2014 SD-WACCM $O_3$
trends do follow the MLS trend results quite well; this is shown in Fig. 25 for central latitudes from 55°S to 55°N.

26       For a consideration of longer time periods, we use the merged $O_3$ data record from GOZCARDS (version 2.20) to display

NH mid-latitude and tropical trends from the models and data for 3 different periods in Fig. 26. The top panels for 1985-1997
focus on the main "declining phase", while the middle panels (1998-2014) show results for the (expected) "early recovery" stage;
the 2005-2014 (bottom panels) results are essentially the same trends as those from Aura MLS (Fig. 24) for that period. The
largest differences between the two GOZCARDS data versions occur in the tropical upper stratosphere for the declining ozone
phase; Fig. S9 displays the tropical trend differences that we obtain for the same three periods as in Fig. 24. In agreement with
this are the trend differences provided by Ball et al. (2017), who showed results for the original (version 1.01) GOZCARDS data
and for SWOOSH. GOZCARDS version 2.20 data are now in better agreement with the merged SWOOSH $O_3$ product (as both
use SAGE II version 7 data); also, Steinbrecht et al. (2017) showed that the two merged records lead to similar (post-2000) trend
results. As mentioned in Sect. 2, the improvements in GOZCARDS version 2.20 ozone are a result of the incorporation of the
SAGE II v7 retrievals (Damadeo et al., 2013), and the use of the MERRA temperatures (used in v7) for the conversion from
density/altitude to the GOZCARDS mixing ratio/pressure grid. We note, however, that the lower stratospheric region exhibits
interannual variability that is several times larger than that in the upper stratosphere, as seen in Fig. A3 for tropical 1998-2014
data versus SD-WACCM anomaly time series. Even fairly subtle differences in time series over a few years can lead to a sign
change in the trends, although there is no statistical significance in the resulting trend differences (see Fig. A3 and Fig. 26,



middle right panel at 68 hPa, for the data and SD-WACCM trend results).

2       To put in perspective what a statistically significant trend difference looks like, we show in Fig. S10 the $O_3$ anomaly time

series for 1998-2014 at 1 hPa for 30°N-60°N, where the SD-WACCM and GOZCARDS trend results lie outside their respective
$2\sigma$ uncertainties (based on Fig. 26, middle left panel); in this case, the FR-WACCM results happen to be in better agreement
with the data.  One aspect that could impact model/data differences is that the models are not sampled, here, following the
sparser (occultation) viewing, neither in latitude nor in time (time within each month and local time also, since model values
represent 24-hr averages). Much denser spatial and temporal sampling is obtained during the MLS observation period, with very
regular sampling; while small systematic differences may affect comparisons between average model and MLS abundances, such
differences should be very consistent from year to year, thus minimizing the impact on derived trend differences. Also, some of
the differences in the upper stratosphere might arise because the averaging of sunset and sunrise occultation data is not as robust
for 1998-2004 as for pre-1998, when SAGE II was operating more continuously in both sunset and sunrise modes (and $O_3$ varies
more strongly with local time at 1 hPa than at lower altitudes). Also, HALOE had decreasing spatio-temporal coverage in the
later years, but these upper stratospheric trend differences would require further investigation. For the 30°S-60°S region (not
shown here), we observe similar results (within error bars) as in Fig. 26 for 30°N-60°N; this includes small negative trends in the
lower stratosphere for all 3 time periods, although with no statistical significance (i.e., consistent with a zero or slightly positive
trend). We also see the same sort of vertical shift in the SD-WACCM profile trends for 1985-1997, as compared to the observed
trend profile, which reaches its most negative value at a slightly higher altitude (lower pressure), but the reason for this shift is
unknown; theoretically, this could be tied to model/data differences in the ClO peak, and its evolution. Our ozone trend results
are largely consistent with other previous work (references mentioned above), which (for records including MLS) typically used
merged $O_3$ from GOZCARDS or from SWOOSH (Davis et al., 2016). We find statistically significant trends (meaning that,
assuming Gaussian statistics, a zero trend is not included in the $2\sigma$ error bar range) mostly in the upper stratosphere, both pre-
1997 and after 1998. While we observe some small $O_3$ decreases in the lower stratosphere post-1998, as obtained recently also
by the novel analyses of Ball et al. (2018), our study finds little statistical significance there, and a fair level of sensitivity to the
starting year or to the data sets used, with a swing to more positive (but marginally significant) results, if the starting year is 2005
and one just uses MLS data. Past work (e.g., Harris et al., 2015) has also shown sensitivity to the series starting and end points;
different regression analysis methods can also lead to some non-negligible differences, as shown for example by Nair et al. (2013)
and Kuttippurath et al. (2015). We also note that past analyses of lower stratospheric tropical $O_3$ data have shown positive
tendencies, based not just on satellite data as indicated here (with marginal significance) from MLS data alone, but also based on
earlier SCIAMACHY data (Gebhardt et al., 2014); in this reference, a positive trend was also obtained from averaged tropical
ozonesonde data. This will continue to require further study, towards a longer-term result.
For $H_2O$, there have been somewhat conflicting past results on stratospheric trends, depending on whether one investigates
sonde or satellite data (e.g., Oltmans et al., 2000; Scherer et al., 2008), and regarding mechanisms that could account for more
than a few tenths of a %/yr increase in $H_2O$, as $CH_4$ increases do not appear to be large enough for this. Beyond the potential
impact of $CH_4$ trends on upper stratospheric $H_2O$, changes in cold point temperatures or in the circulation need to be invoked in
order to account for significant decadal-scale trends in $H_2O$ (e.g., Randel et al., 2000; Rohs et al., 2006; Tian and Chipperfield,
2006; Hegglin et al., 2014). Based on our analyses for 2005-2014 (MLS data versus models), we find in Fig. 27 that this recent
decade shows a positive $H_2O$ trend both in the data and in the SD-WACCM result, which tracks the observations (versus latitude
as well as pressure) better than FR-WACCM does, as already seen in terms of quality of fits and variability. FR-WACCM
exhibits systematically smaller $H_2O$ trend values than both MLS and SD-WACCM at all pressures except near 100 hPa, although
the FR-WACCM and SD-WACCM trend error bars ($2\sigma$) overlap. This overlap for FR-WACCM and MLS is more marginal in




the lower mesosphere, where the impact on $H_2O$ from $CH_4$ decomposition should be at its maximum, and cold point temperature
variability issues are smaller than near the tropical tropopause. Also, the FR-WACCM trends have smaller error bars, given the
lower variability typically found in this model. With such lower variability, detection of a given trend would take less time than
with the observed variability; however, if the trend is larger (as shown by SD-WACCM or MLS), it also becomes easier to
detect. An analysis of $H_2O$ HALOE profiles and ground-based microwave profiles over Hawaii (Nedoluha et al., 2009) showed
that changes in upper stratospheric and mesospheric $H_2O$ are sensitive to the solar cycle (see also the mesospheric GOZCARDS
$H_2O$ time series in Froidevaux et al., 2015), but show only negligible overall trends between 1992 and 2008.
We should also note some drifts have been detected between coincident MLS and sonde $H_2O$ data, mostly since about 2010,
implying that MLS-derived trends are more positive than those from frost point profiles at several sites (Hurst et al., 2016). This
relative drift (of as much as 0.5-1.6%/yr for 2010-2015, depending on altitude and location) could therefore play a role in the
(small, in comparison) discrepancy between model (SD-WACCM) and MLS trends. The SD-WACCM results agree quite well
with MLS in Fig. 27, but they would become larger than MLS (adjusted) results if one were to subtract more than 0.1-0.2%/yr
from these MLS trends. Possible causes of observed drifts between MLS and sonde $H_2O$ data are still being investigated, with
only a small part of this discrepancy currently attributable to a known instrumental degradation issue for MLS $H_2O$, which
probably also impact other MLS data from the 190 GHz spectral region ($N_2O$ in particular). We note from Fig. 27 that $H_2O$ trend
values from both models and MLS data agree better at 100 hPa. Also, we see that $H_2O$ in the tropical region at 80 to 100 hPa
(near the stratospheric water vapor entry level) does not display much of a trend. Other recent studies of entry-level $H_2O$ using
large-scale satellite data and longer-term analyses (starting in the mid-1980s) have concluded that no significant long-term trend
is discernible (Hegglin et al., 2014; Dessler et al., 2014). The former study led to slightly negative lower stratospheric $H_2O$ trends
(although with no statistical significance).
For the 1992-2014 period and using GOZCARDS $H_2O$ data (not shown here), we also find negative, although not statistically
significant, trends in the lower stratosphere, with small (< 0.2%/yr) positive trends in the upper stratosphere and lower
mesosphere; there is close agreement between the FR-WACCM and SD-WACCM trends (typically within 0.1%/yr) for that
period. This GOZCARDS record, however, does not include SAGE II data back to the late 1980s, as was the case for the
analysis by Hegglin et al. (2014), who also obtained positive longer-term (1980-2010) trends in the upper stratosphere from
satellite-derived $H_2O$ anomaly records, merged using a global CCM as a transfer function. As found by others, especially when
dealing with relatively large decadal-type variability (including the QBO), the choice of start and end points, as well as the length
of period studied, can significantly influence the trend values, whether it be for $H_2O$ or for $O_3$. Long-term lower stratospheric
$H_2O$ trend detection is rendered difficult by such variability, including significant short-term water vapor changes (Randel et al.,
2006; Hurst et al., 2011; Fueglistaler, 2012; Urban et al., 2014), as noted here also (for the relatively short time series of Fig. S8).
For HCl, the changes in stratospheric abundances have been non-linear, with a rapid rise prior to 1998, and a slower rate of
decrease after 2004, as expected from the time-shifted abundances of total chlorine at the surface (Froidevaux et al., 2015).
Focusing first on 2005-2014, we show the corresponding model and data HCl trend results for the lower stratosphere in Fig. 28.
The agreement between the SD-WACCM and MLS trends is quite good, especially for the 30°S-60°S bin, although the error
bars are fairly large. However, the (negative) HCl trend results from both models practically always lie below the observed
trends for the three latitude bins shown in Fig. 28. The upper portion of this model/data bias follows what we observe as well in
the upper stratosphere (not shown), where MLS-derived HCl trends are clearly too flat (shallow) compared to expectations (from
model and surface-derived chlorine trends), whereas the upper stratospheric (negative) trends from the original MLS HCl
product were more negative (see Froidevaux et al., 2006; Livesey at al., 2018). As a reminder, the MLS team recommendation is
for data users not to include upper stratospheric MLS HCl data (post-2006) in any trend studies. For the lower stratosphere,



where the HCl line is broader, there is less concern about the inability to track the HCl trend; also, the near-zero drifts (i.e., drifts
< 0.1%/yr) obtained between two separate MLS ozone band retrievals (not shown here), one from the same radiometer as HCl,
and one from the main (very stable) standard MLS ozone product (see Hubert et al., 2016), provide some confidence regarding
the stability of lower stratospheric HCl trends. At low latitudes, MLS HCl shows a positive trend (largest and statistically
significant at 68 hPa, per Fig. 28). The vertical gradient structure in the observed HCl trends is duplicated to some extent by the
SD-WACCM results, although the model trends are always less than those derived from MLS. Latitude/pressure trend
variability, including positive tendencies, could be related to circulation changes, as implied by analyses of short-term increases
in lower stratospheric HCl seen in both ground-based and GOZCARDS data (Mahieu et al., 2014). Given the rapid rise in
chlorine prior to 1998 and the non-linear changes near the peak period, we show in Fig. 29 some of the lower stratospheric time
series (for 3 latitude bins and 3 pressures) from GOZCARDS merged HCl (Froidevaux et al., 2015) and the WACCM runs for
1992-2014. There is fairly good agreement in the non-linear behavior observed in both data and model lower stratospheric series.
The scatter in HCl data decreases after 2005, and the earlier time series suffer from more inhomogeneous sampling, which may
at least in part explain the larger scatter and model/data differences (there is no attempt here to sample the models within each
month like the data, and this would be difficult for a merged data set). There are also regions and periods of slow increases in
HCl in both data and models (Fig. 29), as well as hemispheric differences in the short-term tendencies, as discussed before by
Mahieu et al. (2014) and Froidevaux et al. (2015). The HCl time series are tracked fairly well by SD-WACCM, which generally
matches the data better than FR-WACCM; this is consistent with the understanding that dynamically-driven variations are better
captured by the incorporation in SD-WACCM of realistic meteorological fields (MERRA). Stolarski et al. (2018) have recently
investigated the removal of dynamical variability from MLS lower stratospheric HCl time series by using MLS $N_2O$
measurements as a fitting parameter in the regression analysis; this led to retrieved HCl trends that generally match expectations
based on the rates of decrease in surface total chlorine. The search for detailed explanations of such short-term increases and
variability in lower stratospheric HCl (and other changes in composition) continues to be an interesting area of investigation.
In the upper stratosphere, it has been difficult to explain the details of the observed HCl variations between 1998 and 2002,
including the dip between these years (Waugh et al., 2001). We show in Fig. S11 the near-global (60°S-60°N) HCl time series
from GOZCARDS at 1 hPa. This helps to underscore that there is a systematic model underestimate of HCl in the uppermost
stratosphere; the model/data difference is much smaller if one moves to a pressure close to 5 hPa. While the data uncertainties (of
~0.2 ppbv, based on Froidevaux et al., 2015) encompass the model values, the model total should actually be increased by the
chlorine contribution from very short-lived substances (VSLSs) to the stratosphere; although this contribution is believed to be
less than 0.1 ppbv (Carpenter et al., 2014), recent evidence suggests that there could be a somewhat larger stratospheric chlorine
contribution from VSLSs (Oram et al., 2017). Nevertheless, the historical maximum for total tropospheric chlorine abundance
was about 3.65 ppbv (WMO, 2014), and this should be the maximum total chlorine expected in the uppermost stratosphere.
Upper stratospheric HCl should see a somewhat broader peak than at the surface, with a smaller and time-delayed maximum,
depending on transport-related effects (and age of air spectrum, e.g., see WMO, 2010). While WACCM includes the proper
abundance and evolution of chlorine source gases at the surface, maximum WACCM HCl in the upper stratosphere (and lower
mesosphere) is just under 3.4 ppbv. It is also interesting that the gap between the models (both versions) and the data worsens
from 1992 to 2000, with the HCl peak occurring later in the data (with a broader peak than in the models). After about 2002, the
decrease in near-global HCl roughly follows the model decrease; additional years of HCl data from ACE-FTS should help refine
this comparison.
For $N_2O$ and $HNO_3$, lower stratospheric model trends are compared to the corresponding MLS data trends in Fig. 30. We note
that the MLS standard product right after launch was N2O-640 (retrieved from the 640 GHz radiometer band data), but it was





discontinued after mid-2013, as mentioned earlier, as a result of a rapid hardware degradation issue affecting that band ($N_2O$
only). The current MLS standard product, N2O-190, is retrieved from the 190 GHz band. Figure S12 provides evidence of
negative drifts in lower stratospheric N2O-190, apparently accelerating in the last few years, since the SD-WACCM and actual
$N_2O$ values would be expected to continue to rise slowly after the end date on this plot, notably in the tropical lower stratosphere,
where $N_2O$ should follow tropospheric trends. Indeed, tropospheric $N_2O$ has been increasing at a fairly steady rate of ~0.26%/yr
(WMO, 2014), consistent with the underlying model $N_2O$ and MLS N2O-640 lower stratospheric increases at low latitudes (see
Fig. S12 and especially the tropical trends obtained in Fig. 30 at 100 hPa). The FR-WACCM $N_2O$ trends show slightly poorer
agreement than SD-WACCM versus N2O-640, although this is not statistically significant. The tropical lower stratospheric MLS
N2O-190 trend results (not shown here) are negative (albeit with error bars that encompass small positive trends), but show some
differences versus expectations and the N2O-640 results. As for HCl, interhemispheric differences in lower stratospheric $N_2O$
trends are interesting in terms of their implications for effects relating to transport (age of air) and changes in the circulation. At
lower pressure values, the $N_2O$ trends do not mirror the tropospheric $N_2O$ trends (in %/yr), and other factors play a role (age of
air and changes in circulation, as well as $N_2O$ photodissociation). The asymmetric trend pattern between hemispheres, even if it
is not a long-term trend, may well point primarily to short-term effects tied to asymmetries in the age of air, and therefore, in the
circulation. The asymmetry in age of air results obtained by analyses of (2002-2012) MIPAS $SF_6$ data (Haenel et al., 2015) could
also be related to asymmetries in the $N_2O$ tendencies. They found relatively older air in the northern extra-tropics and younger air
in the southern extra-tropics. This could also imply opposite trends for $N_2O$ in the southern and northern lower stratosphere.
However, Bönisch et al. (2011) have pointed out that different tracers, like $O_3$ and $N_2O$, can be impacted in different ways by
both vertical and quasi-horizontal transport effects, depending on their relative gradients in both vertical and meridional domains.
Moreover, their work indicates that detailed attribution of tracer variations to structural changes in the Brewer-Dobson
circulation is a complex matter, and short-term and longer-term changes may well have different characteristics. Our work here
mainly identifies some similarities between some of the trend patterns versus pressure and latitude from SD-WACCM, in
particular, and the observed trends, for certain time periods. For the $HNO_3$ trends (Fig. 30), we also see good agreement between
models and data for 2005-2014; latitudinal tendencies and interhemispheric differences therein are similar for model and data.
The spatial gradients of these species are different in the lower stratosphere (HCl and $HNO_3$ increase with height, in contrast to
$N_2O$), and we see that the decreasing HCl trends for 2005-2014 at 30°S-60°S (Fig. 28), in particular, are qualitatively similar to
those from $HNO_3$ in this region. For lower stratospheric $HNO_3$, there is an underlying trend part caused by the slow increases in
$N_2O$, as we can observe in longer-term (1980 to present) model time series (not shown here). $N_2O$ and $H_2O$ (source gases for
$HNO_3$) are significantly affected by the QBO and there is a strong related variability in lower stratospheric $HNO_3$. Furthermore,
substantial increases in stratospheric aerosols after large volcanic eruptions have influenced lower stratospheric $HNO_3$ via
heterogeneous hydrolysis of $N_2O_5$ (Arnold et al., 1990; Rinsland et al., 1994), and this will impact $HNO_3$ trends that include
volcanically-perturbed periods. We saw that seasonal enhancements in NOx coming down from the mesosphere can also affect
$HNO_3$ at high latitudes. Some observed short-term trend patterns in HCl, $HNO_3$, and $N_2O$ are better captured by the SD-
WACCM model overall than by FR-WACCM, as we show in Fig. 31 for the 2005-2010 period, relevant to the results from
Mahieu et al. (2014), who emphasized short-term HCl increases during this time. We note the correlation in these short-term
trend results for $HNO_3$ and HCl, but an anti-correlation for $N_2O$ versus HCl (and $HNO_3$).
Finally, to re-emphasize how difficult it can be to detect small underlying trends in the lower stratosphere, in particular, Fig.
32 shows deseasonalized model anomalies from SD-WACCM for the 25-year period between 1990 and the end of 2014, for
$HNO_3$, as well as $O_3$, $H_2O$, and HCl; on this scale, $N_2O$ variability in this region is much smaller (not shown). As noted
previously, and from past work on this topic, fairly sharp drops in water vapor in this region occurred shortly after 2000 and



2011, but we also note the significant decadal-type variability in this region, besides the expected links to QBO- and ENSO-type
variations. There are also non-negligible radiative implications surrounding such variations for $H_2O$ (Solomon et al., 2010;
Gilford et al., 2016; Wang et al., 2016); however, a slow underlying long-term evolution would take time to detect, given the
variability. Moreover, the percent variability is even larger for other species, which correlate well with $H_2O$ in this region;
indeed, the correlation coefficients between these various time series are all between 0.64 and 0.86, although this multi-species
agreement is poorer at many other pressure levels or latitudes. In Fig. S13, we show that the observed variability in this same
region for $O_3$ and $H_2O$ (for 1992-2014) is fairly well matched by the models, although this is somewhat less true for $H_2O$ than for
$O_3$. Besides the importance of circulation effects on tracers in this region, tropopause temperatures will also affect water vapor;
this adds some complexity in terms of exactly modeling its variations in the tropical lower stratosphere.

## 6    Summary and discussion

The climatological averages from FR-WACCM and SD-WACCM for $O_3$, $H_2O$, HCl, $N_2O$, and $HNO_3$ generally compare
favorably (within the $2\sigma$ estimated systematic errors) with the Aura MLS data averages for 2005-2014. Model ozone values are
usually within ±5-10% of the average observations, except in the UTLS. In the lowest portion of the stratosphere, SD-WACCM
generally exceeds the observed ozone means by about 30-50%, with FR-WACCM showing a smaller overestimate; both models
also tend to overestimate the amplitude of the annual cycle in this region. There is also a model low bias (by ~10-30%) from 215
to 261 hPa at low latitudes, which could largely be caused by a known MLS high ozone bias in this region. For $H_2O$, there is a
model low bias (by 5-15%) versus MLS data in the upper stratosphere and most of the mesosphere, although some of this arises
from a small high bias in MLS $H_2O$ versus other satellite data sets (see Hegglin et al., 2013). Also, the models significantly
underestimate the average $HNO_3$ abundances in the upper stratosphere, notably at high latitudes; this largely appears to stem
from missing model ion chemistry, as it relates to particle precipitation effects in the mesosphere, followed by downward
wintertime polar transport of enhanced NOx, and subsequent seasonal increases in $HNO_3$. There is also some model
overestimation by SD-WACCM of MLS $HNO_3$ (by about 40%) at high latitudes for pressures larger than 100 hPa, although
there is a need for further validation of the $HNO_3$ data in this region. In the lower stratosphere at high southern latitudes, the
variations in polar winter/spring composition observed by MLS are generally well matched by SD-WACCM, the main exception
being for the early winter rate of decrease in HCl, which is too slow in the model. Grooß et al. (2018) have provided further
discussion of this discrepancy, which should have little impact on winter/spring polar ozone depletion; indeed, we find good or
better agreement between the seasonal high latitude observations and SD-WACCM for ozone and other species.
Regarding the fitted variability tied to the AO and SAO, there are a few discrepancies between model-derived amplitude
patterns and the corresponding MLS climatology features, but FR-WACCM and SD-WACCM appear to properly capture the
primary processes governing these modes of variability. SD-WACCM generally matches the data sets slightly better than FR-
WACCM does. The $O_3$ AO stratospheric amplitudes are within ~25% of the MLS AO amplitudes. For $H_2O$, both WACCM
versions exhibit AO and SAO patterns that are generally consistent with the observations, and with recently published satellite-
derived results (Lossow et al., 2017a); we also note the WACCM underestimation of $H_2O$ AO and SAO amplitudes in the lower
stratosphere, although this is the region with the smallest amplitudes (< 0.1 ppmv).
We have provided diagnostics for the fits between the WACCM runs and the MLS deseasonalized anomaly time series. These
consist of the correlation coefficient ($R^2$ diagnostic) as well as a diagnostic of RMS differences (model versus data), divided by
the RMS variability in the data; a combined diagnostic (the ratio of the above two diagnostics) is also used to help differentiate
between the two model runs, which are often not too far apart. Not too surprisingly, SD-WACCM, which is driven by realistic
dynamics versus time, generally matches the observed zonal monthly mean anomalies significantly better than does FR-
WACCM. This holds for all five species that we considered, with larger values of $R^2$ and smaller values of the RMS difference





diagnostic. In the tropical lower stratosphere, where there is some nudging to equatorial winds for FR-WACCM (and even more
so for SD-WACCM, see Sect. 2), the FR-WACCM fits to the data are generally improved. However, details of the observed
interplay between SAO, AO, and QBO variations in tropical upper stratospheric ozone are better matched by SD-WACCM
variations than by FR-WACCM. Also, FR-WACCM shows poorer agreement with observed seasonal polar winter/spring lower
stratospheric variations than does SD-WACCM. Finally, in the mesosphere, the water vapor anomalies are better matched by
SD-WACCM than by FR-WACCM.

7       Variability comparisons represent a more fair and useful metric in terms of the characterization of model quality, in particular

for a free-running model. To this end, we have compared the RMS interannual variability from the anomalies in both WACCM
models and observations; we have used both MLS data (2005-2014) and data from longer-term series based on GOZCARDS
data records for $O_3$, $H_2O$, and HCl. One of the main features from these comparisons is that the $H_2O$ variability from the lower
stratosphere to the upper mesosphere is underestimated by both model runs used here; this underestimate can reach a factor of
two, although more typically, it is of order 30%. This implies that a larger number of years would be needed to detect an actual
trend in $H_2O$ than if one uses a model-based prediction (from FR-WACCM); this number of years would be increased by a factor
of 1.2 to 1.6, if one uses the two variability factors mentioned above. Apart from the WACCM underestimate of observed $H_2O$
variability, the observed lower stratospheric variations, including significant drops in the $H_2O$ abundance, are better tracked by
SD-WACCM than by FR-WACCM. The ozone variability is better represented by the WACCM models, with model/data
variability ratios typically within a factor of 0.8 to 1.2. Observed HCl variability is underestimated somewhat by FR-WACCM
for the 1992-2003 period, but not for the later (2005-2014) period; the sparser HALOE sampling, compared to MLS, could
explain some of the underestimate for the early period, especially in the polar regions. For $N_2O$, there is also a model
underestimate (from both FR-WACCM and SD-WACCM) of the lower stratospheric low latitude variability observed by MLS,
although this variability is a small percentage of the mean values.

22       Regarding trends, the model comparisons versus the longer-term ozone data record from GOZCARDS (version 2.20 being

used here) show generally good qualitative agreement in the time series in different latitude bins for both upper and lower
stratospheric change. It is clear from such time series that the larger percent variability in the lower stratosphere will continue to
render trend detection in this region more difficult than in the upper stratosphere. Based on the Aura MLS $O_3$ data record itself,
which has been deemed very stable (Hubert et al., 2016), we observe robust evidence, considering the $2\sigma$ error bars of ~0.2%/yr
(estimated using a block bootstrap method), that there is a positive upper stratospheric ozone trend for 2005-2014, at a rate of
~0.2-0.4%/yr. This is true for all three broad mid-latitude and tropical regions considered here (30°N-60°N, 30°S-60°S, and
20°S-20°N), although the evidence is more marginal in the SH mid-latitudes. The WACCM trends estimated using the same
regression model as used for the MLS data (anomaly) series show generally good agreement with the data trends, although the
error bars are fairly large (for both data and models). Furthermore, the observed trend (relative) dependence on latitude and
pressure is well matched by the SD-WACCM trend results. We have not considered the high latitudes in detail in this work, in
part because of the significant dynamical variability in that region. In this regard, Stone et al. (2018) recently analyzed model
results at high latitudes in the upper stratosphere, and showed that the large variability in that region, including the effects of
solar proton events, is likely to mask detection of recovery (for now), although autumn and winter should exhibit the strongest
recovery signals. In the lower stratosphere, where larger variability exists, the trends we deduce from the data sets and models
agree within fairly large error bars, although there is generally no statistical significance. While there is a tendency for the
GOZCARDS merged $O_3$ record to show small decreasing trends for the 1998-2014, the trend results reverse to near-zero or
slightly positive tendencies (albeit with no robust statistical significance) if one considers the MLS data alone for 2005-2014.
SD-WACCM trend results seem to track these positive tendencies, although with no robust statistical significance, based on our



analyses. The recent work by Ball et al. (2018) indicates a net $O_3$ decrease in the lower stratosphere from about 1998 to the
recent few years; this does not contradict the possibility of a turn-around towards a more positive rate of change in this region
during the more recent 10-12 years. The positive tendency noted here may get more robust with the analyses of more years of
high quality global ozone profiles, and possibly more aligned with longer-term model expectations. Future detailed analyses of
these issues with different regression models and other methods are certainly indicated.
For $H_2O$, the most statistically significant trend result is an upper stratospheric increase for the post-2005 time period, peaking
at slightly more than 0.5%/yr in the lower mesosphere, with MLS and SD-WACCM results agreeing fairly well, and FR-
WACCM showing significantly smaller increases. The fact that the last decade has seen more of an upper stratospheric and
mesospheric $H_2O$ increase than the previous decade appears to correlate with the very shallow maximum that occurred in the last
cycle (number 24) of the solar flux, which seems tied to the shallower dip, and broader overall maximum, in upper mesospheric
$H_2O$ (see the $H_2O$ and solar flux time series in Fig. 16 of Froidevaux et al., 2015). However, the non-linear influence of recent
changes in methane, which include a plateau from 1999-2006, with a return to rising abundances after that (Schaefer et al., 2016;
Nisbet et al., 2016), would also need to be considered for the upper stratosphere. There is also a caveat regarding MLS-derived
$H_2O$ trends, given the existence of non-negligible drifts between sonde and MLS $H_2O$ data (Hurst et al., 2016), at least since
about 2010. Such drifts can only be partially explained by currently known instrumental degradation issues affecting the MLS
retrievals of $H_2O$, with some impact on other data from the 190 GHz radiometer ($N_2O$, in particular). Thus, the MLS-derived
$H_2O$ trend results obtained here are likely to be upper limits; this can probably explain why the model trends (at least from SD-
WACCM) currently lie on the low side of the observed trends. An upcoming update to the MLS retrievals will lead to a
reduction (but not an elimination) of the aforementioned drifts between MLS and sonde $H_2O$ data. There is a continued need for
cross-comparison of the various (diminishing number of) satellite $H_2O$ data sets, as well as $H_2O$ profiles from satellites and
sondes, hopefully leading to a better understanding and mitigation of instrumental issues and drifts between different water vapor
observations.
Our HCl trend analyses reveal broad agreement between the lower stratospheric MLS data (2005-2014) and the models, but
with some systematic differences. As mentioned in the past, upper stratospheric MLS HCl data are not deemed to be reliable
enough for trend studies, since the cessation (in 2006) of the primary target band retrievals for MLS HCl. While lower
stratospheric HCl decreases are generally indicated for 2005-2014, there are some hemispheric differences, and a significant
increase is suggested in the tropical data at 68 hPa; however, there is only a slight positive trend there from the SD-WACCM
result (with no statistical significance). While the lower stratospheric vertical gradients of trend results from MLS HCl are
duplicated to some extent by SD-WACCM, the model trends are always on the low side. There is no clear preference for SD-
WACCM or FR-WACCM trend results, in comparison to the observed lower stratospheric HCl trends. There have also been past
indications of short-term increases in lower stratospheric HCl (Mahieu et al., 2014); the study of such short-term tendencies for
implications regarding circulation changes are worth pursuing further (but outside the scope of this work). There is a need for
more comparisons of the various HCl measurements, satellite-based and ground-based, as well as models, in order to better
understand circulation influences on stratospheric composition, as well as potential measurement-related issues (e.g., from
sampling differences or potential drifts).
For $N_2O$, the asymmetry in MLS-derived trends (for 2005-2012) between hemispheres, with negative trends (of up to about -
1%/yr) in the NH mid-latitudes and positive trends (of up to 3%/yr) in the SH mid-latitudes, is in agreement with the asymmetry
that exists in SD-WACCM results. The small observed positive trends of ~0.2%/yr in the 100 to 30 hPa tropical region are also
consistent with model results (SD-WACCM in particular), which in turn are very close to the known rate of increase in
tropospheric $N_2O$ (at a rate of about +0.26%/yr, see WMO, 2014). In the case of $HNO_3$, the MLS-derived lower stratospheric





trend differences (for 2005-2014) between hemispheres are opposite in sign to those from $N_2O$ (whose spatial gradients are
largely of a sign opposite to those from $HNO_3$) and in reasonable agreement with both WACCM results, despite large error bars
compared to the size of the trends. More detailed analyses would be needed to try to relate such trend asymmetries to changes in
age of air, or circulation.

5       Overall, the models and observations show good agreement in the trends, with somewhat better results for SD-WACCM,

which displays good correlations in the trend behavior versus latitude and pressure. However, the error bars are non-negligible,
and the choice of start and end dates can have a significant impact on trends or tendencies. Given the existence of significant
short-term and decadal-type variations for several lower stratospheric species, one should be cautious not to assign, or
extrapolate, a tendency based on even a decade of data, to an underlying longer-term trend.
The diagnostics provided in this WACCM model evaluation can help distinguish even fairly subtle differences between
models and observations, as well as between models. The generally improved fits to observations from a specified dynamics
model versus a free-running model are to be expected, but also need to be documented (part of the aim of this work). We are also
reminded that observations have their own systematic issues, and close collaboration between modeling groups and instrument
teams can help untangle issues that might be more driven by, or at least influenced by, species-dependent instrumental effects.
Especially when comparing longer-term model time series to observations, even small systematic effects such as measurement
drifts, or data merging issues, can become important for trend diagnostics. Finally, independent CCMs are not created in the
same exact way, and nudging approaches for free-running and specified dynamics models can vary; for example, some models
have an internally-generated QBO, but most do not. While this study focused on (CESM1) WACCM runs, further studies of the
differences between high quality observations and various international models of atmospheric composition would be useful, to
put this work in perspective. This could also be expanded to include some species not considered in this work, and/or with more
of a focus on the upper troposphere.





**Appendix A**
**A1   Examples of model grades**
A grading method that has been applied in some previous comparisons (e.g., see Douglass et al., 1999; Waugh and Eyring,
2008) between atmospheric model values ($M_n$) and observed values ($O_n$) utilizes Eq. (A1) below to arrive at grades between 0
and 1 (and if a grade is < 0, it can be set to 0):
$$grade = 1 - \sum_1^N \frac{|M_n - O_n|}{E_f \times \sigma_n}$$  (A1)
with index $n$ (in a given time series) varying between 1 and $N$ (the total number of monthly values being compared for a given
latitude/pressure bin), and $\sigma_n$ representing the error in the observations. While the error factor $E_f$ should probably be set to 2 or
3, this gives grades that are too small (close to zero or negative) if one applies such a formula to the MLS $O_3$ or $H_2O$ time series,
specifically to data sets with pretty well defined total measurement errors (provided as $2\sigma$ error estimates, per Sect. 4, meaning
an error factor of 2). The grades shown here in Figs. A1 and A2 correspond to error factors ($E_f$) of 2 and 4, respectively. Figure
A2 leads to $O_3$ and $H_2O$ grades that are more useful than Fig. A1; it also shows similarities with the diagnostic based on the RMS
of the differences between model and data, as shown in Sect. 4 (see the description of this diagnostic in Appendix A2).

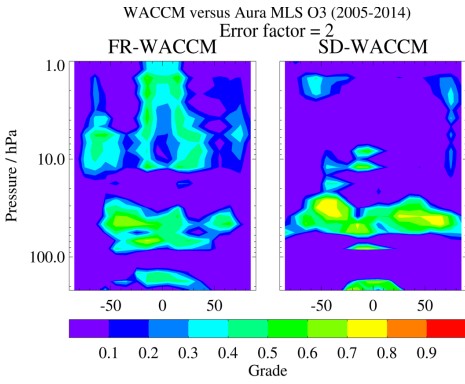

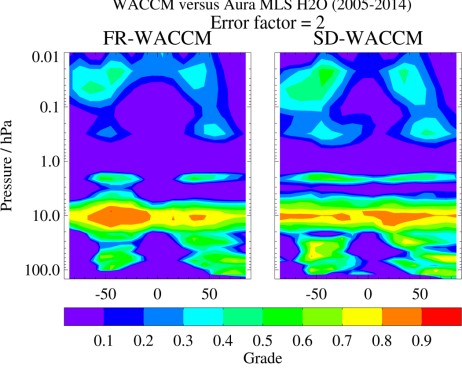

**Figure A1.** Examples of grades for model evaluations of $O_3$ (top two panels) and $H_2O$ (bottom two panels), using a grading methodology that
has been used in the past (see Eq. A1), applied to both FR-WACCM (left panels) and SD-WACCM (right panels) time series versus Aura MLS
time series from 2005 through 2014. These grades are for an error factor of 2 (in Eq. A1).





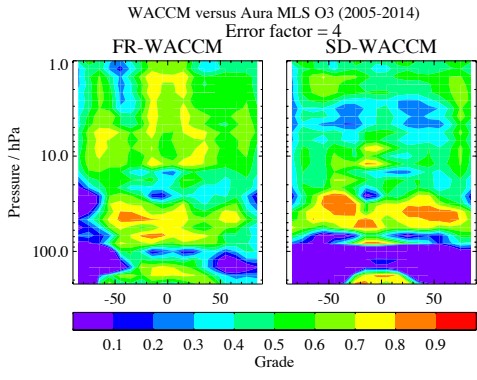

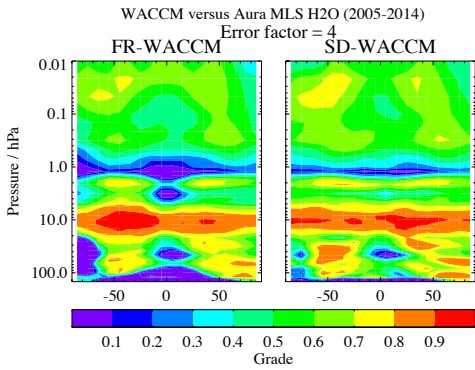

**Figure A2.** As in Fig. A1, for model evaluation grades of $O_3$ and $H_2O$ (WACCM versus Aura MLS data), but for a value of 4 (rather than 2)
for the error factor (see Eq. A1).
**A2  RMS difference diagnostic**
Given a model time series $M_i(t)$ and an observational time series (both series here representing deseasonalized anomalies)
$O_i(t)$, the difference values between the two anomaly series are simply given by
$\Delta_i(t) = M_i(t) - O_i(t)$ (A2)
and the root mean square (RMS) of these anomaly differences (RMSdif) is expressed as
$RMSdif = \sqrt{\frac{1}{N} \sum_i \Delta_i^2}$ (A3)
The RMS value (variability) of the observational series (of anomalies) in this case is
$RMS_O = \sqrt{\frac{1}{N} \sum_i O_i^2}$ (A4)
One of the diagnostics that we use in Sect. 4 to compare how well different models match up with the observed time series is
given by
$D_{RMSdif} = \frac{RMSdif}{RMS_O}$ (A5)
**A3  Regression model**
*Functional form:* The MLR model and fitting methodology used here is similar to the methods used by many others in the
past, with a linear model description that uses annual, semi-annual, QBO, and ENSO terms. Thus, the model function to be fitted
for coefficients $a$, $b$, $c_n$, $d_n$, $f_1$, $f_2$, and $f_3$ has the familiar form:



$y(t) = a + b\,(t - t_m) + \sum_n ( c_n \sin 2\pi t/P_n + d_n \cos 2\pi t/P_n )$

3       $+ f_1\,QBO_1(t) + f_2\,QBO_2(t) + f_3\,ENSO(t)$                                                 (A6)

with the (monthly series) time variable expressed by $t$, and $t_m$ chosen as the series mid-point; the linear trend term is coefficient $b$
above. The *sine* and *cosine* functions provide for periodic variations with periods $P_n$. For our work, we use the two primary
shorter-term periodic oscillations, annual (12-month period) and semi-annual (6-month period), in Equation (A6). The QBO is
also a major source of variability in stratospheric composition time series. As a QBO proxy, we include the variability in
monthly mean tropical wind series, as is commonly done; we use the linear combination of (roughly orthogonal) equatorial wind
series at 50 hPa and 30 hPa as the $QBO_1$ and $QBO_2$ functions above, to account for phase shifts in the series at different
locations. Monthly mean zonal equatorial wind data are made available by the Freie Universität Berlin (see http://www.geo.fu-
berlin.de/en/met/ag/strat/produkte/qbo/ for data access information and references). We have also tested fits with the zonal mean
wind vertical shear (gradient) rather than the wind itself as a proxy, but this did not make significant changes in the trends (or
improvements in the residuals). The ENSO proxy follows the monthly mean multivariate ENSO Index (MEI), which combines
data from six main geophysical variables over the tropical Pacific (see Wolter and Timlin, 1993, 1998;
https://www.esrl.noaa.gov/psd/enso/mei/index.html). Also, since the solar cycle 11-yr term can be highly correlated with the
linear term, especially for shorter-term records like the time series from MLS data alone, we have not added a proxy solar term in
this study. For further discussions of alternate fitting methods (e.g., methods using effective equivalent chlorine time series as a
proxy), the reader is referred to the abundant literature on trend assessments (see WMO, 2014 and references therein). Our main
goal here is to retrieve trends and trend errors from the data and the models in the same way. An example of deseasonalized
ozone time series in the tropics at two pressure levels is provided in Fig. A3, which shows MLS data and SD-WACCM time
series, along with the fits and the linear trends.
*Trend errors:* For the evaluation of error bars in the linear trends, we have used the method of bootstrap resampling (Efron
and Tibshirani, 1986). As others have done for ozone trend analyses (Randel and Thompson, 2011, Bourassa et al., 2014), we
have applied this using block bootstrapping (using yearly blocks of data), thereby preserving some of the dependency in the time
series. Basically, one samples and (randomly) replaces blocks of yearly data for a large number of resampling cases (on the
residuals), and then calculates the standard deviation of the large number of trend results (linear fits) to arrive at the trend
uncertainties; note that we use $2\sigma$ values as error bars in our trend comparison plots (which is very close to the 95% bounds). We
have used 20,000 samples in our bootstrap analyses; changing this number (e.g., by several thousand) does not alter the results
significantly, as long as one chooses a large enough total number of resampled cases. An alternative method is to attempt to
correct trend uncertainties for the autocorrelation of the residuals after the regression fit (Tiao et al., 1990; Weatherhead et al.,
1998; Santer et al., 2000). The existence of non-random residuals effectively implies that the number of independent data points
is less than the number making up the original time series. The end result is that trend uncertainties are (and should be) larger
than if one neglects these effects. We find that trend errors from this bootstrap method for our time series examples are often
larger than the more simplistic/standard calculations by factors ranging from about 1.2 to 2 or more. We have checked our trend
error calculations with the OSIRIS team, based on a sample time series, as they have used the same block bootstrap approach
(Bourassa et al., 2014).





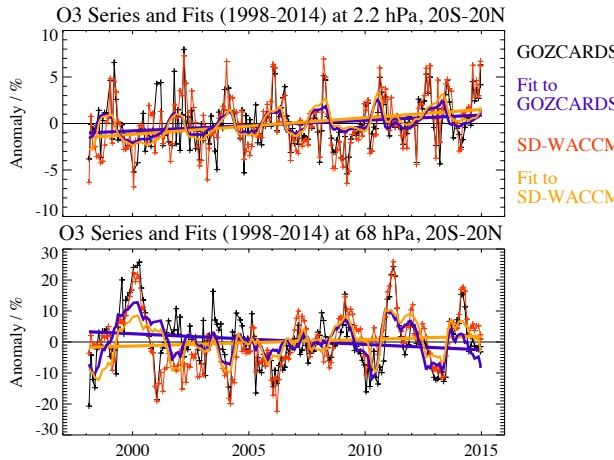

**Figure A3.** Deseasonalized monthly mean anomaly time series for O$_3$ (%) at 2.2 hPa (top panel) and 68 hPa (bottom panel), for 1998 through 2014, for averages over 20°S-20°N. Note that the y-axis range for 68 hPa (bottom) is 3 times larger than for 2.2 hPa (top). The data set (black) is from the GOZCARDS ozone record, with the SD-WACCM (simulated) series (red) also shown. Fits to the observational series are in purple, and fits to the model series are in orange; the fitted time series functions (curves) and the fitted linear components (straight lines) are shown.

*Data availability.* Aura MLS data used in this work are monthly zonal means derived from Level 2 MLS data, which are accessible from the Goddard Earth Sciences Data and Information Services Center (GES DISC), funded by NASA's Science Mission Directorate; a link for MLS Level 2 data access can be found at https://mls.jpl.nasa.gov/data/. GOZCARDS data sets can be obtained (by entering GOZCARDS in the search) at http://disc.gsfc.nasa.gov. More recent years (and version updates) will be made available at this site, or can be obtained by request to the first author. The WACCM model output used here is provided in some of the references and is available from the NCAR Earth System Grid at https://www.earthsystemgrid.org/search.html?Project=CCMI1. For WACCM, we thank NASA Goddard Space Flight Center for the MERRA data (accessed freely online at http://disc.sci.gsfc.nasa.gov/).

*Author contributions.* L. Froidevaux produced the majority of this manuscript, including the Figures. D. Kinnison provided the model runs used in these comparisons to observational data from Aura MLS and GOZCARDS; he also described the models used here and provided substantial guidance and comments for this manuscript. H.-J. Wang and J. Anderson, along with L. Froidevaux, were key participants in the development and creation of the GOZCARDS data sets, including the recently updated version (2.20) for merged ozone used herein. R. Fuller was another key participant in the GOZCARDS data production, and he also provided programming support for these model intercomparisons.

*Competing interests.* The authors declare that they have no conflicts of interest.

*Special issue statement.* This manuscript is intended as part of the Chemistry-Climate Modelling Initiative (CCMI) inter-journal special issue (ACP/AMT/ESSD/GMD SI).

*Acknowledgements.* Work at the Jet Propulsion Laboratory, California Institute of Technology, was performed under contract with the National Aeronautics and Space Administration (NASA). GOZCARDS data were initially produced under the NASA Making Earth System Data Records for Use in Research Environments (MEaSUREs) program, with data continuity/updates now provided under MLS funding. We acknowledge insights and comments regarding aspects of this work from Rolando Garcia, Nathaniel Livesey, Jessica Neu, and Michelle Santee. We also appreciate discussions and comparison work regarding trends and error bars with Doug Degenstein and Chris Roth, as well as Emmanuel Mahieu (and via some of his European contacts). The National Center for Atmospheric Research (NCAR) is sponsored by the U.S. National Science Foundation (NSF). WACCM is a component of NCAR's Community Earth System Model (CESM), which is supported by the NSF and the Office of Science of




the U.S. Department of Energy. Computing resources for WACCM were provided by NCAR's Climate Simulation Laboratory, sponsored by NSF and other agencies; we also acknowledge the computational and storage resources of NCAR's Computational and Information Systems Laboratory (CISL).

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



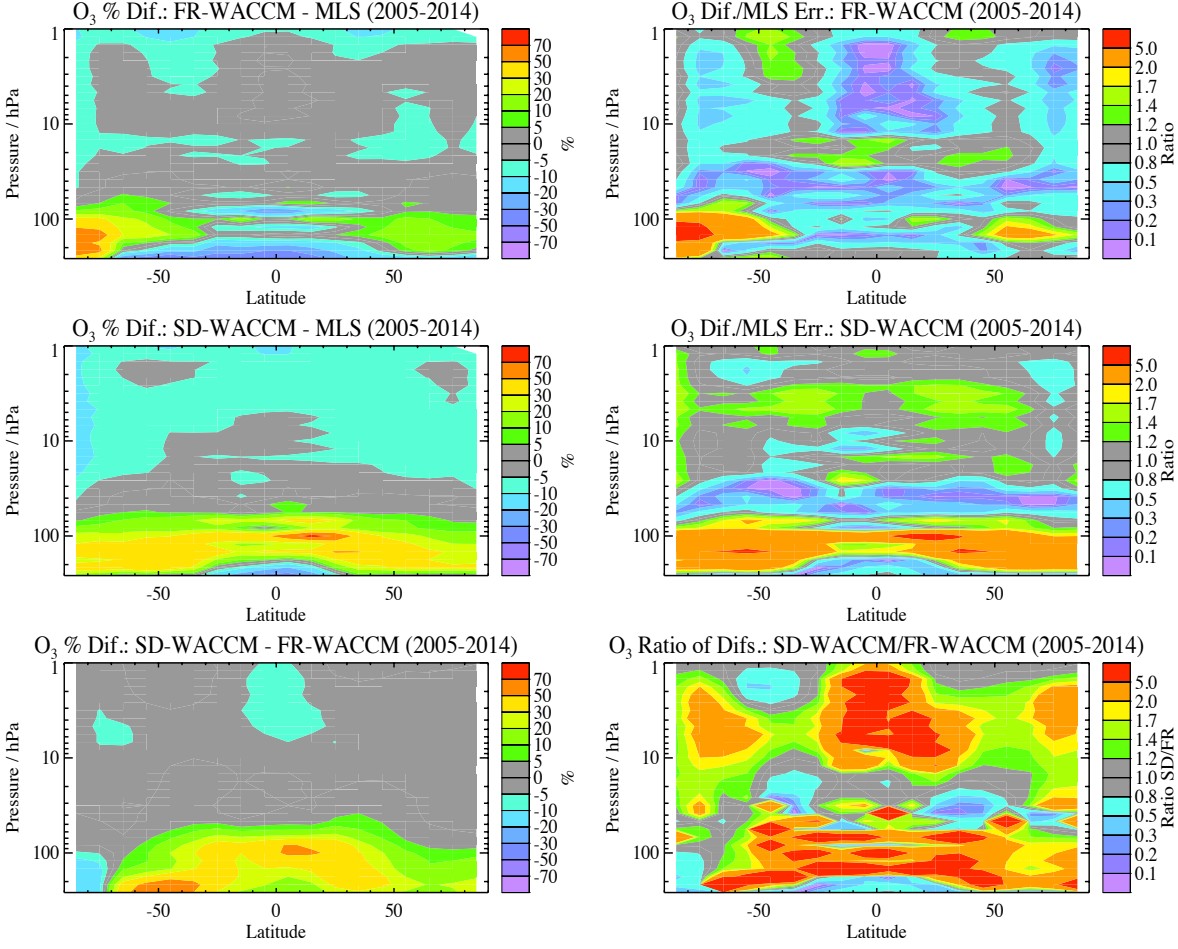

**Figure 1.** The left panels show percent differences ((model-data) divided by data) for binned climatological average O₃ from 2005 through 2014 (see Fig. S1 for these averages), for (top panel) the free-running model (average of 3 realizations) FR-WACCM, (middle panel) the specified dynamics model version SD-WACCM, and (bottom panel) SD-WACCM minus FR-WACCM, also as a percent difference. The two top right panels give ratios of the absolute value of average model (FR-WACCM in top panel, SD-WACCM in middle panel) minus average MLS O₃ to the MLS systematic O₃ errors, based on the climatological fields and estimated MLS errors (2σ estimates, see text and Fig. S2). The bottom right panel gives the ratios of average absolute differences between SD-WACCM and MLS to those differences for FR-WACCM and MLS; e.g., this shows that upper stratospheric tropical SD-WACCM mean O₃ is larger than O₃ from FR-WACCM, which is why SD-WACCM matches MLS better there (see top 2 right panels).





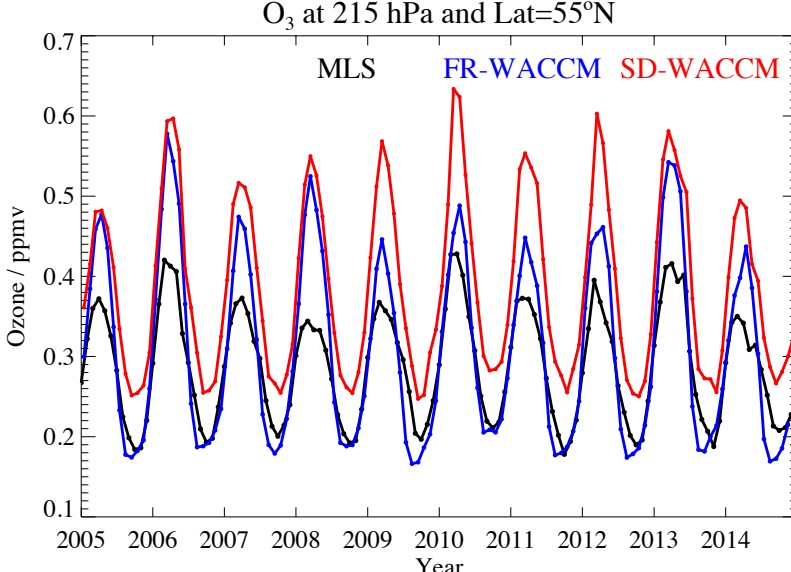

5 **Figure 2.** Monthly mean ozone time series at 215 hPa and the 55°N latitude bin (for averages over 50°N-60°N) from MLS, FR-
6 WACCM, and SD-WACCM (see legend for color coding).





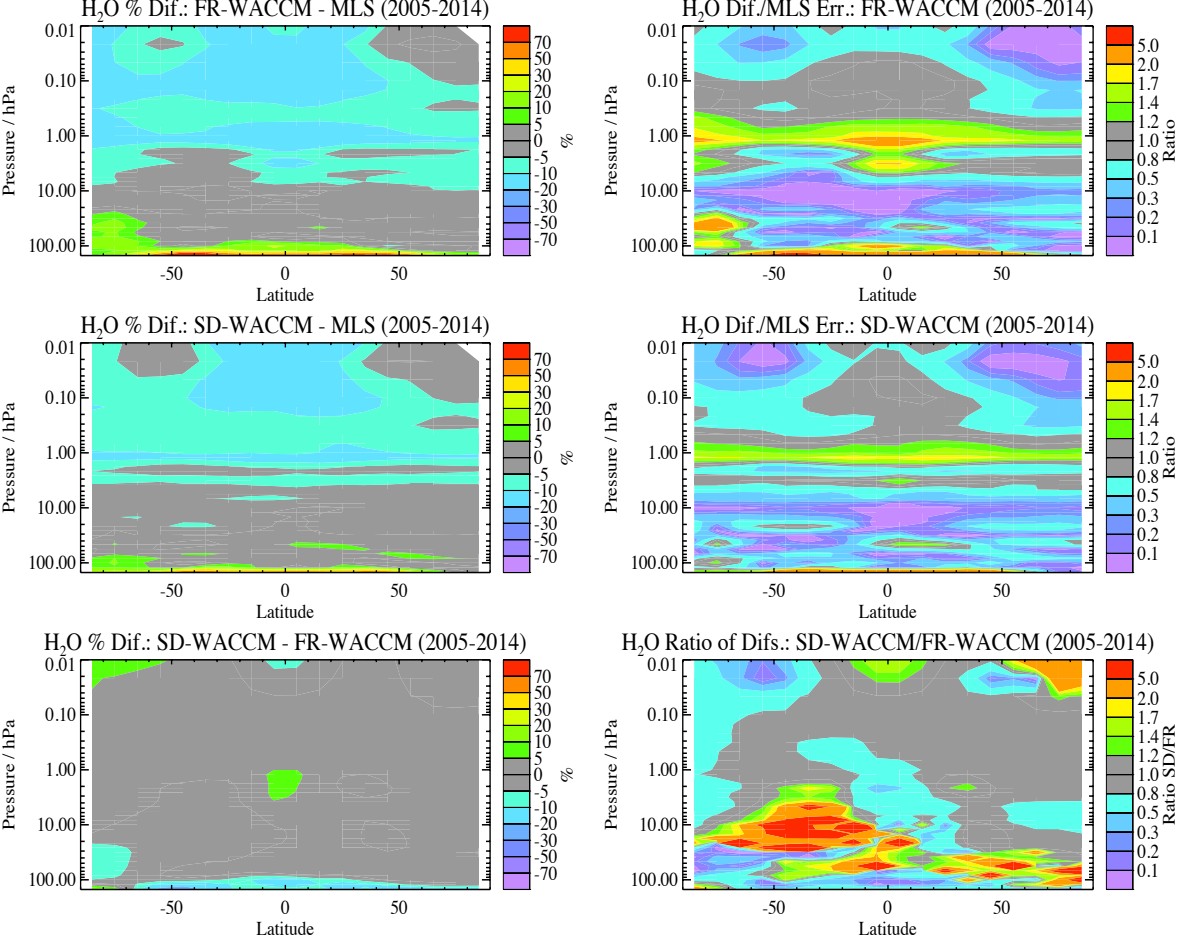

5  **Figure 3.** Same as Fig.1, but for stratospheric and mesospheric water vapor.



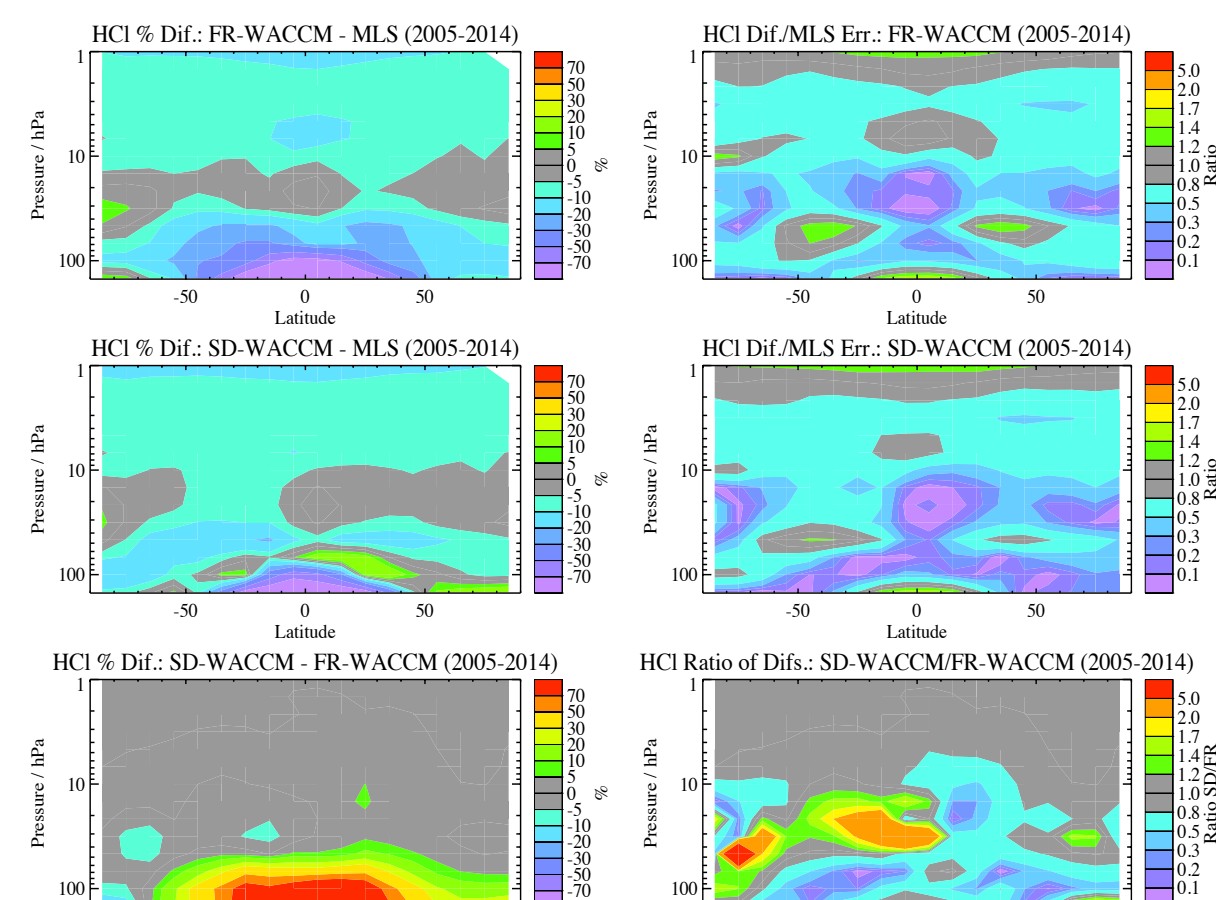

**Figure 4.** Same as Fig. 1, but for stratospheric HCl.





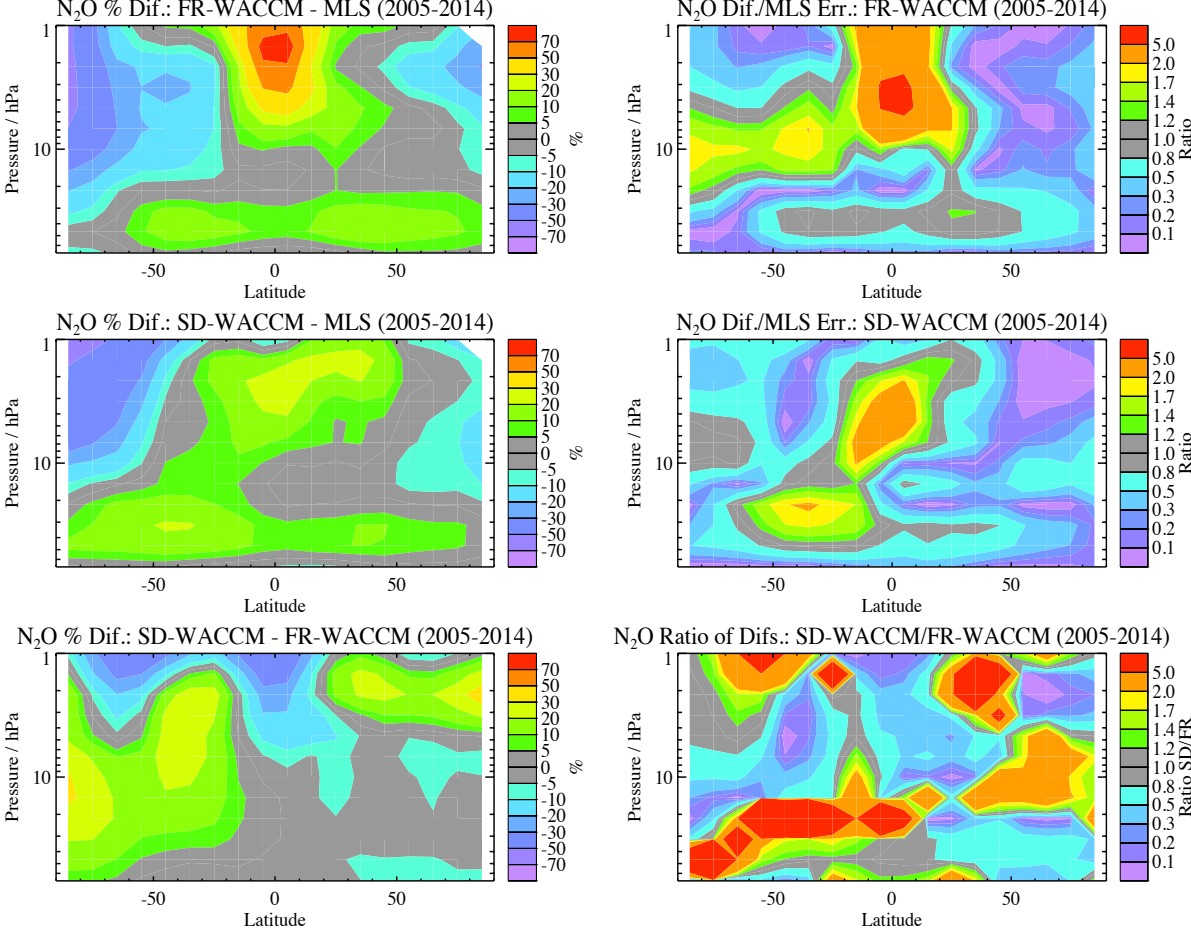

**Figure 5.** Same as Fig. 1, but for stratospheric N$_2$O.



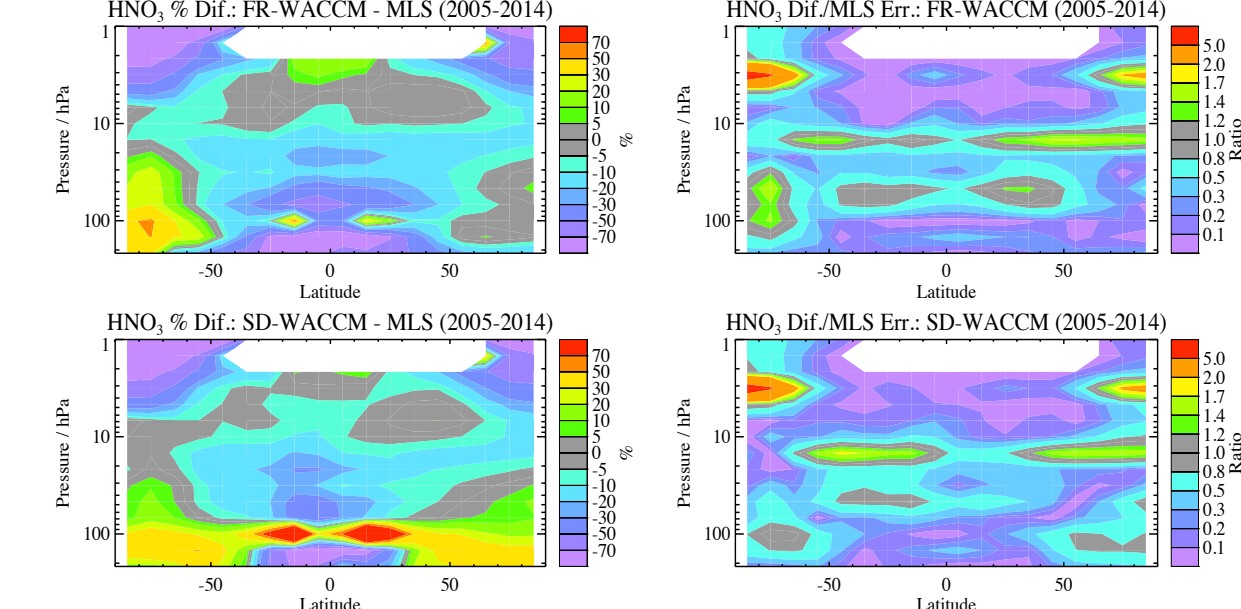

5  **Figure 6.** Same as Fig. 1, but for stratospheric HNO₃.



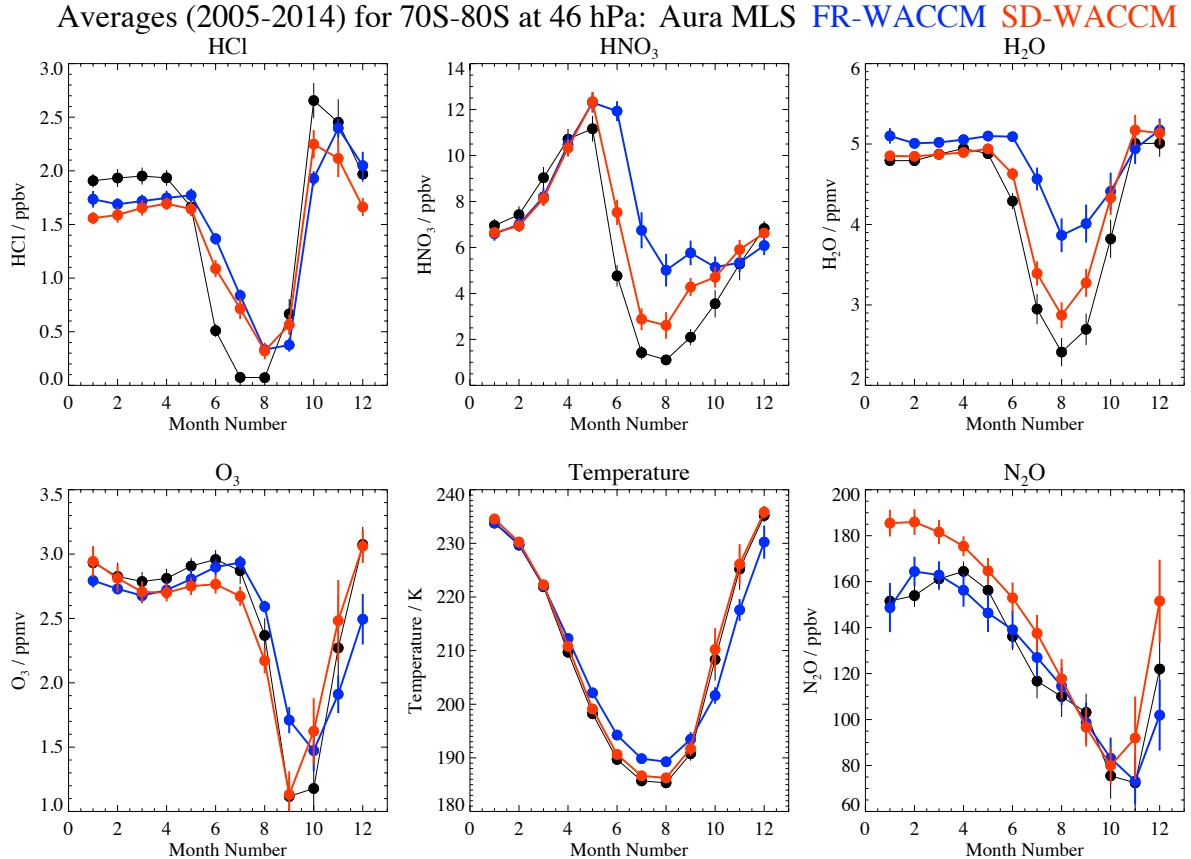

**Figure 7.** Each of the panels shows average seasonal changes from 2005-2014 for the 70°S-80°S region at 46 hPa. Data values (black) are from Aura MLS and model comparisons (FR-WACCM in blue, SD-WACCM in red) are provided for HCl (top left), HNO$_3$ (top center), H$_2$O (top right), O$_3$ (bottom left), temperature (bottom center), and N$_2$O (bottom right). For each month, the error bars represent twice the standard errors in the means, based on the set of 10 monthly averages (from 2005 through 2014).



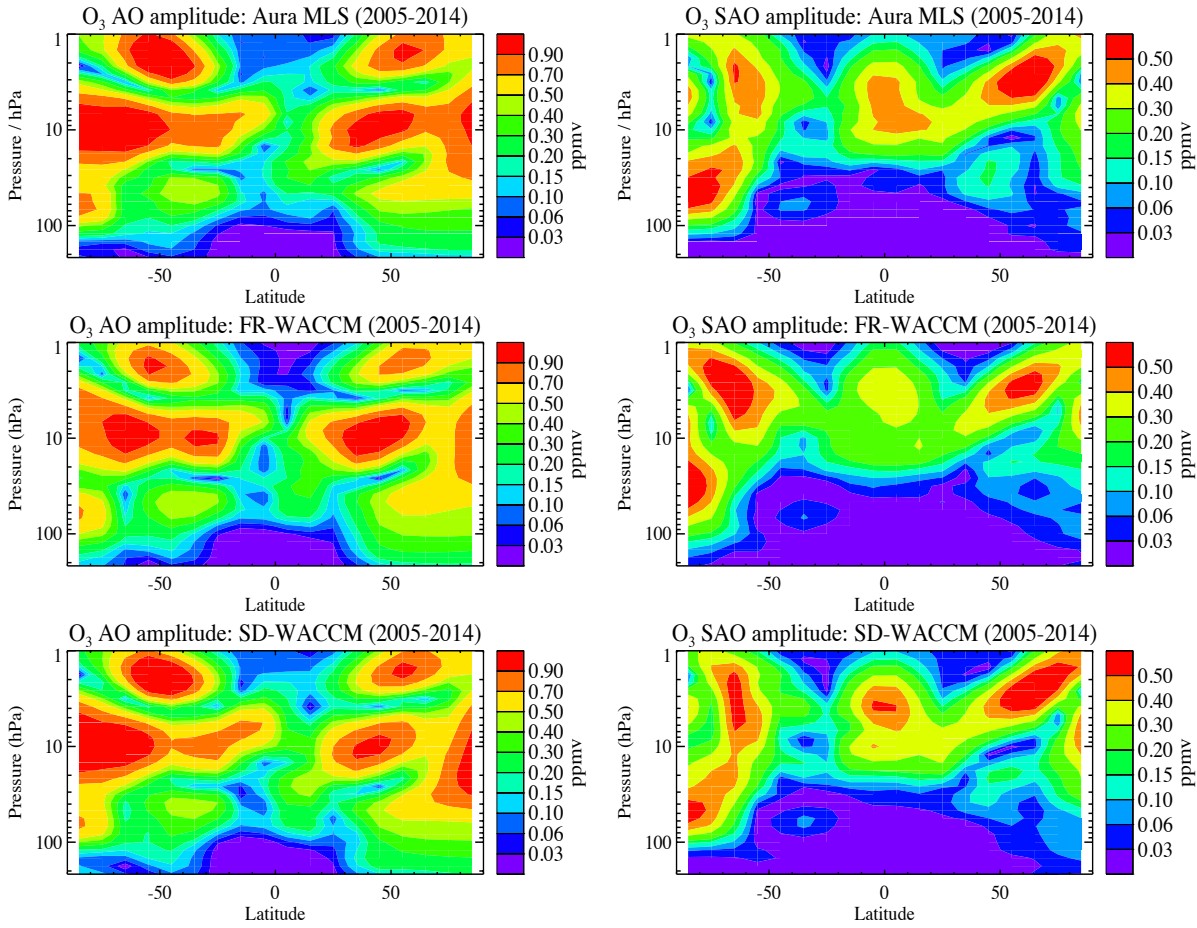

**Figure 8.** Amplitude of the stratospheric ozone annual cycle (left panels) and semi-annual cycle (right panels) for Aura MLS (top), FR-WACCM (middle), and SD-WACCM (bottom), based on fits to time series from 2005 through 2014.




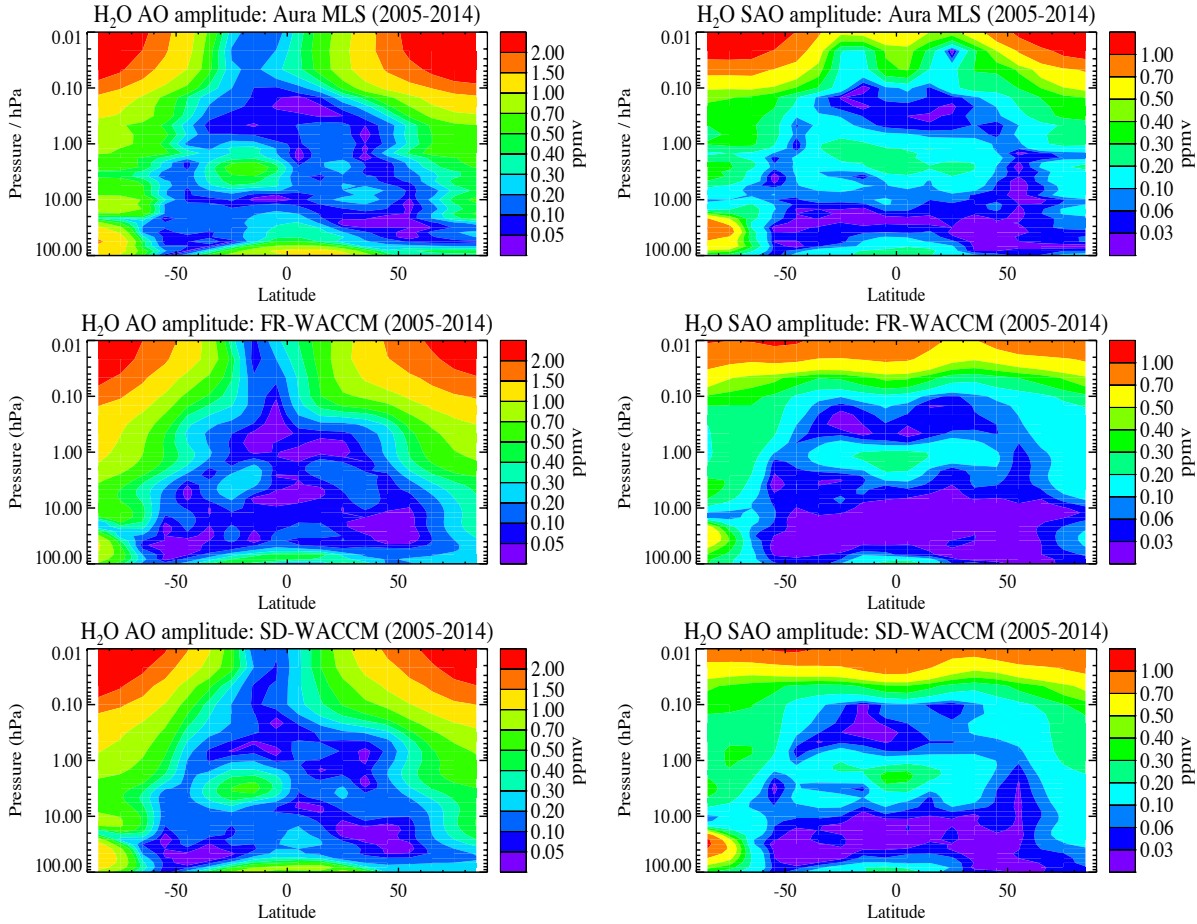

**Figure 9.** Same as Fig. 8, but for $H_2O$ annual and semi-annual cycles in the stratosphere and mesosphere.




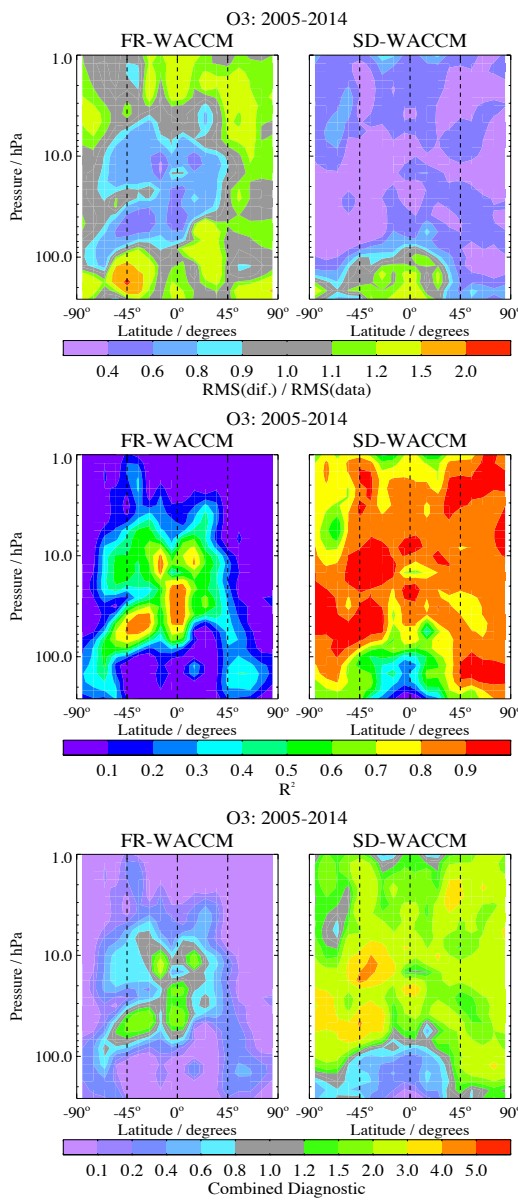

**Figure 10.** Latitude/pressure contours of diagnostics that show how well the deseasonalized anomalies of model ozone time series (FR-WACCM at left, SD-WACCM, at right) compare to MLS O$_3$ anomaly series for 2005-2014. Top panels show the RMS difference diagnostic (see text) and middle panels show R$^2$ values; small RMS difference values represent a closer fit, while large R$^2$ values represent highly correlated results. The bottom panels provide a combined diagnostic, namely the ratio of R$^2$ to the RMS difference diagnostic from the top panels; larger values here represent a better result for comparisons to the observed time series.





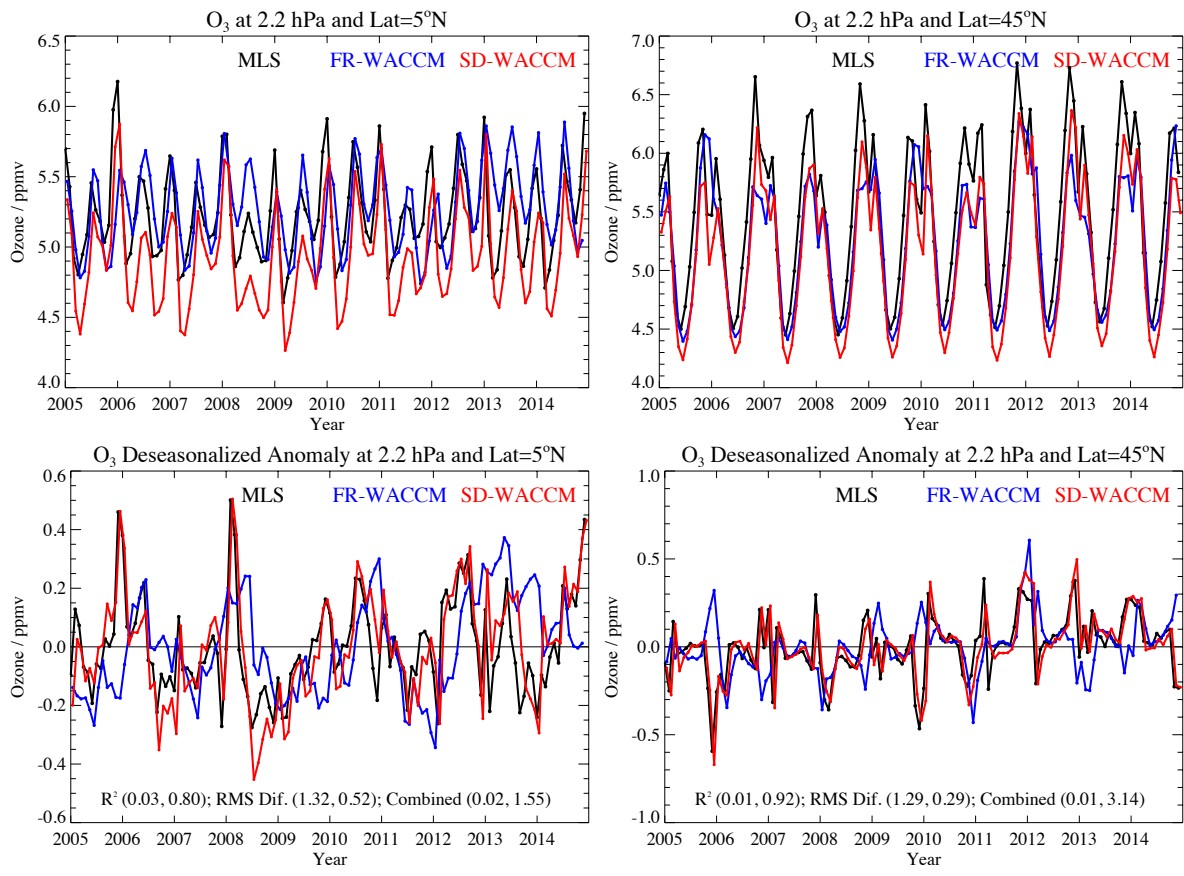

**Figure 11.** Time series of monthly zonal mean $O_3$ mixing ratios at 2.2 hPa (top panels) and deseasonalized anomalies
(bottom panels), with the 0-10°N and 40°N-50°N latitude bins on the left and right, respectively. The two model time
series (FR-WACCM in blue and SD-WACCM in red) are compared to the MLS series (in black) for 2005-2014. Diagnostic
values (see text for a description) are shown in parentheses in the bottom two panels, with the 1st number referring to
FR-WACCM and the 2nd number to SD-WACCM.



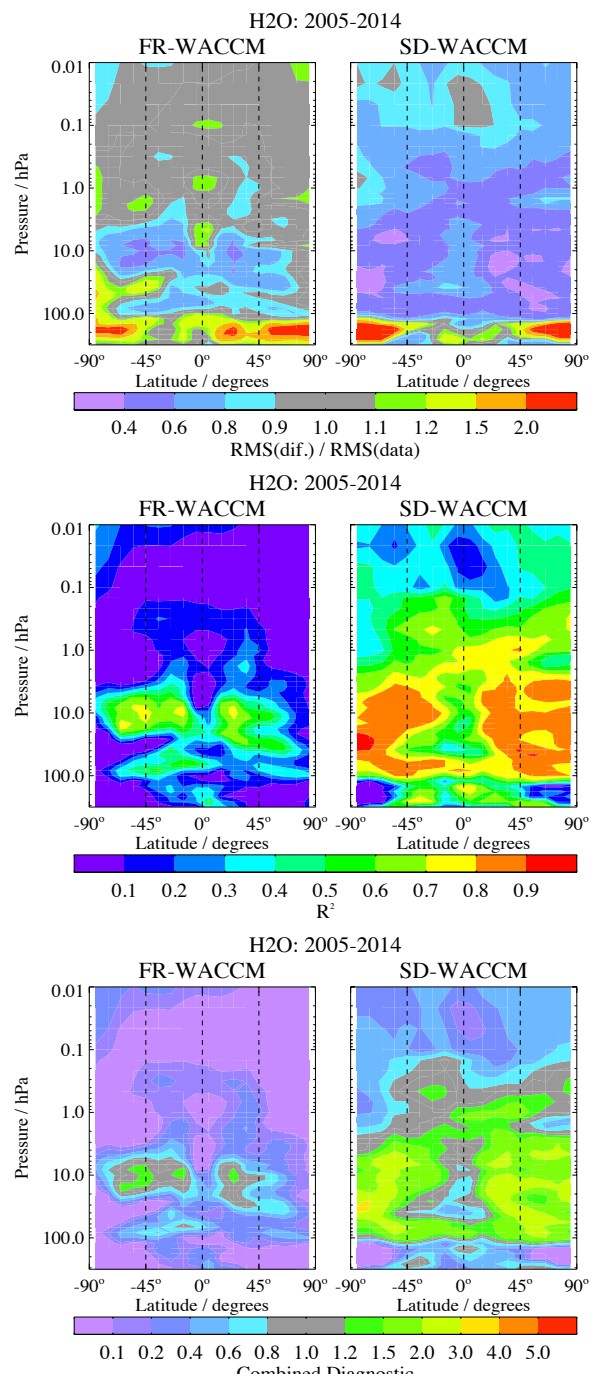

4
5    **Figure 12.** Same as the Fig. 10 diagnostics, but for $H_2O$ up to 0.01 hPa.
6





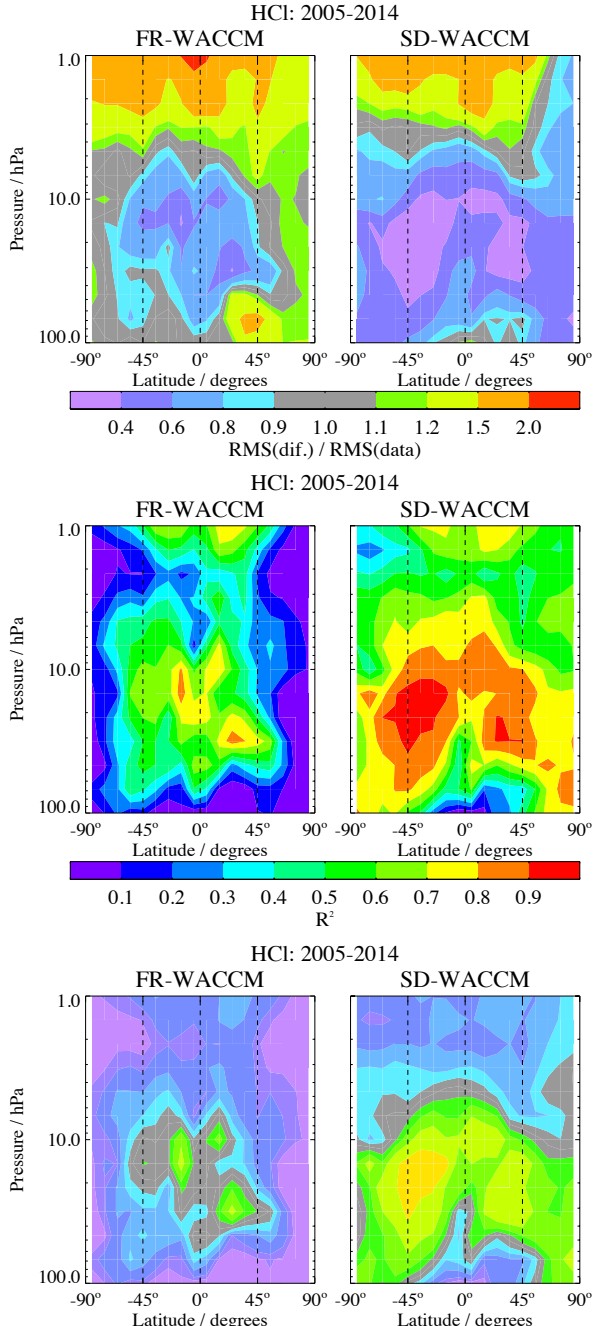

4
5    **Figure 13.** Same as the Fig. 10 diagnostics, but for stratospheric HCl.

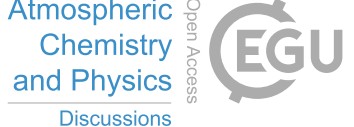



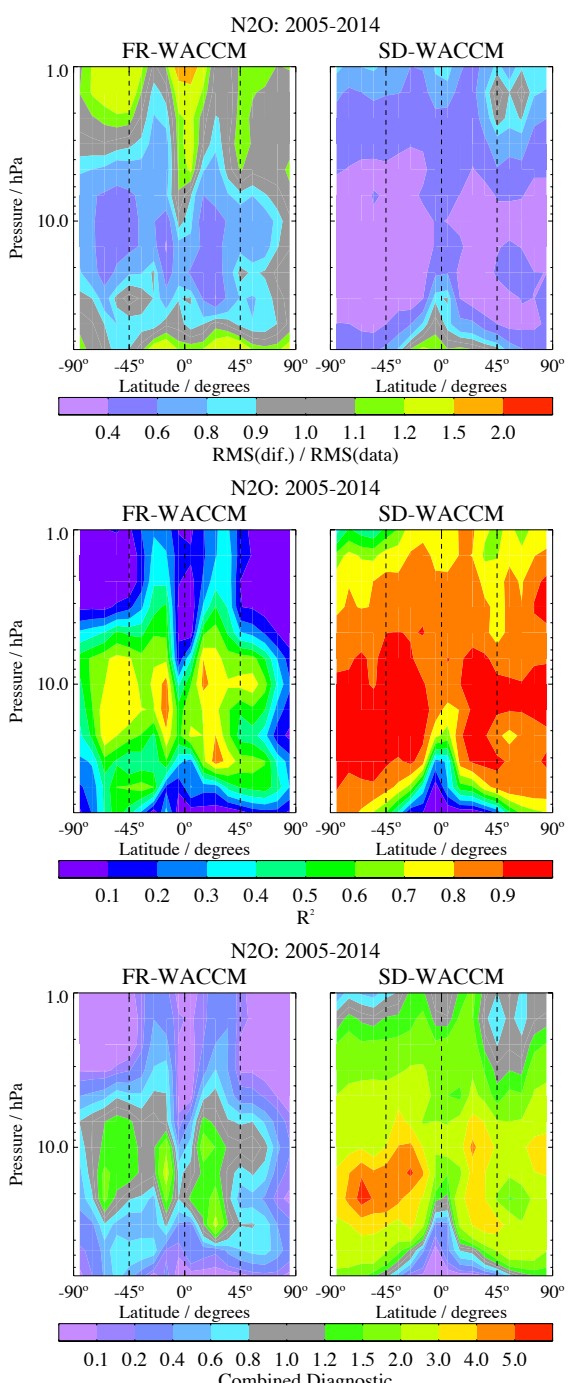

4
5   **Figure 14.** Same as the Fig. 10 diagnostics, but for $N_2O$ (from 68 to 1 hPa).





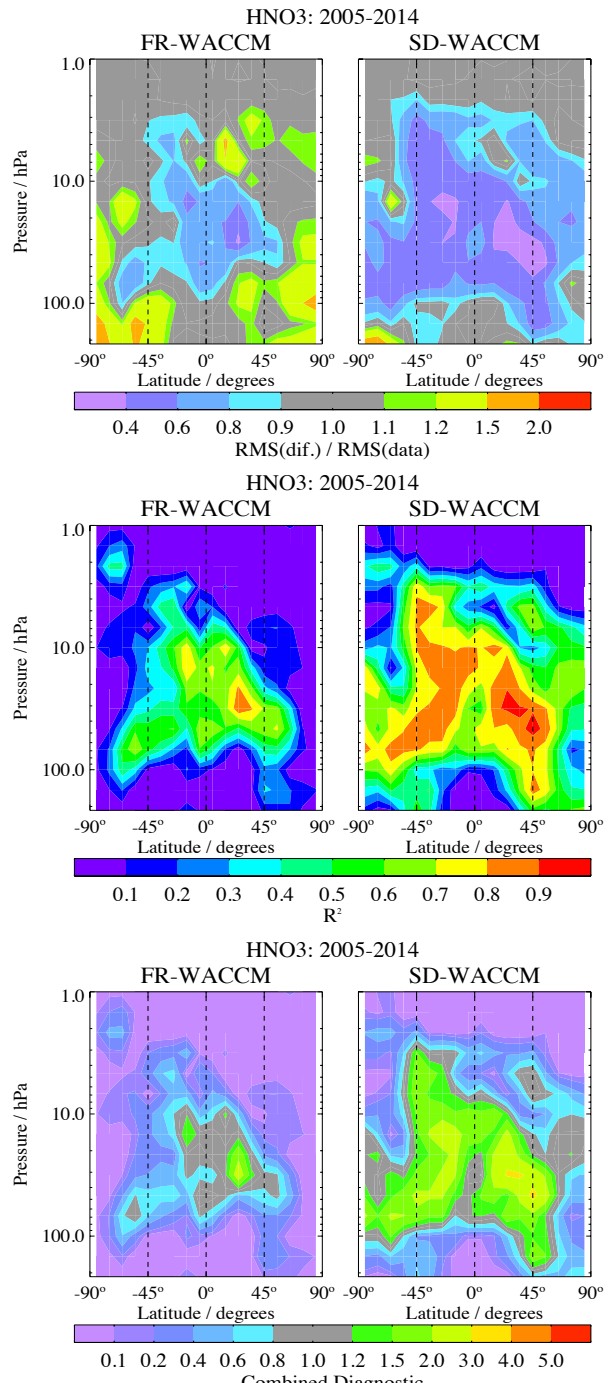

4
5   **Figure 15.** Same as the Fig. 10 diagnostics, but for HNO₃ (from 215 to 1 hPa).





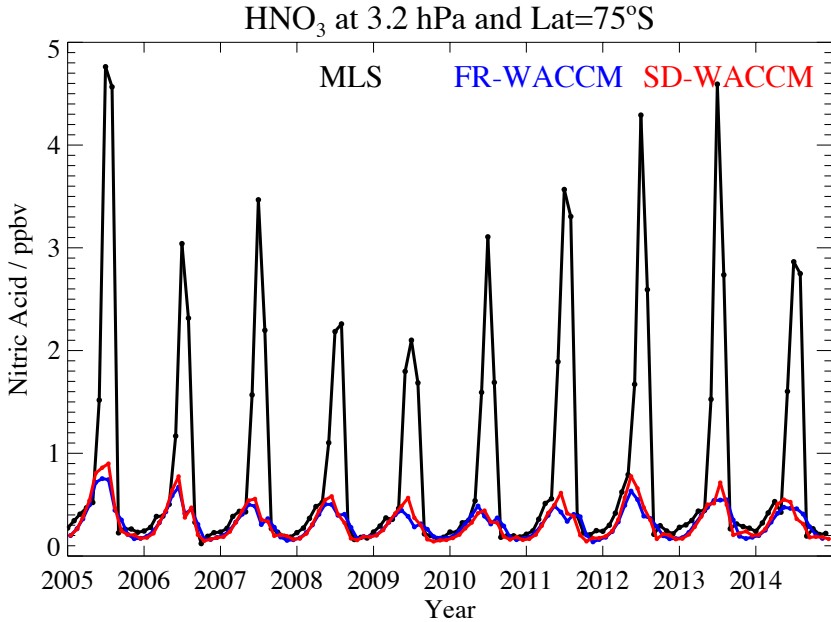

**Figure 16.** HNO₃ monthly zonal mean mixing ratio time series (2005 through 2014) from MLS, FR-WACCM, and SD-
WACCM for 3.2 hPa and 70°S-80°S.



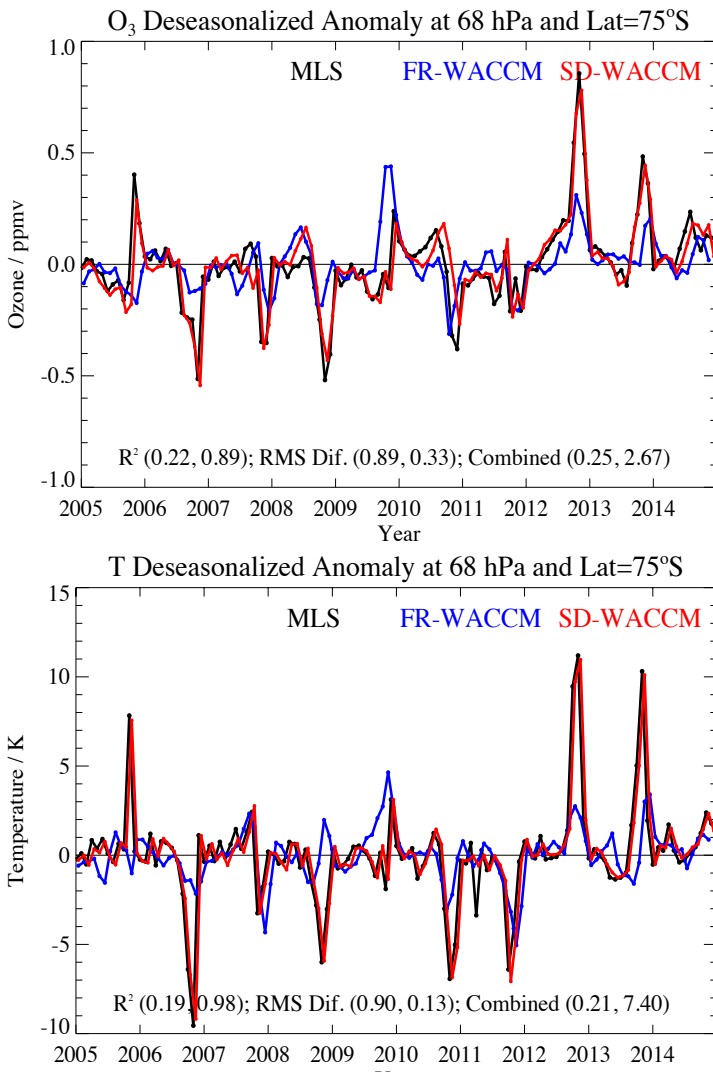

4
**Figure 17.** Deseasonalized anomaly time series (2005-2014) at 68 hPa and 70°S-80°S from Aura MLS, FR-WACCM, and SD-WACCM for ozone (top panel) and temperature (bottom panel); calculated values are provided in parentheses for the $R^2$, RMS fit, and combined diagnostics used in this work (the 1st number refers to FR-WACCM and the 2nd number to SD-WACCM).


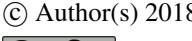


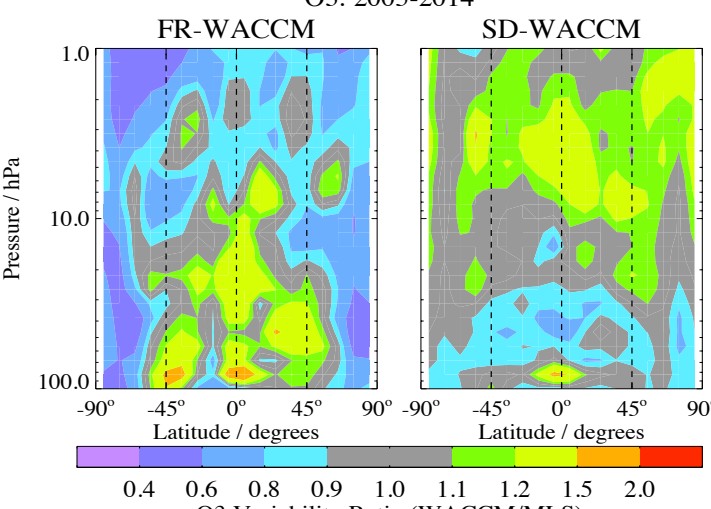

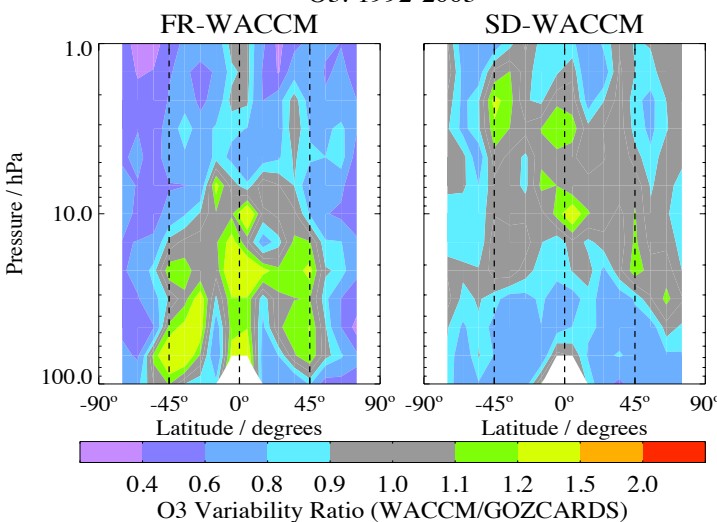

**Figure 18.** Variability ratios (model results divided by data results) for stratospheric O₃, with FR-WACCM results on the left,
and SD-WACCM on the right. Before calculating the ratios, the variability values are obtained as the root mean square of
detrended deseasonalized monthly anomaly time series, and expressed as a percentage of mean (climatological) abundances; the
top panels show comparisons to MLS data for 2005-2014, whereas the bottom panels are for 1992-2003 comparisons to
GOZCARDS.



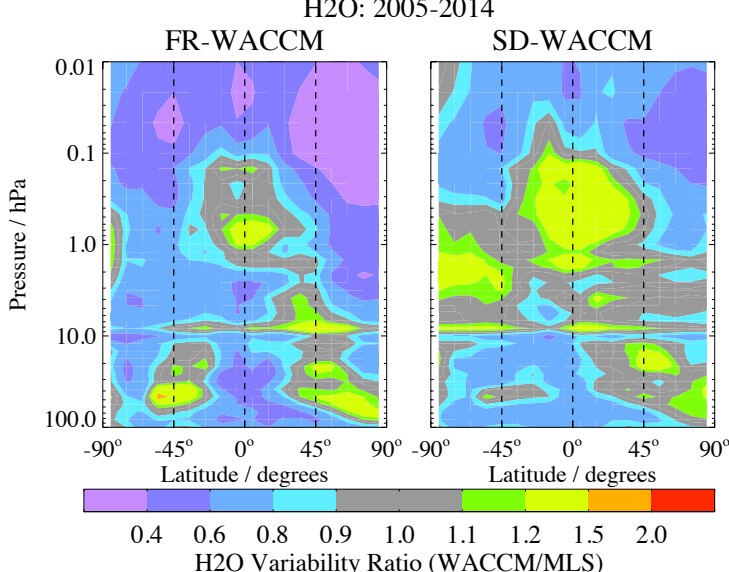

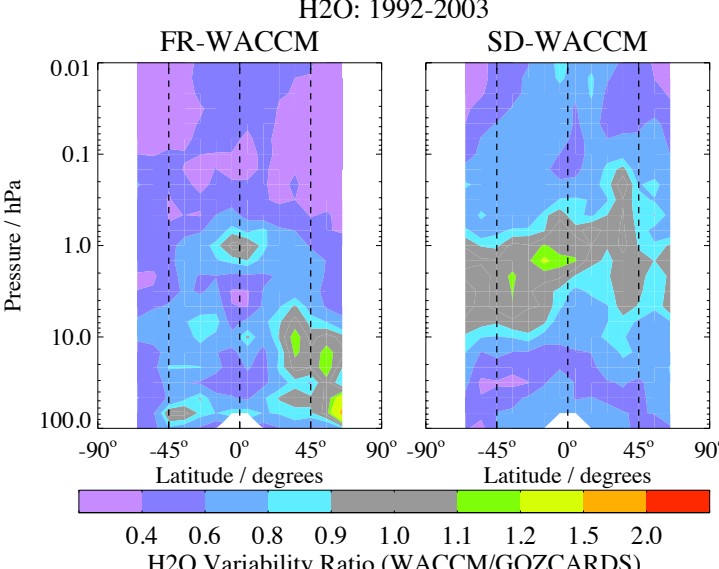

4
5 **Figure 19.** Same as Fig. 18, but for ratios (model/data) of $H_2O$ stratospheric and mesospheric variability for two different
6 time periods.





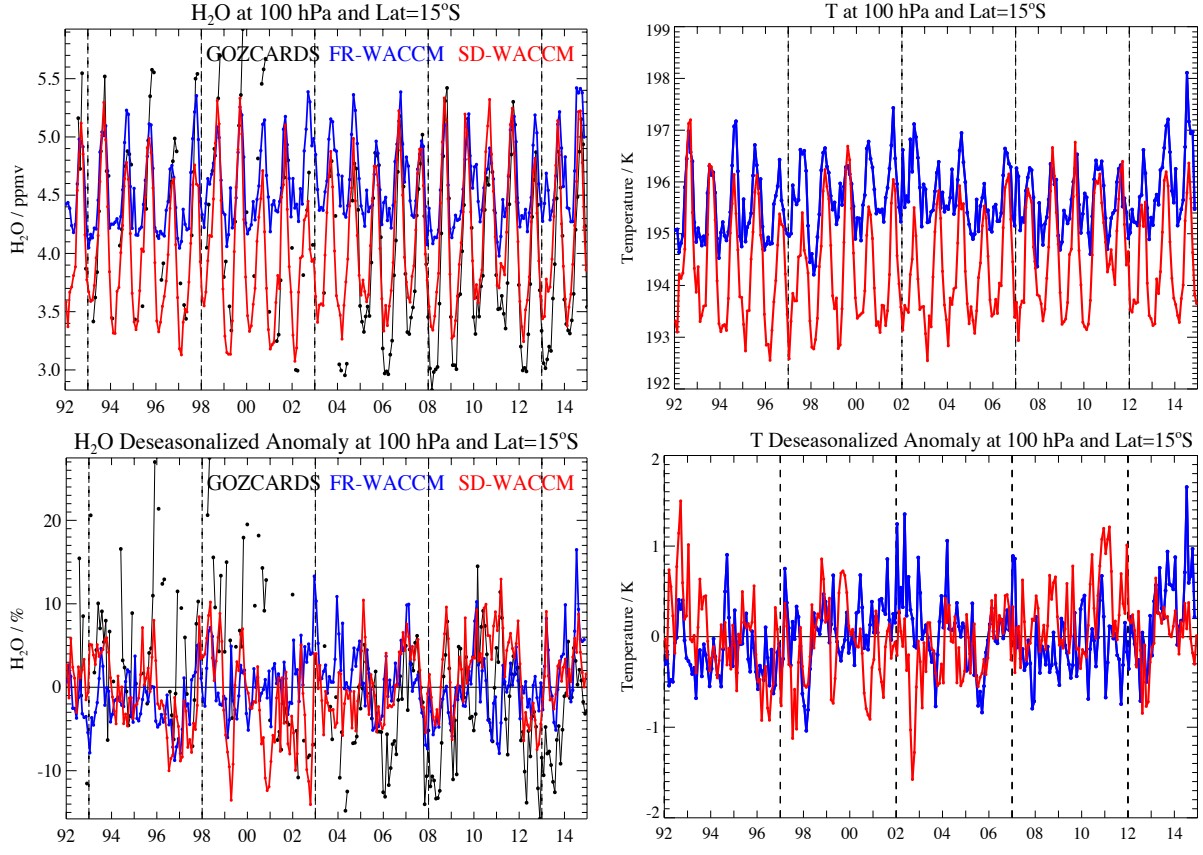

**Figure 20.** Time series (1992-2014) at 100 hPa and 10°S-20°S for temperature (right two panels) and $H_2O$ (left two panels), with
deseasonalized anomalies shown in the bottom two panels. The temperature plots just show the two models (FR-WACCM in
blue, SD-WACCM in red), whereas the $H_2O$ series show the comparisons for the models versus GOZCARDS merged $H_2O$ data
(in black).




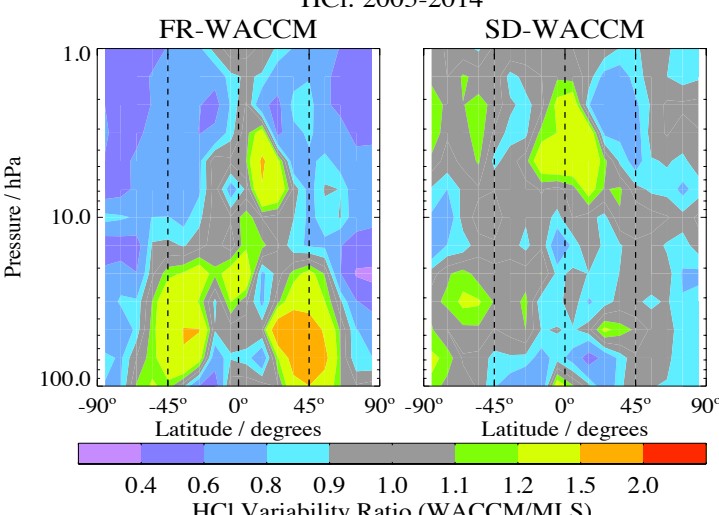

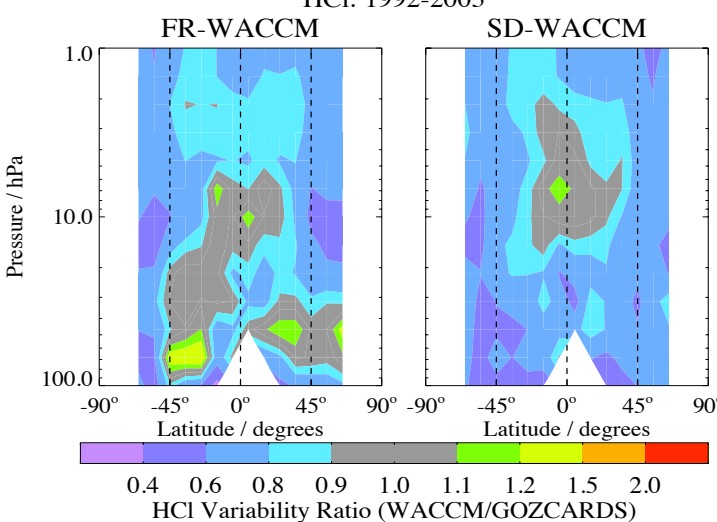

**Figure 21.** Same as Fig. 18, but for ratios (model/data) of HCl stratospheric variability.

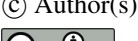



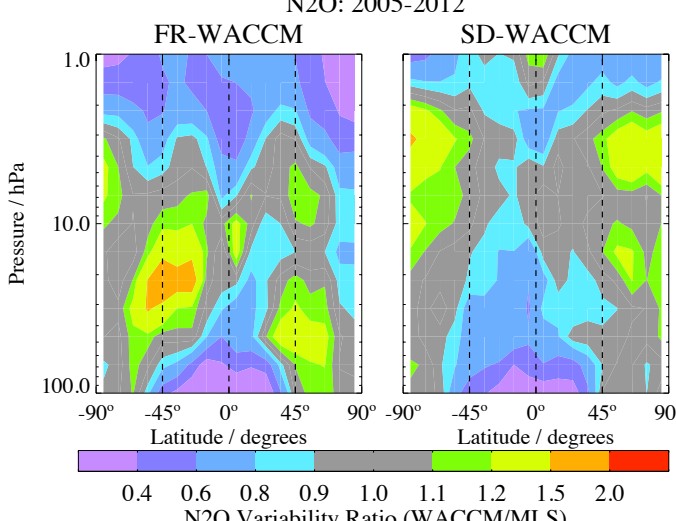

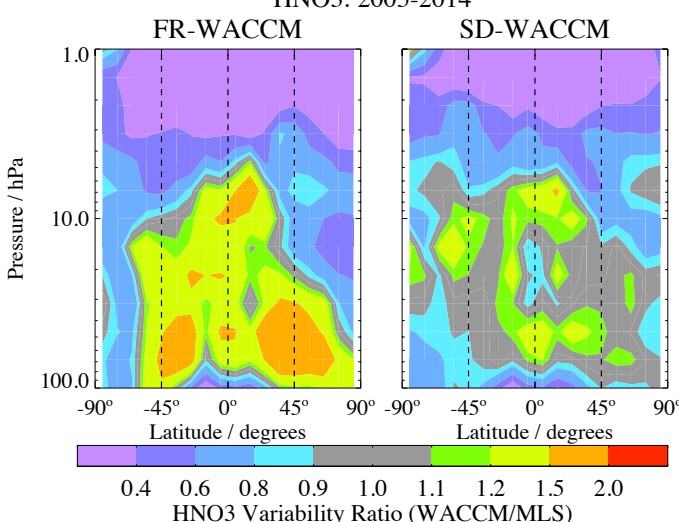

**Figure 22.** Same as Fig. 18, but for ratios (model/data) of the variability in stratospheric $N_2O$ for 2005-2012 (top panels) and
$HNO_3$ for 2005-2014 (bottom panels). The MLS N2O-640 product (see text) is used for the comparisons in the top panels.





**Figure 23**. Sample time series of deseasonalized ozone anomalies (%) from 1979 through 2014 from the GOZCARDS data record (version 2.20) compared to the corresponding model anomalies from FR-WACCM (blue) and SD-WACCM (red). Upper stratospheric series at 3.2 hPa are shown in left panels and lower stratospheric series at 68 hPa are on the right; three latitude bins are displayed (30°N-60°N, top; 20°S-20°N, middle, and 30°S-60°S, bottom).



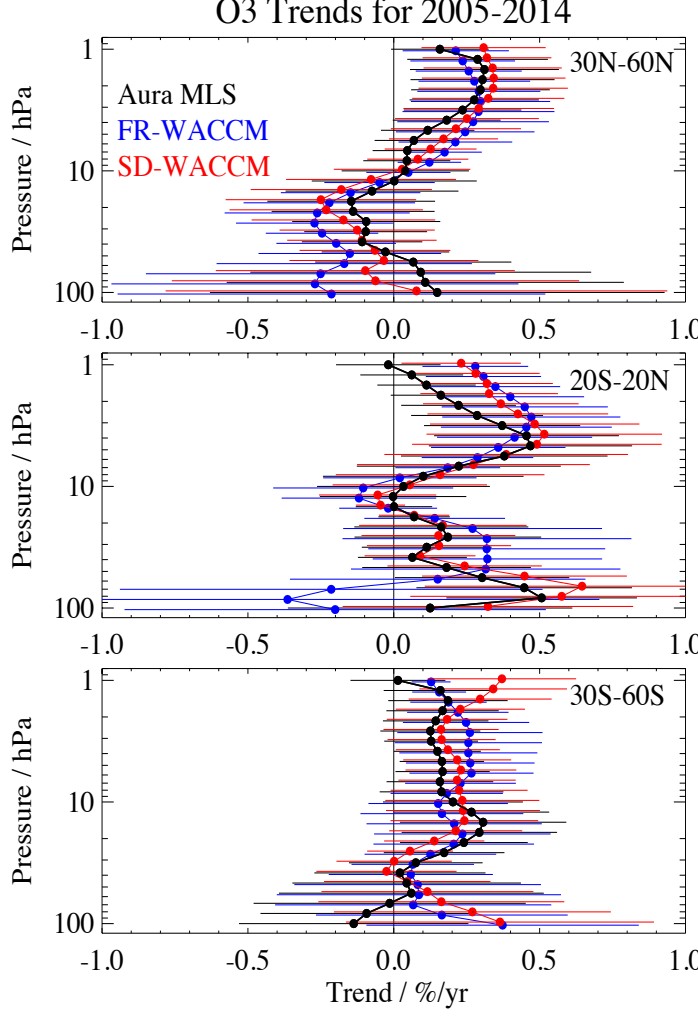

**Figure 24.** Ozone stratospheric trends for 2005 through 2014 obtained from monthly zonal mean data (version 4.2 Aura MLS)
and models (FR-WACCM and SD-WACCM), after multiple linear regression analyses of deseasonalized anomaly time series, as
described in the text. Each panel refers to results from different latitude band average series (see legend). The error bars are 2σ
estimates based on bootstrap resampling results (see text).



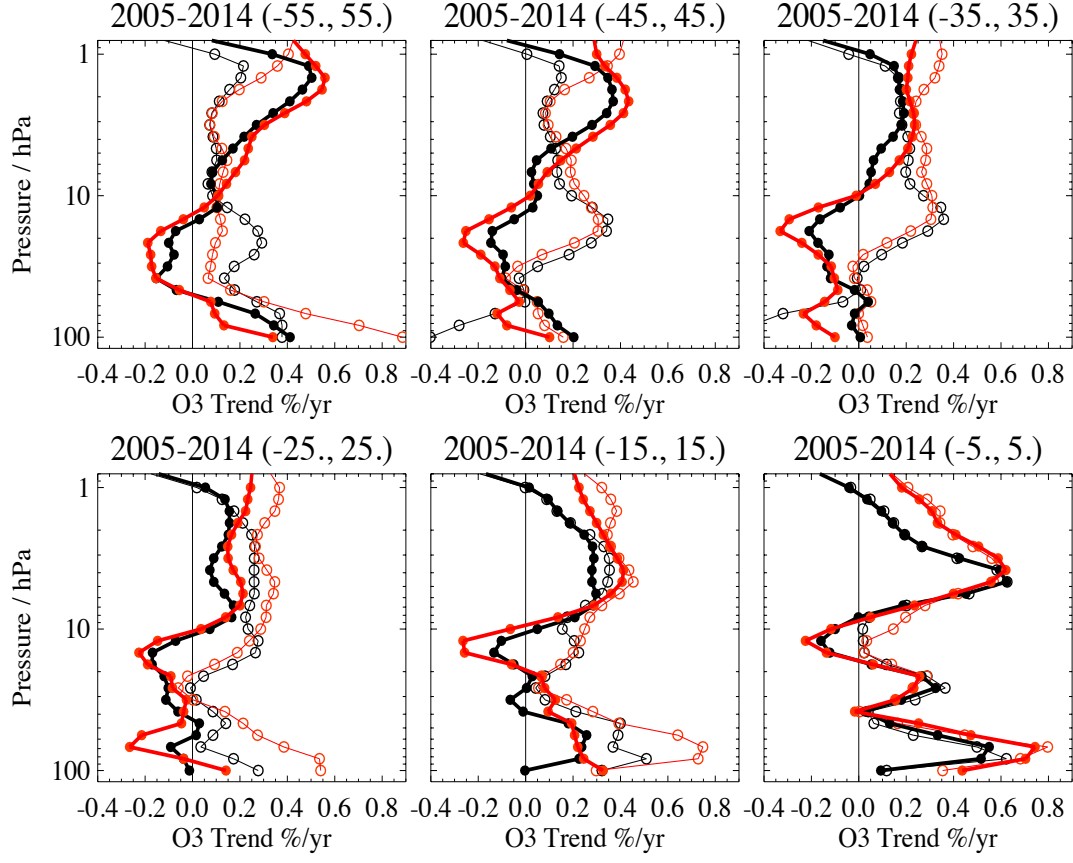

**Figure 25.** Ozone trends in different latitude bins for SD-WACCM (red) versus MLS data (black) for 2005-2014. Closed and
open circles are for northern and southern latitude bins, respectively. For clarity, error bars are omitted here, as these generally
show that model/data trend differences are not significant for this time period.





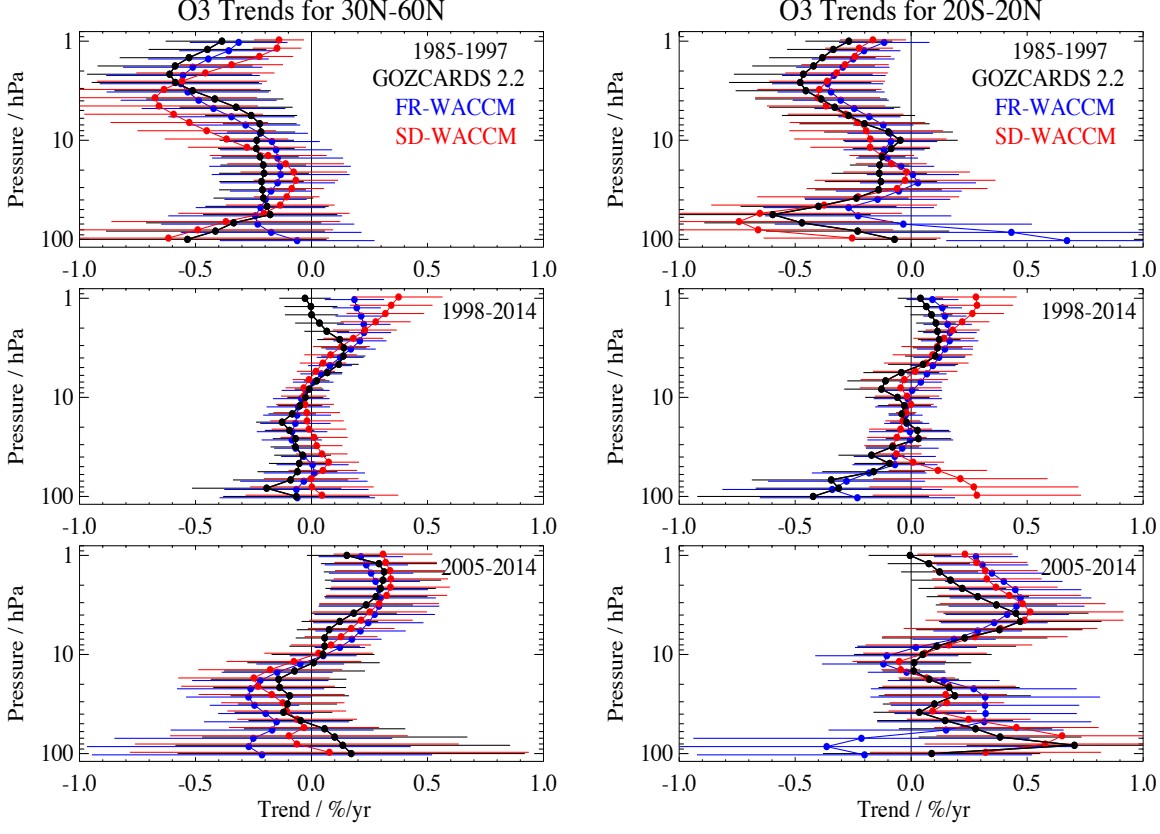

4
**Figure 26.** Same as Fig. 24 for ozone trend comparisons, except for the use of two latitude bands (30°N-60°N on left side and
20°S-20°N on right side) and three different time periods (from top to bottom panels, see legend); the data record here is from
GOZCARDS merged ozone version 2.20 (see text).






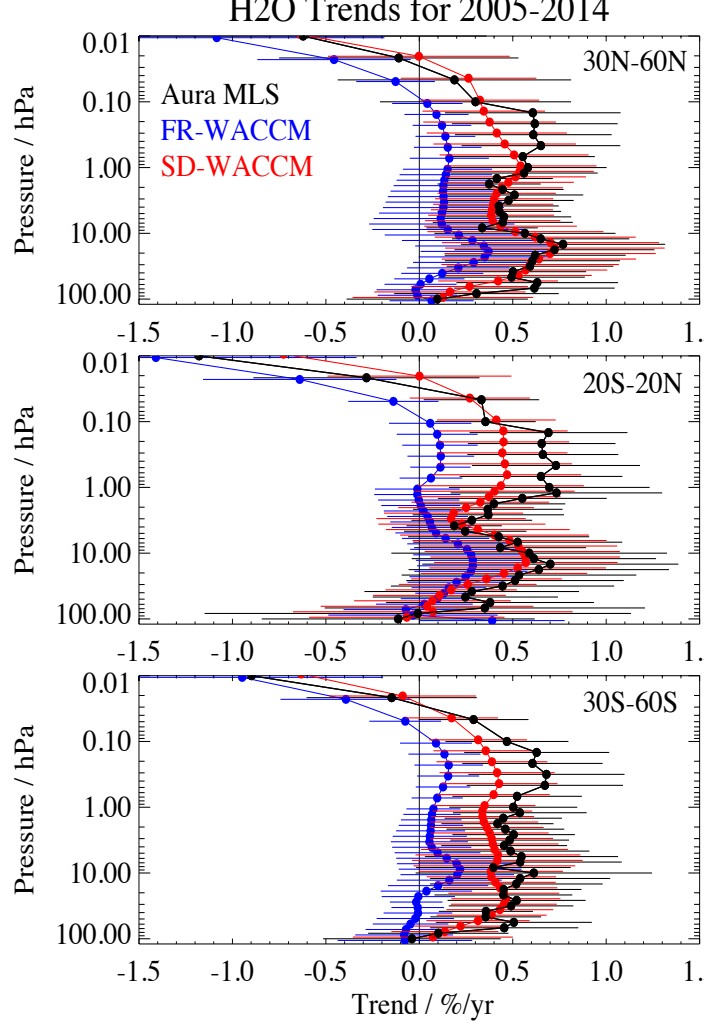

5 **Figure 27.** Trends in three latitude bins for stratospheric and mesospheric H$_2$O from an analysis of the 2005-2014 MLS data and
6 the two WACCM models over the same time period.





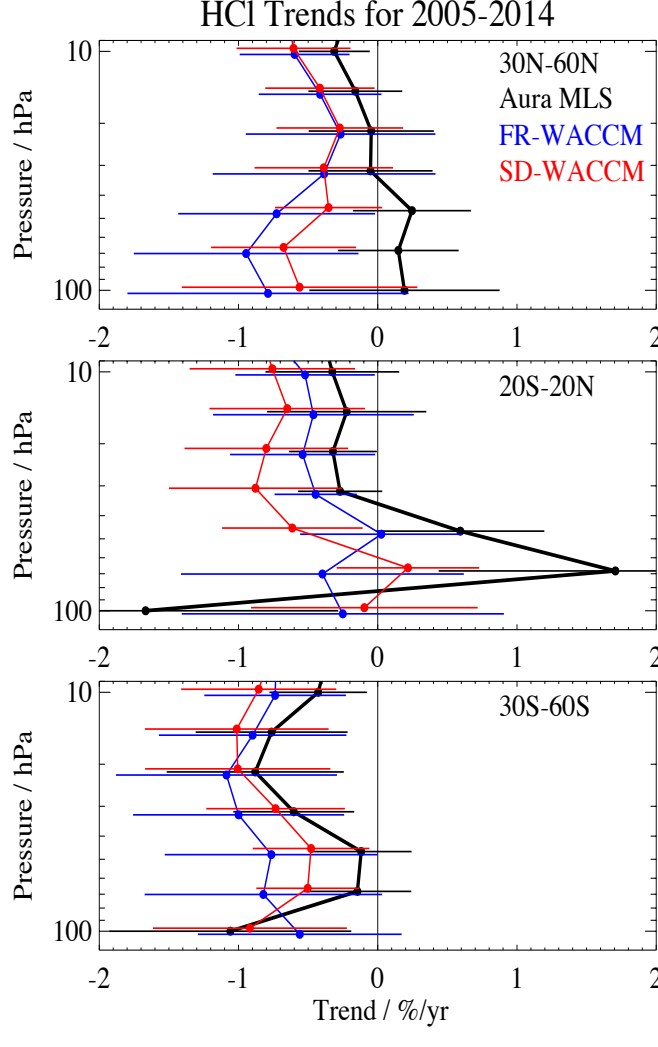

**Figure 28.** Same as Fig. 24, but for HCl data and model trends in the lower stratosphere.





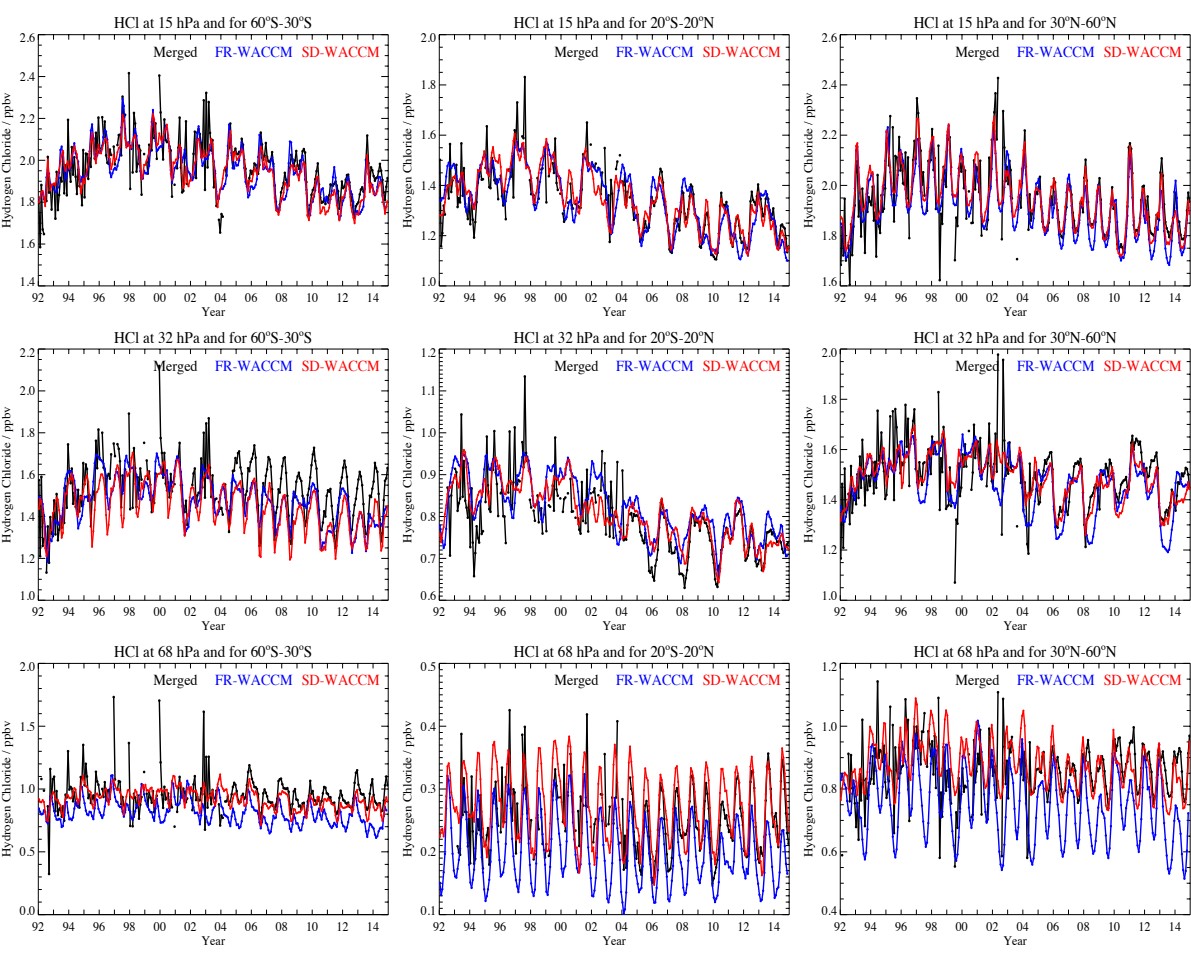

**Figure 29.** Time series (1992-2014) of lower stratospheric HCl (ppbv) for the GOZCARDS HCl merged data record (black), as
well as models (FR-WACCM in blue and SD-WACCM in red). Each panel is for a different pressure level and latitude bin, as
labeled (15 hPa, top; 32 hPa, middle; 68 hPa, bottom); the three latitude bins used in this work are 30°S-60°S (left panels), 20°S-
20°N (middle panels), and 30°N-60°N (right panels).



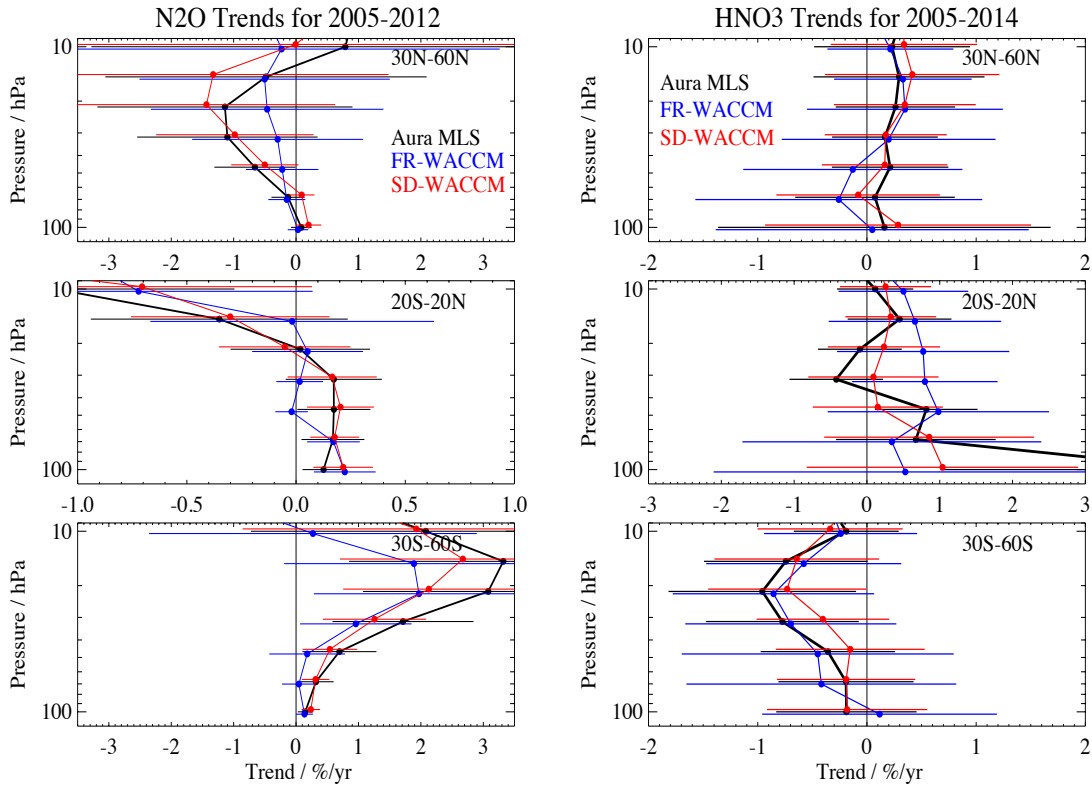

**Figure 30.** Same as Fig. 28, but for $N_2O$ (left 3 panels) and $HNO_3$ (right 3 panels) data and model trends in the lower
stratosphere. The $N_2O$ data results are from the N2O-640 MLS product (retrieved from the 640 GHz radiometer band data),
which was discontinued in 2013 because of an instrument issue affecting this band (see text), and these data and model trends
apply to the 2005-2012 period. The $HNO_3$ trend results (data and models) are for 2005-2014.



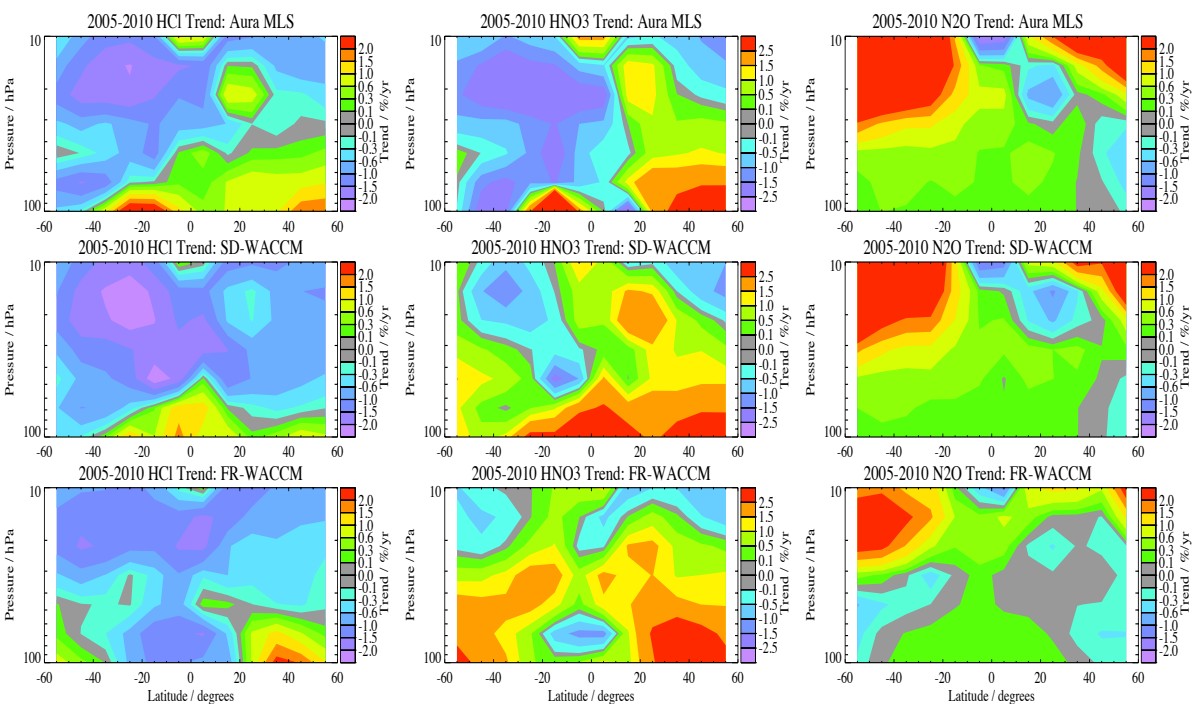

5  **Figure 31**. Pressure/latitude contour comparison of short-term (2005-2010) lower stratospheric trends from Aura MLS data (top
6  panels) versus model trends (from SD-WACCM in middle panels and FR-WACCM in bottom panels). The left column is for
7  HCl, the middle column for HNO₃, and the right column is for N₂O (using data from the MLS 640 GHz radiometer band).



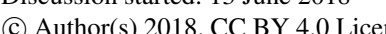

**Figure 32.** 25-year simulation from SD-WACCM for 1990 through 2014, showing monthly mean deseasonalized anomalies (%)
for $O_3$, $H_2O$, $HNO_3$, and $HCl$ at 68 hPa for 10°S-10°N, in a region where dynamically-induced variations, both short- and long-
term, appear to affect these species in similar ways. Some smoothing (using a running mean with a ±7-month window) has been
applied to highlight the variations longer than a year.

