# Peer review of "Evaluation of CESM1 (WACCM) free-running and specified-dynamics atmospheric composition simulations using global multi-species satellite data records"

_Atmospheric Chemistry and Physics, 2018_

## Referee Comment (RC1) · Anonymous Referee #1 · 18 Aug 2018

This paper evaluates two versions of the WACCM model using satellite observations, mainly from AURA MLS, but also using multi-instrument compilations. The paper contains some interesting results, but it is also very long (70 pages in the submitted format and 32 figures, plus supplement) and focussed on one specific model.

The paper contains an evaluation of the model (SD and FR) for 5 species compared to satellite observations. The paper points out general agreement and areas of disagreement, but the reasons for any disagreement are not really looked into (except for HNO3 and the lack of ion chemistry). The comparison of the performance of the SD and FR

models is a main focus of the paper. There are differences but overall conclusions on the accuracy of SD models, for example, seem to be missing.

The paper also uses both the models and data to look at trends. Reading the abstract paragraph which summarises the trend work does not give me a clear view of the main scientific points that have come out of the trend work. Is there something new about the observed known recent upper stratospheric ozone increase (i.e. recovery)? Or are the main points related to whether SD or FR simulations are better for studying past trends of different types of species (and I realise there are potential issues with both approaches). The paper also discusses metrics which can be used to evaluate CCM runs using observations. There is a lot of information here but again the main messages and recommendations are not clear to me.

For me as a reviewer the questions about this paper are (i) what are new scientific results related to CCMs (including diagnostics) or trends in general and (ii) why does the evaluation of the two WACCM versions belong in an ACP paper, rather than the sister journal Geophysical Model Development (GMD). At the moment, and using the abstract as a basis, I really don't get the main scientific advances which would justify ACP versus GMD.

My recommendation is that the work needs to be presented with clearer scientific messages coming through in the abstract and conclusions. Work which does not directly contribute to the ACP-level results could be put in a GMD paper, or an expanded supplement.

Minor comments

Page 1. Line 20. Can you be quantitative when discussing model over/underestimates?

Page 1. Lines 26-27. In what way are the detailed interactions not as well represented?

Page 2. Line 12. 'differences' rather than 'variability'?

Page 2. Line 14. 'driven' – not the correct word for what is inside the model. Usual used for the external forcings like winds or emissions etc.

Page 2. Line 17-18. I think you should say a lot more about other SD work and cite papers, as SD v FR is a main focus of this paper. This would help to think about whether the WACCM SD results may be applicable to other SD models?

Page 2. Line 27. Explain 'high quality'.

Page 2. Line 36-39. Can you give examples of trend studies that have had these problems? Again, for the trend results presented here to be of scientific interest to the community, we need to know about issues of what has been done before.

Page 3. Line 34. After reading these sections it is not clear to me if ACE data (and which version) is included in either of the GOZCARDS versions. Please clarify.

Page 4. Line 39. Clarify that 'organic halogens' are the source gases.

Page 5. Line 2. So the FR WACCM is relaxed to the observed tropical winds (QBO). What is the implications of that for the comparison? Does that constrain some of the comparisons? What would happen without this relaxation? (Why is it done?).

Page 5. Line 11. New paragraph before 'Both'.

Page 6. Line 28. The model comparisons don't use the satellite averaging kernels (or temporal sampling I suppose?). Can you add more details on why you see no reason to apply the AKs?

Page 7. Line 13. Any idea why there are larger differences for SD WACCM? What are the implications for SD studies?

Page 7. Line 16. Explain 'good dynamical tracer' for non experts.

Page 12. Line 36. 'do not have the right chemistry'. I would suggest rephrasing this.

---

## Referee Comment (RC2) · Anonymous Referee #2 · 24 Aug 2018

The manuscript aims to evaluate the stratospheric composition of the free-running and specified-dynamics version of CESM1 (WACCM). The evaluations are based on comparisons to satellite measurements including single-instrument and merged data records. The model diagnostics include zonal monthly mean comparisons, seasonal and semi-annual cycles as well as long-term trends. All evaluations are described in detail and valuable information on various aspects of the model performance is provided. Overall, the manuscript is of great interest for scientist directly working with WACCM or potentially with other earth-system models. Therefore, such a detailed

manuscript would seem much more appropriate in a journal focused on geoscientific model development/validation and I would urge the authors to submit it to a journal focused on this topic.

Major comments

1) The paper delivers a lot of valuable and detailed information, however, is overall very long. In particular, the number of figures could be reduced from 32 to around 20. To give one example, Figure 2 is only discussed very briefly in the text in order to illustrate mean biases and annual cycle differences shown elsewhere and could be removed.

2) Differences are often only listed and not explored more in detail. To give one example, model HCl shows systematic differences in the lower stratosphere (evaluation based on Fig. 4) and a discussion relating those differences to shortcomings in the model transport or model chemistry would be interesting. Given the length of the manuscript, one could focus on the gases for which the detected differences are discussed in terms of model behavior (e.g., HNO3). Differences for other gases can be mentioned in the manuscript with the according figures being moved to the supplement.

3) In section 3, existing evaluations of WACCM and the WACCM composition in particular should be discussed. Such references come up in the latter part of the manuscript. If they are given combined in this section, it will easier for the reader to identify what the current challenges are and what is new in this manuscript.

Minor comments:

1) Consider changing the title to 'Evaluation of CESM1 (WACCM) free-running and specified-dynamics stratospheric composition simulations using global multi-species satellite data records'.

2) Page 5, line 31 – Page 6, line 2: This text could be moved to the discussion of the MLS data record in section 2.1.

3) Page 7, line 24: Do you mean all earth system model or just WACCM with the term

'general model underestimation'?

4) Page 9, line 7 -10: Here, and also in other places, the sentence is too long for easy understandability. Consider splitting into two sentences at the semicolon.

5) Page 12, line 5-8: The statement is made for the upper mesosphere. But isn't it also true for the stratosphere?

6) Page 13, line 7: MIPAS has been used earlier in the manuscript.
* * *

---

## Author Comment (AC1) · 18 Oct 2018

**Reply to the review from Referee 1**

We are thankful to this referee for the review and the associated suggestions, listed in italics below. We provide our detailed responses (regular font) and plans; our revised manuscript will be available in a fairly short time.

It would seem from the referee comments that there are no demonstrable big issues with the science (or math), besides some requested clarifications, and we are pleased that Referee 1 found our manuscript to contain "some interesting results" [and Referee 2 found "a lot of valuable and detailed information"]. We hope, furthermore, that the plans we describe herein for clearer messages and revisions will be of a satisfactory enough nature, or we will need to ask for more specific comments.

*(1) This paper evaluates two versions of the WACCM model using satellite observations, mainly from Aura MLS, but also using multi-instrument compilations. The paper contains some interesting results, but it is also very long (70 pages in the submitted format and 32 figures, plus supplement) and focused on one specific model.*

**Reply:** We are planning to cut down on the length of this manuscript, mainly by relegating some of the less critical Figures to the Supplement. Although this does not necessarily translate into a very large cut in terms of text length, we consider this work to be a fairly comprehensive analysis, which therefore leads to a longer paper; there have definitely been some longer (atmospheric) papers in the literature, and specifically in ACP. Turning this into two separate papers mainly for the sake of overall length seems too artificial, and this would be quite an elaborate proposition, with the need for some duplication regarding both the data sets and the models; as an aside, this would actually lead to more reviewing work for the community. We hope to have shown that detailed analyses are necessary to enable identification of both good agreement (a result in itself) or significant differences between model runs and the data sets, but also for some of the more subtle differences, and furthermore, that an understanding and discussion of error bars and potential data issues is important. We will also strive to reduce the amount of text in the revised manuscript, especially where some less critical aspects can be discussed more succinctly, or taken out altogether. In particular, we plan to shorten Section 5.1.1 (pages 11-13) to a text length that roughly matches (rather than exceeds) the text length of Section 5.1.2 (on variability issues); the cuts to Sect. 5.1.1 will be of order 30% (or more).

In terms of reducing the amount of Figures and related changes, our specific plans are to remove Figs. 13, 14, 15, and 17 from the main text (and relegate these to the Supplement, with a slightly shortened discussion), since these mainly reinforce the expectation (already noted for $O_3$ and $H_2O$) of better model/data fits from SD-WACCM, as one might expect from a model with better dynamical constraints than the FR-WACCM version. Such an expectation does not hold for the variability diagnostics, so these are really best left in the main text, although we will plan to displace Figure 22 (on the $N_2O$ and $HNO_3$ variability comparisons), and move it to the Supplement. Moreover, we feel that Figure 31 on lat/p contours of short-term trend for various species can be moved to the Supplement, as it is less critical, and given past (and ongoing) work on this topic. While Figure 32 is interesting to us, it is more of a side note on lower stratospheric tropical cohesiveness for various species exhibiting similar dynamical variability, so we decided that the text and Figure in this case can be eliminated altogether without much of an impact on this paper.

In summary, the total number of Figures in the main text will be trimmed down by almost a quarter, with a more manageable total of 25 Figures; writing up a multi-year effort of (part-time) work on detailed model/data comparisons is bound to lead to a longer manuscript than several shorter analyses; to our knowledge, fits, correlations, variability, and trend comparisons are rarely investigated to this extent in model/data comparisons, even for a single model (or two flavors of one model). This, with some reductions (and clarifications) in the text (including the Abstract and Conclusions section), will at least show our good faith effort towards the referee comments. Recommending a goal of exactly 20 Figs. (as done by Referee 2) is rather arbitrary, but our point here is that we have considered these requests with some care, and that we are being responsive.

*(2) The paper contains an evaluation of the model (SD and FR) for 5 species compared to satellite observations. The paper points out general agreement and areas of disagreement, but the reasons for any disagreement are not really looked into (except for $HNO_3$ and the lack of ion chemistry).*
**Reply:** There are several aspects to these model/data comparisons. Looking carefully into reasons for disagreement can be the subject of separate papers altogether, possibly involving new model runs (which would take quite a bit of time), and this would also increase the length of this (already long) manuscript. We will point to where some likely causes can be mentioned, although in all fairness, we believe that this has already been done in several places (see more in the numbered

list below), and beyond the HNO$_3$ issue mentioned by this referee. However, we are also adding more information and discussion in various places (see further below).

In particular, we provide further explanatory material in some of the following areas, ignoring from this list the HNO$_3$ issues (and lack of full model chemical pathways) already sufficiently described in the manuscript. Without the exact revised text for now, please see the following list of specifics, although some of the items in this list are there to provide some rebuttal to implications that we provide few explanations besides showing the comparisons themselves (or the advantages of one model version versus the other). We strongly believe that these comparisons (in themselves) are worth displaying in a publication, even if this only applies to the WACCM model, which is considered state-of-the art. Moreover, and almost as importantly, we have shown that areas of disagreement very often fall within the estimated error bars, so there are not that many really significant discrepancies; we hope to take some credit, in fact, for trying to be careful about including realistic error bars in many of these comparisons. However, an investigation into other models for similar areas of agreement or disagreement is beyond the scope of this work, which we consider a first step that can help other modeling groups focus on certain regions of potential disagreement. Later on, another paper could hopefully identify where, and maybe why, certain models do better than others in certain places or time periods; in fact, some of this may already be "in the works" or near completion (based on a list of planned studies for CCMI at [www.met.reading.ac.uk/%7Eqr903932/CCMIwebsite/Wordpress_PDFs/CCMI1_PlannedAnalysis_20170715.pdf](www.met.reading.ac.uk/%7Eqr903932/CCMIwebsite/Wordpress_PDFs/CCMI1_PlannedAnalysis_20170715.pdf) ).

1. *Page 6, 1$^{st}$ paragraph*: We now make the point regarding the Fig. 2 (and Fig. 1) model/data lower stratospheric O$_3$ differences near 50°N-60°N (even if it may be obvious) that transport-related model issues (not chemistry issues) are the most likely reason for the models to significantly overestimate mean ozone and its seasonal cycle at mid- to high latitudes. In addition, we are adding related information in the text for H$_2$O comparisons, given that we also see a significant (factor of two) WACCM model overestimate of the MLS H$_2$O fields (mean value and seasonal amplitude) in the same region (detailed plots not shown); this discrepancy goes beyond a (previously documented) 30-40% dry bias of MLS H$_2$O versus sonde data a few km below the tropopause. However, digging into model details (or even the meteorological fields),

in addition to possible other data sources (or data issues) for $O_3$ (and $H_2O$) comparisons would need to be the subject of a new investigation, interesting as it might be.

2. *Page 8, lines 6-14, and Fig. 7*: Regarding the seasonal changes over Antarctica, our analyses include species other than HCl and provide more of a climatological description regarding this discrepancy in HCl behavior than what was shown in the paper by Groos et al, (ACP, 2018). The latter work attempted to ascribe such a discrepancy to various factors, without a fully satisfactory answer, and we do not currently have further thoughts on this topic, as more detailed investigations (not speculation) would be required to make further progress. On the same topic, we do provide a likely explanation for the better matches from SD-WACCM (vs FR-WACCM), namely the connection to more realistic temperatures.

3. *Page 11, lines 36-38 and Section 5.1.1*: The better SD-WACCM results (here and in this section more generally) regarding model/data fit diagnostics and model/data correlation coefficients are related to the better dynamical description for the "specified-dynamics" version of WACCM, as we point out in this section in more than one place (see also the 2^nd part of the top paragraph on page 12, regarding $H_2O$ comparisons). This is the main result from the discussions on pages 11-13, and a result that is worth including in this paper (in our opinion), even if there are other (probably more illuminating) results.

4. *Section 5.1.2 on model/data variability comparisons*: The interannual variability in monthly means represents a useful diagnostic of model/data comparisons, and it also relates to trends and detectability of trends, as we point out in this section. The main variability disagreement between models and data involves water vapor, a species that is also more difficult to model, given its different phases and its more complex pathways for entry into the stratosphere, the influence of ENSO and cold point temperatures, as well as the QBO and circulation changes, along with changes in methane and (mainly in the mesosphere) the solar cycle impact. Some of these processes (or their variability at least) are possibly not sufficiently well represented in either SD-WACCM or FR-WACCM, but there are better fits to the data from SD-WACCM. Also, on the variability issue, we do make the point that the $H_2O$ interannual variability is underestimated by the models, and since the uncertainty in trend detection

depends on the variability, a larger than modeled atmospheric variability implies that it will take longer than expected to detect long-term trends in water vapor.

5. *Section 5.2 (trend comparisons)*: Here are the main points for each species:

- **$O_3$:** MLS data alone have not been used yet to document trends (for the MLS years of operation that overlap the model runs), so this is a novel result, even if the time period is short enough that the ($2\sigma$) trend error bars are often fairly large. If one averages the results over the upper stratosphere, there are robust indications of an increase, based on the MLS data alone, and this avoids some issues associated with merged data sets (e.g., changes in spatio-temporal coverage between different instruments). It is also interesting to see indications of increases in the tropical lower stratosphere (albeit with less robustness than in the upper stratosphere), in apparent agreement with the SD-WACCM results. Most notably, the lat,/height patterns of trends, ignoring the absolute error bars, are remarkably similar for MLS data and SD-WACCM (see Fig. 25); we feel that this is a very informative plot. Furthermore, version 2.20 of GOZCARDS $O_3$ is evaluated for trends in this work, and we highlight some differences versus the original GOZCARDS data set.

- **$H_2O$:** The main points for this species are now made more relevant, we hope, in the context of what one might expect from longer-term trends versus what happened during the shorter-term (2005-2014) versus MLS trends, which are significantly larger than what one would expect from the water vapor changes caused by increases in methane alone. We also note that FR-WACCM trend results are significantly smaller than SD-WACCM (and observations), but this does not imply a longer-term systematic underestimate from FR-WACCM, based also on our looks at longer-term time series (although these are not displayed specifically in this manuscript). The Abstract has now been changed to reflect these points as well.

- **HCl:** We are not planning much change regarding these results, and we think that the main points are clear enough: there is some underestimation of MLS HCl trends from the models, and some LS tropical positive trends in these observations which deserve further investigation. However, we will add a pointer to recent work (if it gets in press soon) that shows the impact on HCl trends of a better treatment of VSLS and their trends, as this seems to be a way to close at least part of the gap

(model versus data trends). The other issue could be related to an MLS overestimate of the HCl LS trends (as this is what happens in the upper stratosphere, a known issue for MLS HCl there).

- **N₂O and HNO₃:** There is good agreement for these two species overall, in terms of the model versus data trends. Some of these trend variations versus height (particularly for $N_2O$) must be related to stratospheric age of air and circulation, but we also clearly see (in time series not shown here) that the QBO, in particular, has a large impact on the variability, as one moves away from the tropopause region; this is a well-known feature. This large percent variability (as one reaches the mid-stratosphere) swamps the underlying long-term trends in $N_2O$. The WACCM time series capture the observed (MLS) variability remarkably well, and the trends for 2005-2014 reflect this sort of agreement (Fig. 30). There are some slightly larger differences in terms of the somewhat poorer phasing of variability (and fits to the data) for FR-WACCM, but the main features versus latitude and height are well reproduced. This also holds for $HNO_3$. We will thus add a few words very similar to these in this part of Section 5.2, in terms of our understanding (and at least partial explanation) of these trends and their variations.

Our draft revised Abstract (see below) also hopefully clarifies the main points in a somewhat better way (without making it much longer), as a response to the Referee comments. The revised text will add related information for clarifications and context; it will also be trimmed elsewhere to try to address the issue of paper length inasmuch as possible (without losing too much content).

*(3) The comparison of the performance of the SD and FR models is a main focus of the paper. There are differences but overall conclusions on the accuracy of SD models, for example, seem to be missing.*

**Reply:** We do not fully understand this comment, but we will attempt a reply that covers the options. It is really beyond the scope of this work to try to dive into why SD models differ from one another, if that is the reviewer's point, although we think that this would be an interesting study for the future. We have examined only the SD-WACCM/MERRA model in detail in this study using multiple diagnostics. There will be future papers that compare processes and biases

between the participating CCMI SD models (as mentioned earlier). The point of our work is to perform a detailed model/observational analysis of two configurations (FR and SD) using the same modeling system (i.e., CESM framework). Here, the tracer advection routine (Lin, flux form finite volume) and WACCM chemistry module (for gas-phase, heterogeneous, and photolysis reactions) are identical between the two configurations. The differences between the two configurations are mainly due to how the circulation is derived. The FR configuration allows the ozone to be interactive with the heating rates and therefore circulation. The SD configuration uses a specified meteorology that drives the circulation. Therefore, when we compare the FR and SD model versions observation-based diagnostic, the "goodness" of the results between FR and SD removes uncertainty of both the advection and chemistry assumptions (since they are the same). However, there is still uncertainty in the derivation of the circulation in FR and the nudging approach used with the observed meteorology. The approach described in this paper is essentially a first step in understanding how well models represent biases and variability in comparison to observations. The next step could be, of course, to examine diagnostics across multiple model systems, but not here (see also our response to item (1)). We plan to change part of the Introduction (and maybe the model section also) in the revised version to better motivate the purpose of our analyses of FR-WACCM versus SD-WACCM, as mentioned above.

If, on the other hand, the reviewer is asking about the accuracy of the specific SD-WACCM model run used here, most of the comparisons here show that there are few large areas of disagreement, beyond the error bars in the MLS data, so this is a clear statement (we believe) regarding the model accuracy (absolute), in comparison to state-of-the art observations; we also identified a few areas of disagreement. We could add (in the revised version) percentage difference numbers regarding the "accuracy level" (model/data agreement level) for each species, if this is what the reviewer is asking. We have preferred to let the first few Figures (Figs. 1,3,4,5,6 regarding climatological levels of agreement) speak for themselves. One often obtains levels of model/data agreement within about $\pm 5$ to 10%. However, quoting a more detailed range of "accuracies" versus species, pressure, and latitude, can add up to a fair amount of text. We have already highlighted regions where we believe that model issues might need more investigation, and some regions where data issues could also contribute to the differences (e.g., where more difficult retrievals and/or fewer data validation possibilities exist). The right panels in these climatological comparison Figures help to take into account the systematic errors in the MLS data. If the Referee

really wants us to add more numbers in the text (or in the already long Abstract), we can try to do this in the revised manuscript, but we would otherwise stick to the fact that one can extract numbers out of the Figures already present in the manuscript. If another model wishes to "measure up" to the same data sets, new Figures of this kind would need to be produced, for comparison purposes.

*(4) The paper also uses both the models and data to look at trends. Reading the abstract paragraph which summarises the trend work does not give me a clear view of the main scientific points that have come out of the trend work. Is there something new about the observed known recent upper stratospheric ozone increase (i.e. recovery)? Or are the main points related to whether SD or FR simulations are better for studying past trends of different types of species (and I realise there are potential issues with both approaches). The paper also discusses metrics which can be used to evaluate CCM runs using observations. There is a lot of information here but again the main messages and recommendations are not clear to me.*

**Reply:** There are both types of aspects in our results, and while we thought that this was already fairly clear, we can try to clarify where needed, if we are given more specifics from the referee, after our revised version is finalized. Indeed, some points are made in terms of trends themselves (e.g., $O_3$ trends that are positive in the lower stratosphere over the MLS period, whereas longer time periods have indicated some decreases – so further confirmation with more years of data should be worthwhile in the near future), while other points clearly deal with the comparisons with model trends. In many aspects, SD-WACCM matches the latitude/height behavior of observed ozone trends quite well, and also matches the observed $H_2O$ trends better than FR-WACCM.

*For me as a reviewer the questions about this paper are*

*(i)  what are new scientific results related to CCMs (including diagnostics) or trends in general*

**Reply:** Please see our replies above, as this reviewer comment is mainly a summary comment.

*and*

*(ii) why does the evaluation of the two WACCM versions belong in an ACP paper, rather than the sister journal Geophysical Model Development (GMD). At the moment, and using the abstract as a basis, I really don't get the main scientific advances which would justify ACP versus GMD.*

*My recommendation is that the work needs to be presented with clearer scientific messages coming through in the abstract and conclusions. Work which does not directly contribute to the ACP-level results could be put in a GMD paper, or an expanded supplement.*

**Reply:** In response to this, we have made some changes, notably to the Abstract, main text, and conclusions, with more useful information to help strengthen the results on ozone and $H_2O$ trends. Short of the revised version (which we are finalizing soon), please see the revised (draft) Abstract at the end of this reply, with the highlighted parts as a guide to the non-minor changes.

Stratospheric science has progressed to the point of being quite well understood from the point of view of very sophisticated tools, like SD-WACCM (with mostly correct representations, or parameterizations, of the physics and chemistry), and this limits the extent of significant new advances. However, this manuscript is (in our view) one of the more comprehensive studies that confronts such a model with multi-year and multi-species data sets, for species with different lifetimes and gradients, so that a fuller depiction of areas of agreement or disagreement can be revealed. On the trends side, there is good overall agreement within the error bars; more specifically, the degree of agreement for SD-WACCM in terms of the latitudinal and vertical patterns is actually striking (see Fig. 25 in particular), if one ignores the issue of absolute error bars. Figure 25 is also an example that could be illuminating for other model comparisons, in due time (not here). Such excellent agreement in the patterns of trends is a model success worth documenting, in our view; otherwise, it could become just "word of mouth" between modeling groups, and we feel that the actual publication is important, after careful (time-consuming) analyses. While there have been some rather broad trend comparisons in the past between averaged data sets and averaged models, there are few that go into a lot of detail for different model runs; more of this type of work may well be in preparation elsewhere.

On the issue of trimming (or splitting) the manuscript, we do feel strongly that using the Supplement is a much better way to help cut down somewhat on the main paper, rather than to somewhat artificially break up this comprehensive work, given that this would also require a significant amount of duplication and extra work. We believe that, after some trimming of Figures and less essential text, and other clarifications, as mentioned earlier in more detail, this paper will be improved. On the other hand, there is a need for some added text in order to explain some issues better, namely for water vapor trends, their magnitude (in relation to what one would expect from methane increases), and the differences between the two models. In the end, we feel that setting

an arbitrary length goal does not make much sense, when a lot of comparison work is investigated (or even summarized) for multiple species with different lifetimes, in order to confront the models with a multi-dimensional and multi-faceted atmosphere. However, we will heed the advice regarding a trimmed down revised version, and we thank both referees for these comments.

Regarding the Journal issue, we feel quite strongly that such a paper is (or can certainly be) in the ACP domain, given that the model description is really a small part of this manuscript (WACCM having been used and described previously, including in GMD, *Morgenstern et al*., gmd-10-639-2017), and that there are some scientific results discussed here (to be further clarified, as mentioned in our replies and in the upcoming revision), even if some of this confirms past/recent work, but from our own model/data comparisons. There is some "grey region" between ACP and GMD papers, with the latter being more geared towards model description and development (if one looks through many of those articles), although there are some model evaluation papers there as well. To be more specific, we include Table 1 at the end of this reply, and this provides a summary of all the papers that are part of the current CCMI special issue, which is what we are submitting to here; this special issue encompasses several journals (including ACP and GMD). As one can see from Table 1, the more recent papers have nearly all been part of ACP, after some initial work with much more of a model description focus. Some of the articles in ACP could compare broadly to the work we are trying to present, with a combination of model and data (and comparisons). We also feel that there are detailed aspects of the MLS data sets described in our work (regarding absolute error bars and trend uncertainties, including some drift issues) that would be of much interest to the stratospheric component of the ACP readership. Without attempting to be more comprehensive, we can state that we did consider the Journal topic seriously, which also led to some delays. We also consulted with the ACP editors on this topic, and we are pleased that they agree with our views; this topic is also something that editors consider as part of the pre-review process. It is also true that going through another 4 months of review with a completely new set of reviewers and editors is a considerable burden not just on the authors, with further time delays, but also on the reviewer community (especially for longer papers). We are thus thankful for the support we obtained towards finalizing this process for ACP, and we feel that we can now focus our efforts to that end; we would very much welcome reviewer support on this aspect as well.

*Minor comments*

*Page 1. Line 20. Can you be quantitative when discussing model over/underestimates?*

**Reply:** Certainly, these Abstract sentences are now rewritten for clarification, as follows:

"There are a few significant model/data mean biases, such as for lower stratospheric $O_3$, for which the models at mid- to high latitudes overestimate the mean MLS values by as much as 50% and the seasonal amplitudes by ~60%. Another clear difference occurs for $HNO_3$ during recurring winter periods of strong $HNO_3$ enhancements at high latitudes; the strong model underestimate in this case (by a factor of about 2 to 6) stems from the omission of ion chemistry relating to particle precipitation effects, in the global models used here." The relevant sections in the text will also be adjusted to match these more quantitative points.

*Page 1. Lines 26-27. In what way are the detailed interactions not as well represented?*

**Reply:** We have decided that this result, although correct, is not needed in the Abstract, given that one expects a free-running model to be less in-phase with actual dynamical situations represented better by SD-WACCM (and the observational record). This will therefore be removed from the Abstract, although the relevant (fairly brief) discussion can stay in the main text, as a demonstration of these somewhat subtle, but real differences, between model 'flavors' and observations.

*Page 2. Line 12. 'differences' rather than 'variability'?*

**Reply:** Yes, this wording is changed to 'differences'.

*Page 2. Line 14. 'driven' – not the correct word for what is inside the model. Usually used for the external forcings like winds or emissions etc.*

**Reply:** Yes, this wording is changed to "driven by time-dependent boundary conditions", without mentioning the photochemical reactions (which can be taken as a 'given', given other references to the model).

*Page 2. Line 17-18. I think you should say a lot more about other SD work and cite papers, as SD v FR is a main focus of this paper. This would help to think about whether the WACCM SD results may be applicable to other SD models?*

**Reply:** We understand the importance of comparing various SD models, and we have discussed this earlier in our reply to item (3). We plan to change part of the Introduction in the revised version to better motivate the purpose of our analyses of FR-WACCM versus SD-WACCM, as mentioned above.

*Page 2. Line 27. Explain 'high quality'.*
**Reply:** That is a fair comment, especially for a reader who might not know enough about the MLS data; however, for this Introduction, it would seem best not to try to give a detailed list of references on validation, etc… so we can just remove this somewhat vague wording for simplicity (and we are keeping the manuscript length in mind as well).

*Page 2. Line 36-39. Can you give examples of trend studies that have had these problems? Again, for the trend results presented here to be of scientific interest to the community, we need to know about issues of what has been done before.*
**Reply:** Yes, we can/will refer to some published work for ozone (Ball et al., ACP, 2017, acp-17-12269-2017) that points to regions/periods of trend differences that can be traced to data set issues and/or merging issues (for example, regarding merged SBUV data or an older ozone data version from GOZCARDS). While uncertainties relating to data merging are not easy to quantify, more work should ultimately be done on such a topic (separately from our current manuscript, of course); for SBUV, some work has been done regarding the propagation of uncertainties (*Frith et al.*, ACP, 2017, acp-17-14695-2017). Incidentally, data merging uncertainty issues point to a good reason to at least try to start using MLS data alone (as there are no data merging or sampling difference issues) for trend work, versus model results and in general.

*Page 3. Line 34. After reading these sections it is not clear to me if ACE data (and which version) is included in either of the GOZCARDS versions. Please clarify.*
**Reply:** Certainly, this text will be clarified, by changing it to: "ACE-FTS data were not included in these more recent years." The version matters less, since there is only one choice for recent years. We also plan to add the following sentence (just before paragraph 3), to clarify what was done for the special v2.20 GOZCARDS ozone product. "We note that no ACE-FTS data were included in this newer version of GOZCARDS $O_3$."

*Page 4. Line 39. Clarify that 'organic halogens' are the source gases.*

**Reply:** Yes, this will be changed to 'organic halogens' to specify the source.

*Page 5. Line 2. So the FR WACCM is relaxed to the observed tropical winds (QBO). What is the implication of that for the comparison? Does that constrain some of the comparisons? What would happen without this relaxation? (Why is it done?).*

**Reply:** If one wants to represent the observed stratospheric variability, one has to include QBO forcing in the tropical region; without this, the variability would be much less realistic, and less accurate. This was also the specification for the CCMI scenario (REF-C1), to include either a nudged QBO or an interactively-derived QBO (if possible). The latest version of FR-WACCM, recently released to the community, now has an interactive QBO. This was not available for this CCMI assessment.

*Page 5. Line 11. New paragraph before 'Both'.*

**Reply:** Yes, this is changed to a new paragraph.

*Page 6. Line 28. The model comparisons don't use the satellite averaging kernels (or temporal sampling I suppose?). Can you add more details on why you see no reason to apply the AKs?*

**Reply:** Some discussion of this aspect of the comparisons was already provided regarding Fig. 2 model/data differences on page 6 (lines 26-29), and this is a generic type of response for these comparisons (as has also been verified in the context of other comparisons of MLS data versus models, notably for water vapor). The MLS instrument system has sharply peaked vertical Averaging Kernels as a result of its limb viewing geometry and field of view characteristics, with stratospheric vertical resolutions of order 2.5 to 4 km in most cases (species) of relevance here. The largest impacts (from neglecting profile smoothing) can be expected in the upper troposphere, at least when comparing to fine resolution sonde profiles. Examples of smoothed and unsmoothed ozone comparisons are provided in the original MLS ozone validation paper (Fig. 6) by *Froidevaux et al.* (JGR, 2008, 10.1029/2007JD008771), in the context of comparisons versus SAGE II, which has a vertical resolution finer than 1-2 km; this shows that the effects are typically quite small (less than a few percent) even for SAGE-type profiles. The WACCM model profiles are provided on a

grid that is not substantially finer than the MLS retrieval grid, and such profiles will thus be affected even less. Also, both model runs in this case are on the same vertical grid (and the model profiles do not generally differ by very large amounts); they will be affected in the same (small) way by a small amount of smoothing to match the MLS retrieval grid. While we could add more words to this effect, we will probably not plan to lengthen the manuscript much regarding this point, given that we have at least touched on this topic already.

*Page 7. Line 13. Any idea why there are larger differences for SD WACCM? What are the implications for SD studies?*

**Reply:** Transport-related model issues, as mentioned regarding some regions of disagreement between ozone observations and data, could also impact the lower stratospheric HCl abundances. However, the HCl amounts in this region are quite small, so we do not wish to over-emphasize this sort of discrepancy for this species. Finally, it is also a region where the MLS retrievals are less well constrained, in terms of percentage accuracy at least, although this does not help to alleviate model-to-model differences. We should probably not overemphasize such large percentage differences, given the low abundances in this case.

*Page 7. Line 16. Explain 'good dynamical tracer' for non-experts.*

**Reply:** Yes, we will add some words here "$N_2O$, a long-lived species in the lower stratosphere, which means that good (or poor) model/data agreements in this region can confirm (or deny) accurate model representations of the dynamics."

*Page 12. Line 36. 'do not have the right chemistry'. I would suggest rephrasing this.*

**Reply:** Yes, we can rephrase this to 'do not include the necessary photochemical pathways, including the effects of energetic particle precipitation on ion chemistry in the upper atmosphere'.

**Revised Abstract:**

We evaluate the recently delivered Community Earth System Model version 1 (CESM1) Whole Atmosphere Community Climate Model (WACCM) using satellite-derived global composition datasets, focusing on the stratosphere. The simulations include free-running (FR-WACCM) and specified-dynamics (SD-WACCM) versions of the model. Model evaluations are made using global monthly zonal mean time series obtained by the Aura Microwave Limb Sounder (MLS), as well as longer-term global data records compiled by the Global Ozone Chemistry and Related Trace gas Data Records for the Stratosphere (GOZCARDS) project. A recent update (version 2.20) to the original GOZCARDS merged ozone ($O_3$) data set is used.

We discuss upper atmospheric climatology and zonal mean variability using $O_3$, hydrogen chloride (HCl), nitrous oxide ($N_2O$), nitric acid ($HNO_3$), and water vapor ($H_2O$) data. There are a few significant model/data mean biases, such as for lower stratospheric $O_3$, for which the models at mid- to high latitudes overestimate the mean MLS values by as much as 50% and the seasonal amplitudes by ~60%; such differences require further investigations, but would appear to implicate a transport-related issue in the models. Another clear difference occurs for $HNO_3$ during recurring winter periods of strong $HNO_3$ enhancements at high latitudes; the strong model underestimate in this case (by a factor of about 2 to 6) stems from the (known) omission of ion chemistry relating to particle precipitation effects, in the global models used here. In the lower stratosphere at high southern latitudes, the variations in polar winter/spring composition observed by MLS are generally well matched by SD-WACCM, the main exception being for the early winter rate of decrease in HCl, which is too slow in the model. In general, we find that the latitude/pressure distributions of annual and semi-annual oscillation amplitudes derived from the MLS data are properly captured by the corresponding model values.

One of the model evaluation diagnostics we use represents the closeness of fit between the model/data anomaly time series, and we also consider the correlation coefficients. Not surprisingly, SD-WACCM, which is driven by realistic dynamics, generally matches observed deseasonalized anomalies better than FR-WACCM does. We use the root mean square variability as a more valuable way to estimate differences between the two models and the observations. We find, most notably, that FR-WACCM underestimates the observed interannual variability for $H_2O$ by ~30%, typically, and by as much as a factor of two in some regions; this has some implications for estimates of the time needed to detect small trends, based on model predictions.

We have derived trends using a multivariate linear regression (MLR) model, and there is a robust signal in both MLS observations and WACCM of an upper stratospheric $O_3$ increase from 2005 to 2014 by ~0.2-0.4%/yr ($\pm$ 0.2%/yr, $2\sigma$), depending on which broad latitude bin (tropics or mid-latitudes) is considered. In the lower stratosphere, some decreases are indicated for 1998-2014 (based on merged GOZCARDS $O_3$), but we find near-zero or positive trends when using MLS $O_3$ data alone for 2005-2014. The SD-WACCM results track these observed tendencies, although there is little statistical significance in either result; however, the patterns of $O_3$ trends versus latitude and pressure are remarkably similar between SD-WACCM and MLS results. For $H_2O$, the most statistically significant trend result for 2005-2014 is an upper stratospheric increase, peaking at slightly more than 0.5%/yr in the lower mesosphere, in fairly close agreement with SD-WACCM trends, but with smaller values in FR-WACCM. As shown before by others, there are multiple factors that can influence low-frequency variability in $H_2O$; indeed, these recent short-term trends go beyond what one would expect from changes associated with a slow, secular increase in

==methane.== For HCl, while the lower stratospheric vertical gradients of MLS trends are duplicated to some extent by SD-WACCM, the model trends (decreases) are always on the low side of the data trends. There is also little model-based indication (in SD-WACCM) of a significantly positive HCl trend derived from the MLS tropical series at 68 hPa. ==These differences deserve further study.== For $N_2O$, the MLS-derived trends (for 2005-2012) point to negative trends (of up to about -1%/yr) in the NH mid-latitudes and positive trends (of up to about +3%/yr) in the SH mid-latitudes, in good agreement with the asymmetry that exists in SD-WACCM trend results. The small observed positive $N_2O$ trends of ~0.2%/yr in the 100 to 30 hPa tropical region are also consistent with model results (SD-WACCM in particular), which in turn are very close to the known rate of increase in tropospheric $N_2O$. In the case of $HNO_3$, MLS-derived lower stratospheric trend differences (for 2005-2014) between hemispheres are opposite in sign to those from $N_2O$ and in reasonable agreement with both WACCM results.

The data sets and tools discussed here for the evaluation of the models could be expanded to additional comparisons of species not included here, as well as to model intercomparisons using a variety of CCMs, ==in order to search for systematic differences versus observations or between models==, keeping in mind the range of model parameterizations and approaches.

**Table 1.** Pubs. in CCMI special issue (mostly ACP papers recently, with a variety of topics/thrusts).

| Reference | Title | Type of study (model vs data, etc…) | Some novel aspects of atm. science? | Mostly model description or model analyses? > not much data |
|---|---|---|---|---|
| Jockel, P. et al. (2016), 10.5194/**gmd**-9-1153-2016 **GMD** | Earth System Chemistry integrated Modelling (ESCiMo) with the Modular Earth Submodel System v-2.5 | *One model* with different scenarios | Not really | Yes, model sensitivity (scenario) runs |
| Tilmes, S. et al. (2016), 10.5194/**gmd**-9-1853-2016 **GMD** | Representation of the CESM1 CAM4-chem within the CCMI | *One model* (different scenarios) & some data | Not really | Model evaluation studies |
| Strode, S. A. et al. (2016), 10.5194/**acp**-16-7285-2016 | Interpreting space-based trends in CO with multiple models. | Model and data | Yes, in terms of model/data differences. | A combination of models and data |
| Morgenstern, O. et al. (2017), 10.5194/**gmd**-10-639-2017 **GMD** | Review of the global models used within phase 1 of CCMI. | Descriptions of various CCMI models | Not directly | Model descriptions only |
| Fernandez, R. P. et al. (2017), 10.5194/**acp**-17-1673-2017 | Impact of biogenic VSL bromine on the Antarctic $O_3$ hole during the 21$^{st}$ century. | *One model* and data - with model predictions | Not directly, but based on model predictions | Yes, mostly model predictions |
| Smalley, K. M. et al. (2017), 10.5194/**acp**-17-8031-2017 | Contribution of different processes to changes in tropical LS $H_2O$ in CCMs. | Models and some data | Yes, based on model behaviors & inferences | Yes, mostly model analyses |
| Hardiman, S. C. et al. (2017), 10.5194/**gmd**-10-1209-2017 **GMD** | The Met Office HadGEM3-ES CCM: evaluation of strat. dynamics, impact on $O_3$ | *One model:* different simulations (FR vs SD) | Not directly | Yes, mostly model analyses and evaluations |
| Lin, M. et al. (2017), 10.5194/**acp**-17-2943-2017 | US surface $O_3$ trends & extremes (1980- 2014): quantifying the roles of rising Asian emissions, domestic controls, wildfires, and climate. | *One model* with data comparisons | Yes, based on one model's behavior & inferences | Mostly model inferences (with some data comparisons) |
| Maycock, A. C. et al. (2018), 10.5194/**acp**-18-11323-2018 | The representation of solar cycle signals in strat $O_3$- Part-2: Analysis of global models. | Mostly multi-model results | Not directly, mostly model dependence on inputs | Yes, mostly a model sensitivity study |

| Reference | Title | Type of study (model vs data, etc…) | Some novel aspects of atm. science? | Mostly model description or model analyses? >not much data |
|---|---|---|---|---|
| Morgenstern, O. et al. (2018), 10.5194/**acp**-18-1091-2018 | $O_3$ sensitivity to varying greenhouse gases and ozone-depleting substances in CCMI-1 simulations. | Multi-model description & consistency of responses to forcings | Not directly | Yes, a model sensitivity study |
| Revell, L. E. et al. (2018), 10.5194/**acp**-17-13139-2017 | Impacts of Mt. Pinatubo volcanic aerosol on the tropical stratosphere in CCM simulations using CCMI & CMIP6 stratos. Aerosol data | *One model*. Sensitivity of T and $O_3$ response to volcanic aerosol data | Not directly | Yes, mostly a model sensitivity study |
| Hou, P. et al. (2018) **acp**-18-8173-2018 | Sensitivity of atmos. aerosol scavenging to precip. intensity and frequency in context of climate change | Some data but mostly a prediction sensitivity study | Yes, but based on prediction sensitivities | Yes, mostly a model sensitivity study (with different met. fields) |
| Phalitnonkiat, P. et al. (2018), 10.5194/**acp**-18-11927-2018 | Extremal dependence between T and $O_3$ over the continental US. | Some data but mostly multi-model prediction | Yes, but based on model predictions | Yes, mostly a model sensitivity study |
| Orbe, C. et al. (2018), 10.5194/**acp**-18-7217-2018 | Large-scale tropospheric transport in the CCMI simulations. | Multi-model diffs.: AOA, transport. | Not directly | Yes, mostly a model sensitivity study |
| Wu, X. et al. (2018), 10.5194/**acp**-18-7439-2018 | Spatial and temporal variability of interhemispheric transport times. | *One model*: Variability of idealized tracers | To some extent, based on model sensitivity | Yes, mostly a model sensitivity study (of variability) |
| Dietmuller, S. et al. (2018), 10.5194/**acp**-18-6699-2018 | Quantifying the effect of mixing on the mean age of air in CCMVal-2 and CCMI-1 models. | Multi-model look: factors influencing AOA | Not directly | Yes, mostly a model sensitivity study |
| Dhomse, S. S. et al. (2018), 10.5194/**acp**-18-8409-2018 | Estimates of ozone return dates from CCMI simulations. | Multi-model estimates: $O_3$ return dates | Yes, but based on predictions | Yes, mostly a model sensitivity study |
| Ayarzaguena, B. et al. (2018), 10.5194/**acp**-18-11277-2018 | No robust evidence of future changes in major stratospheric sudden warmings: a multi-model CCMI assessment | Multi-model study of major strat. sudden warmings | Yes, based on model predictions | Yes, mostly a model sensitivity study |

| Reference | Title | Type of study (model vs data, etc…) | Some novel aspects of atm. science? | Mostly model description or model analyses? >not much data |
|---|---|---|---|---|
| Lamy, K. et al. (2018), ACPD, 10.5194/**acp**-2018-525 | UV radiation modelling using output from the CCMI | Multi-model UVI versus climo UVI data | Yes, based on model results | A combination of models and data |
| Revell, L. E. et al. (2018) **acp**-2018-615 | Tropospheric ozone in CCMI models and Gaussian emulation to understand biases in the SOCOLv3 CCM. | Multi-model comparison of tropos. ozone vs data | Mostly geared towards model refinements | A combination of models and data |

---

## Author Comment (AC2) · 18 Oct 2018

**Reply to the review from Referee 2**

We are thankful to this referee for the review and the associated suggestions, listed in italics below. We provide our detailed responses (regular font) and plans; our revised manuscript will be available in a fairly short time. For added information, we have provided the revised Abstract in our reply here, although most of those (highlighted) changes were done as a response to comments from the other Referee.

It would seem from the referee comments that there are no demonstrable big issues with the science (or math), besides some requested clarifications, and we are pleased that Referee 2 found our manuscript to contain "a lot of valuable and detailed information" [and Referee 1 also found "some interesting results"].

*The manuscript aims to evaluate the stratospheric composition of the free-running and specified-dynamics version of CESM1 (WACCM). The evaluations are based on comparisons to satellite measurements including single-instrument and merged data records. The model diagnostics include zonal monthly mean comparisons, seasonal and semi-annual cycles as well as long-term trends. All evaluations are described in detail and valuable information on various aspects of the model performance is provided. Overall, the manuscript is of great interest for scientist directly working with WACCM or potentially with other earth-system models. Therefore, such a detailed manuscript would seem much more appropriate in a journal focused on geoscientific model development/validation and I would urge the authors to submit it to a journal focused on this topic.*
**Reply:** We do not agree that the fairly comprehensive analyses presented here are of more limited interest to modeling groups only, because there are also inferences from data sets that have not been presented before (in particular trend analyses from MLS data alone). We have also added a few clarifications to better explain certain aspects of these trends and comparisons (largely in response to the other Referee).

Regarding the Journal issue, we feel quite strongly that such a paper is (or can certainly be) in the ACP domain, given that the model description is really a small part of this manuscript (WACCM having been used and described previously, including in GMD, *Morgenstern et al.*, gmd-10-639-2017), and that there are some scientific results discussed here (to be further clarified, as mentioned in our replies and in the upcoming revision), even if some of this confirms past/recent work, but from our own model/data comparisons. There is some "grey region" between ACP and

GMD papers, with the latter being more geared towards model description and development (if one looks through many of those articles), although there are some model evaluation papers there as well. To be more specific, we include Table 1 at the end of this reply, and this provides a summary of all the papers that are part of the current CCMI special issue, which is what we are submitting to here; this special issue encompasses several journals (including ACP and GMD). As one can see from Table 1, the more recent papers have nearly all been part of ACP, after some initial work with much more of a model description focus. Some of the articles in ACP could compare broadly to the work we are trying to present, with a combination of model and data (and comparisons). We also feel that there are detailed aspects of the MLS data sets described in our work (regarding absolute error bars and trend uncertainties, including some drift issues) that would be of much interest to the stratospheric component of the ACP readership. Without attempting to be more comprehensive, we can state that we did consider the Journal topic seriously, which also led to some delays. We also consulted with the ACP editors on this topic, and we are pleased that they agree with our views; this topic is also something that editors consider as part of the pre-review process. It is also true that going through another 4 months of review with a completely new set of reviewers and editors is a considerable burden not just on the authors, with further time delays, but also on the reviewer community (especially for longer papers). We are thus thankful for the support we obtained towards finalizing this process for ACP, and we feel that we can now focus our efforts to that end; we would very much welcome reviewer support on this aspect as well.

**Major comments**

*1) The paper delivers a lot of valuable and detailed information, however, is overall very long. In particular, the number of figures could be reduced from 32 to around 20. To give one example, Figure 2 is only discussed very briefly in the text in order to illustrate mean biases and annual cycle differences shown elsewhere and could be removed.*

**Reply:** We really prefer to keep Fig. 2 in the main text as this does show much better than Fig. 1 how the models and the data differ (in certain regions) in terms of not only the mean differences, but also the annual cycle; these differences are now also better quantified in the upcoming revised version (in answer to a comment from reviewer 1).

We are planning to cut down on the length of this manuscript, mainly by relegating some of the less critical Figures to the Supplement. Although this does not necessarily translate into a very large cut in terms of text length, we consider this work to be a fairly comprehensive analysis, which therefore leads to a longer paper; there have definitely been some longer (atmospheric) papers in the literature, and specifically in ACP. Turning this into two separate papers mainly for the sake of overall length seems too artificial, and this would be quite an elaborate proposition, with the need for some duplication regarding both the data sets and the models; as an aside, this would actually lead to more reviewing work for the community. We hope to have shown that detailed analyses are necessary to enable identification of both good agreement (a result in itself) or significant differences between model runs and the data sets, but also for some of the more subtle differences, and furthermore, that an understanding and discussion of error bars and potential data issues is important. We will also strive to reduce the amount of text in the revised manuscript, especially where some less critical aspects can be discussed more succinctly, or taken out altogether. In particular, we plan to shorten Section 5.1.1 (pages 11-13) to a text length that roughly matches (rather than exceeds) the text length of Section 5.1.2 (on variability issues); the cuts to Sect. 5.1.1 will be of order 30% (or more).

In terms of reducing the amount of Figures and related changes, our specific plans are to remove Figs. 13, 14, 15, and 17 from the main text (and relegate these to the Supplement, with a slightly shortened discussion), since these mainly reinforce the expectation (already noted for $O_3$ and $H_2O$) of better model/data fits from SD-WACCM, as one might expect from a model with better dynamical constraints than the FR-WACCM version. Such an expectation does not hold for the variability diagnostics, so these are really best left in the main text, although we will plan to displace Figure 22 (on the $N_2O$ and $HNO_3$ variability comparisons), and move it to the Supplement. Moreover, we feel that Figure 31 on lat/p contours of short-term trend for various species can be moved to the Supplement, as it is less critical, and given past (and ongoing) work on this topic. While Figure 32 is interesting to us, it is more of a side note on lower stratospheric tropical cohesiveness for various species exhibiting similar dynamical variability, so we decided that the text and Figure in this case can be eliminated altogether without much of an impact on this paper.

In summary, the total number of Figures in the main text will be trimmed down by almost a quarter, with a more manageable total of 25 Figures; writing up a multi-year effort of (part-time) work on detailed model/data comparisons is bound to lead to a longer manuscript than several

shorter analyses; to our knowledge, fits, correlations, variability, and trend comparisons are rarely investigated to this extent in model/data comparisons, even for a single model (or two flavors of one model). This, with some reductions (and clarifications) in the text (including the Abstract and Conclusions section), will at least show our good faith effort towards the referee comments. Recommending a goal of exactly 20 Figs. is rather arbitrary, but our point here is that we have considered these requests with some care, and that we are being responsive.

*2) Differences are often only listed and not explored more in detail. To give one example, model HCl shows systematic differences in the lower stratosphere (evaluation based on Fig. 4) and a discussion relating those differences to shortcomings in the model transport or model chemistry would be interesting. Given the length of the manuscript, one could focus on the gases for which the detected differences are discussed in terms of model behavior (e.g., HNO$_3$). Differences for other gases can be mentioned in the manuscript with the according figures being moved to the supplement.*

**Reply:** Yes, we pursued this type of reorganization, as explained above, with what we would consider a reasonable amount of delegating of material to the Supplement. We find some value in the remaining Figures, and feel that using a somewhat arbitrary number (such as 20) is not justified for a paper that covers a fair amount of ground and wishes to confront the models with a multi-species approach, in order to check for potential areas of weakness. Just stating good agreement and putting almost every Figure in the Supplement could work also, in principle, but that would be the other extreme, with a nearly complete lack of visual confirmation, which we think is important to preserve as well. Also, while we are striving to cut down on the length here, there are other long papers in the literature (but we will most likely avoid this sort of length in the future).

*3) In section 3, existing evaluations of WACCM and the WACCM composition in particular should be discussed. Such references come up in the latter part of the manuscript. If they are given combined in this section, it will easier for the reader to identify what the current challenges are and what is new in this manuscript.*

**Reply:** Yes, this section and/or the Introduction will be modified in the revised version (without adding too much length) to take this into account; in particular, we will add some motivation for the comparisons done here for FR-WACCM versus SD-WACCM (and observations).

*Minor comments:*

*1) Consider changing the title to 'Evaluation of CESM1 (WACCM) free-running and specified-dynamics stratospheric composition simulations using global multi-species satellite data records.*
**Reply:** Given that water vapor is considered all the way through the mesosphere, we prefer to stick to our original title, but we did consider this suggestion.

*2) Page 5, line 31 – Page 6, line 2: This text could be moved to the discussion of the MLS data record in section 2.1.*
**Reply:** While this could be done in principle, we feel that the species-specific discussions of error bars and validation work is really best kept as part of the discussions for each species, and that the flow is less awkward this way; we have thus not tried to reorganize these portions of text.

*3) Page 7, line 24: Do you mean all earth system model or just WACCM with the term 'general model underestimation'?*
**Reply:** We mean just the WACCM models here. This is clarified in the revised version by stating "model underestimation by both WACCM versions." However, it is implicit that other models without the proper (more complicated) chemical processes and energetic particle pathways will also underestimate $HNO_3$ in the same fashion.

*4) Page 9, line 7 -10: Here, and also in other places, the sentence is too long for easy understandability. Consider splitting into two sentences at the semicolon.*
**Reply:** Yes, we will start a new sentence instead of using a semi-colon, if/as that may help. We will also consider some other places for such an issue.

*5) Page 12, line 5-8: The statement is made for the upper mesosphere. But isn't it also true for the stratosphere?*
**Reply:** The statement (regarding worse diagnostic values) is somewhat true for the upper stratosphere as well, but we are mainly referring to SD-WACCM here; nevertheless, we have

modified the revised text to state that the (SD-WACCM) diagnostics "are of poorest quality in the mesosphere" (etc…).

*6) Page 13, line 7: MIPAS has been used earlier in the manuscript.*

**Reply:** Yes, thank you; this is readily fixed by defining MIPAS earlier on in the text (in the 2nd part of section 4).

**Revised Abstract:**

We evaluate the recently delivered Community Earth System Model version 1 (CESM1) Whole Atmosphere Community Climate Model (WACCM) using satellite-derived global composition datasets, focusing on the stratosphere. The simulations include free-running (FR-WACCM) and specified-dynamics (SD-WACCM) versions of the model. Model evaluations are made using global monthly zonal mean time series obtained by the Aura Microwave Limb Sounder (MLS), as well as longer-term global data records compiled by the Global Ozone Chemistry and Related Trace gas Data Records for the Stratosphere (GOZCARDS) project. A recent update (version 2.20) to the original GOZCARDS merged ozone ($O_3$) data set is used.

We discuss upper atmospheric climatology and zonal mean variability using $O_3$, hydrogen chloride (HCl), nitrous oxide ($N_2O$), nitric acid ($HNO_3$), and water vapor ($H_2O$) data. There are a few significant model/data mean biases, such as for lower stratospheric $O_3$, for which the models at mid- to high latitudes overestimate the mean MLS values by as much as 50% and the seasonal amplitudes by ~60%; such differences require further investigations, but would appear to implicate a transport-related issue in the models. Another clear difference occurs for $HNO_3$ during recurring winter periods of strong $HNO_3$ enhancements at high latitudes; the strong model underestimate in this case (by a factor of about 2 to 6) stems from the (known) omission of ion chemistry relating to particle precipitation effects, in the global models used here. In the lower stratosphere at high southern latitudes, the variations in polar winter/spring composition observed by MLS are generally well matched by SD-WACCM, the main exception being for the early winter rate of decrease in HCl, which is too slow in the model. In general, we find that the latitude/pressure distributions of annual and semi-annual oscillation amplitudes derived from the MLS data are properly captured by the corresponding model values.

One of the model evaluation diagnostics we use represents the closeness of fit between the model/data anomaly time series, and we also consider the correlation coefficients. Not surprisingly, SD-WACCM, which is driven by realistic dynamics, generally matches observed deseasonalized anomalies better than FR-WACCM does. We use the root mean square variability as a more valuable way to estimate differences between the two models and the observations. We find, most notably, that FR-WACCM underestimates the observed interannual variability for $H_2O$ by ~30%, typically, and by as much as a factor of two in some regions; this has some implications for estimates of the time needed to detect small trends, based on model predictions.

We have derived trends using a multivariate linear regression (MLR) model, and there is a robust signal in both MLS observations and WACCM of an upper stratospheric $O_3$ increase from 2005 to 2014 by ~0.2-0.4%/yr (± 0.2%/yr, 2σ), depending on which broad latitude bin (tropics or mid-latitudes) is considered. In the lower stratosphere, some decreases are indicated for 1998-2014 (based on merged GOZCARDS $O_3$), but we find near-zero or positive trends when using MLS $O_3$ data alone for 2005-2014. The SD-WACCM results track these observed tendencies, although there is little statistical significance in either result; however, the patterns of $O_3$ trends versus latitude and pressure are remarkably similar between SD-WACCM and MLS results. For $H_2O$, the most statistically significant trend result for 2005-2014 is an upper stratospheric increase, peaking at slightly more than 0.5%/yr in the lower mesosphere, in fairly close agreement with SD-WACCM trends, but with smaller values in FR-WACCM. As shown before by others, there are multiple factors that can influence low-frequency variability in $H_2O$; indeed, these recent short-term trends go beyond what one would expect from changes associated with a slow, secular increase in

==methane.== For HCl, while the lower stratospheric vertical gradients of MLS trends are duplicated to some extent by SD-WACCM, the model trends (decreases) are always on the low side of the data trends. There is also little model-based indication (in SD-WACCM) of a significantly positive HCl trend derived from the MLS tropical series at 68 hPa. ==These differences deserve further study.== For $N_2O$, the MLS-derived trends (for 2005-2012) point to negative trends (of up to about -1%/yr) in the NH mid-latitudes and positive trends (of up to about +3%/yr) in the SH mid-latitudes, in good agreement with the asymmetry that exists in SD-WACCM trend results. The small observed positive $N_2O$ trends of ~0.2%/yr in the 100 to 30 hPa tropical region are also consistent with model results (SD-WACCM in particular), which in turn are very close to the known rate of increase in tropospheric $N_2O$. In the case of $HNO_3$, MLS-derived lower stratospheric trend differences (for 2005-2014) between hemispheres are opposite in sign to those from $N_2O$ and in reasonable agreement with both WACCM results.

The data sets and tools discussed here for the evaluation of the models could be expanded to additional comparisons of species not included here, as well as to model intercomparisons using a variety of CCMs, ==in order to search for systematic differences versus observations or between models==, keeping in mind the range of model parameterizations and approaches.

**Table 1.** Pubs. in CCMI special issue (mostly ACP papers recently, with a variety of topics/thrusts).

| Reference | Title | Type of study (model vs data, etc…) | Some novel aspects of atm. science? | Mostly model description or model analyses? > not much data |
|---|---|---|---|---|
| Jockel, P. et al. (2016), 10.5194/ **gmd**-9-1153-2016 **GMD** | Earth System Chemistry integrated Modelling (ESCiMo) with the Modular Earth Submodel System v-2.5 | *One model* with different scenarios | Not really | Yes, model sensitivity (scenario) runs |
| Tilmes, S. et al. (2016), 10.5194/ **gmd**-9-1853-2016 **GMD** | Representation of the CESM1 CAM4-chem within the CCMI | *One model* (different scenarios) & some data | Not really | Model evaluation studies |
| Strode, S. A. et al. (2016), 10.5194/ **acp**-16-7285-2016 | Interpreting space-based trends in CO with multiple models. | Model and data | Yes, in terms of model/data differences. | A combination of models and data |
| Morgenstern, O. et al. (2017), 10.5194/ **gmd**-10-639-2017 **GMD** | Review of the global models used within phase 1 of CCMI. | Descriptions of various CCMI models | Not directly | Model descriptions only |
| Fernandez, R. P. et al. (2017), 10.5194/**acp**-17-1673-2017 | Impact of biogenic VSL bromine on the Antarctic $O_3$ hole during the 21$^{st}$ century. | *One model* and data - with model predictions | Not directly, but based on model predictions | Yes, mostly model predictions |
| Smalley, K. M. et al. (2017), 10.5194/ **acp**-17-8031-2017 | Contribution of different processes to changes in tropical LS $H_2O$ in CCMs. | Models and some data | Yes, based on model behaviors & inferences | Yes, mostly model analyses |
| Hardiman, S. C. et al. (2017), 10.5194/ **gmd**-10-1209-2017 **GMD** | The Met Office HadGEM3-ES CCM: evaluation of strat. dynamics, impact on $O_3$ | *One model:* different simulations (FR vs SD) | Not directly | Yes, mostly model analyses and evaluations |
| Lin, M. et al. (2017), 10.5194/ **acp**-17-2943-2017 | US surface $O_3$ trends & extremes (1980- 2014): quantifying the roles of rising Asian emissions, domestic controls, wildfires, and climate. | *One model* with data comparisons | Yes, based on one model's behavior & inferences | Mostly model inferences (with some data comparisons) |
| Maycock, A. C. et al. (2018), 10.5194/ **acp**-18-11323-2018 | The representation of solar cycle signals in strat $O_3$- Part-2: Analysis of global models. | Mostly multi-model results | Not directly, mostly model dependence on inputs | Yes, mostly a model sensitivity study |

| Reference | Title | Type of study (model vs data, etc…) | Some novel aspects of atm. science? | Mostly model description or model analyses? >not much data |
|---|---|---|---|---|
| Morgenstern, O. et al. (2018), 10.5194/**acp**-18-1091-2018 | $O_3$ sensitivity to varying greenhouse gases and ozone-depleting substances in CCMI-1 simulations. | Multi-model description & consistency of responses to forcings | Not directly | Yes, a model sensitivity study |
| Revell, L. E. et al. (2018), 10.5194/**acp**-17-13139-2017 | Impacts of Mt. Pinatubo volcanic aerosol on the tropical stratosphere in CCM simulations using CCMI & CMIP6 stratos. Aerosol data | *One model*. Sensitivity of T and $O_3$ response to volcanic aerosol data | Not directly | Yes, mostly a model sensitivity study |
| Hou, P. et al. (2018) **acp**-18-8173-2018 | Sensitivity of atmos. aerosol scavenging to precip. intensity and frequency in context of climate change | Some data but mostly a prediction sensitivity study | Yes, but based on prediction sensitivities | Yes, mostly a model sensitivity study (with different met. fields) |
| Phalitnonkiat, P. et al. (2018), 10.5194/**acp**-18-11927-2018 | Extremal dependence between T and $O_3$ over the continental US. | Some data but mostly multi-model prediction | Yes, but based on model predictions | Yes, mostly a model sensitivity study |
| Orbe, C. et al. (2018), 10.5194/**acp**-18-7217-2018 | Large-scale tropospheric transport in the CCMI simulations. | Multi-model diffs.: AOA, transport. | Not directly | Yes, mostly a model sensitivity study |
| Wu, X. et al. (2018), 10.5194/**acp**-18-7439-2018 | Spatial and temporal variability of interhemispheric transport times. | *One model:* Variability of idealized tracers | To some extent, based on model sensitivity | Yes, mostly a model sensitivity study (of variability) |
| Dietmuller, S. et al. (2018), 10.5194/**acp**-18-6699-2018 | Quantifying the effect of mixing on the mean age of air in CCMVal-2 and CCMI-1 models. | Multi-model look: factors influencing AOA | Not directly | Yes, mostly a model sensitivity study |
| Dhomse, S. S. et al. (2018), 10.5194/**acp**-18-8409-2018 | Estimates of ozone return dates from CCMI simulations. | Multi-model estimates: $O_3$ return dates | Yes, but based on predictions | Yes, mostly a model sensitivity study |
| Ayarzaguena, B. et al. (2018), 10.5194/**acp**-18-11277-2018 | No robust evidence of future changes in major stratospheric sudden warmings: a multi-model CCMI assessment | Multi-model study of major strat. sudden warmings | Yes, based on model predictions | Yes, mostly a model sensitivity study |

| Reference | Title | Type of study (model vs data, etc...) | Some novel aspects of atm. science? | Mostly model description or model analyses? >not much data |
|---|---|---|---|---|
| Lamy, K. et al. (2018), ACPD, 10.5194/**acp**-2018-525 | UV radiation modelling using output from the CCMI | Multi-model UVI versus climo UVI data | Yes, based on model results | A combination of models and data |
| Revell, L. E. et al. (2018) **acp**-2018-615 | Tropospheric ozone in CCMI models and Gaussian emulation to understand biases in the SOCOLv3 CCM. | Multi-model comparison of tropos. ozone vs data | Mostly geared towards model refinements | A combination of models and data |

---

## Referee Report (RR1)

This is a very comprehensive paper, encompassing model-model, model-data and data-data comparisons, and will be a fantastic resource for the WACCM community. The authors have followed many of the previous reviewers' suggestions, but I feel that some of these points still need to be addressed, particularly regarding clarity and length. I appreciate the authors have put a lot of work into this analysis, and do not suggest discarding large sections entirely, but I think some tightening up of the text and figures will improve uptake (see my specific suggestions, below).

As a general comment, I agree with a comment by one of the previous reviewers that discussion of differences between FR-WACCM and SD-WACCM is fairly limited. The authors argue (perhaps rightly so) that this would justify a separate paper in its own right, but the discussion of data-data differences is often fairly comprehensive, so at times the paper feels a little unbalanced.

The authors have done a good job at setting the paper within the existing body of literature. Overall, I think this paper is close to being ready for publication, and my specific comments below aim to improve the length, clarity and uptake of the paper.

1. Abstract. Page 1, lines 12-19 are clear, as are lines 10-12 on page 2. In between is a lot of detail that I think could be simplified. What are the simple, take home messages of this study? What is new and exciting?
2. Figures. To reduce the number of figures, and reduce the amount of cross-referencing between different subplots, I suggest as an example for Figure 1:
a) Changing the colormap to red-white-blue, where red shows positive values and blue shows negative values, or something similar. The current colormap is not intuitive to read, and readers with color blindness will not be able to distinguish the red/green shading.
b) Add contour lines showing the climatological mean state for MLS, MLS and FR-WACCM (for the three plots on the left-hand side of figure 1). Then you can get rid of the left-hand column of figure S1.
c) Apply hatching over the areas where the model is outside the error limits on the observations (or where it's inside the error limits, if you prefer). Then you can get rid of the right-hand column of figure 1.
d) Add labels (a) (b) etc to the subpanels – referring to Fig. 1c is more concise than Fig. 1 (left column, bottom panel) etc.

In total, this reduces nine sub-plots down to three. For a nice example of how this was done in a previous WACCM study, see e.g. Figure 3 of
https://agupubs.onlinelibrary.wiley.com/doi/full/10.1002/2014GL061627

3. Appendices. There is quite a bit of overlap between the appendices and main text. I wondered if some of the discussion from the main text should be moved to the appropriate appendix, to help make the main text more concise, and the appendices stand alone. E.g. Page 8, lines 3-23 could go to Appendix A1; Page 10, lines 4-28 could go to Appendix A2, along with page 11 lines 3-28; page 14 lines 29-39 could go to Appendix A3. This is just a suggestion, however.
4. Discussion of figure S15, page 16 lines 5-20 – could this form a separate stand-alone appendix?
5. Section 6 is largely repetitive of prior sections. I suggest replacing this simply with a short 'Conclusions' section, and ensuring that the discussion here is incorporated earlier into the

text, as sections 4 and 5 already contain rather a lot of discussion. In one of the responses to the previous reviewers, the authors presented a bullet-point list showing what was new for the evaluation of each species. Perhaps that type of approach could be included here as well?

6. More accurate cross-referencing would help, e.g. p13 line 11 refers to the 'trends section' – please also state the section number.

---

## Author Response (AR2)

**Reply regarding comments on the Froidevaux et al. manuscript (ACPD).**

Most of the changes in this revised version can be viewed in the tracked-changes version that we have re-submitted, although very small editorial changes are often not shown (i.e. we accepted these changes already), to keep the more important changes most visible. The total length (main text + Appendices) has not changed much at all, given that we tried hard to respond to the goals of more clearly separating the more novel/atmospheric change results; this involved processing the datasets for trends to 2018, so more text had to be included for this discussion (as well as in the Abstract (see yellow highlights and in the Conclusions). To compensate, we tightened up some of the model evaluation details by making the wording more concise, and delegating some paragraphs to the Appendix (see more details below). Finally, we have added references to two papers: a reference to Chipperfield et al. (2018), which is also relevant regarding more recent ozone trends (but only through 2017), and one to Douglass et. al. (2018) regarding patterns of variations for various species (including species discussed in this paper), although these authors focused more on model/data column results.

We do believe that this is now in a state that is significantly improved, and we thank all the referees and the editor for their thoughtful comments.

1. *Abstract suggestions.*
  *Mainly, we were asked to extract the most important points, especially regarding what is more novel/exciting in this work*
- We have done this, and made the first main points regarding mainly observational results, with an emphasis on trends that include the latest data from MLS (through 2018), and for the longer-term (GOZCARDS time series as well). This required a number of changes in several of the trend-related Figures (the addition of grey points for the more recent trend results). See Figures 19 through 25; also some of these trend results changed, even for time series through 2014, although mainly for H2O in the upper mesosphere, as a result of the longer-term proxy used in the regression fits, to account for solar cycle-related impacts, which is also an improvement to this work.

2. *We have indeed simplified Figures like Fig. 1*, and reduced these all to three panels; also for Figs. 3, 4, 5, and 6. Fairly small text changes had to be made to account for these Figure changes (see grey highlighted region, pages 5,6,7).

(a) *no color scheme changes* were made at this stage in this long manuscript, as we have drawn from many programs to produce all the various plots. Also, inevitable trials and errors would occur for us to finalize such a change throughout the paper, which is a lot to ask, at this late stage in quite a long review process already, in our opinion, even if it sounds simple (for a short paper with 4 Figures, say). Also, we have had long discussions in the past with other colleagues about such color schemes, and this is somewhat subjective (and had to be placed in the context of our past decisions). However, we would seriously consider making such a change for other papers in the future, as we see that this seems to be a fairly recent trend/push, although not required (yet) by journals (as far as we know).

(b) *We have kept Fig. S1 and others like that* - just in the Supplement so this does not take up much real room for this manuscript. Also, it is really not that easy to see overplotted contours on top of contours, and especially if there are also hatched regions to indicate significance.

(c) *We did apply some hatching (crosses)* for regions where the differences are not significant enough. See the text and new Figs. (1, 3, 4, 5, 6).

(d) *We did not add actual labels*, as this would probably need to be done more homogeneously for the many Figures in the manuscript, and this would really not save much space at all.

3. *Appendices (and related reshuffling of text to the Appendices). A few suggestions for possible change were provided by the referee.*

> *pg. 8, lines 3-23 could go to Appendix A1.*
- Yes, we have decided to move this somewhat tangential discussion of other grading/evaluation methods largely to Appendix A1 (where this is discussed). We had to leave some context and lead-in for this Appendix, so a few lines were left in the main text for this purpose. The discussion in the Appendix (including the transfer of text) was shortened, also because parts of Figs. 1, 3, 4, 5, 6 have been deleted (so it is not possible refer to these discarded plots for diagnostic comparisons).

> *pg. 10, lines 4-28, and pg. 11, lines 3-28 could go to Appendix A2.*
- We disagreed with this suggestion, in part, because it would involve too many changes and would disrupt the flow that exists right now (in terms of Fig. 10 - and therefore, regarding other similar Figs. (12,14,15) later on). More importantly, we feel that, while these Figs. are somewhat less eye-opening than others, given the expectations for SD-WACCM versus FR-WACCM, this confirmation of expectations still needs to be formally provided regarding such model goodness of fits tests; relegating all this work to an Appendix seems a bit extreme to us, and one could thereby go too far in terms of the balance between main text and Appendices (and there are 3 of these already).

> *pg. 14, lines 29-39 could be in Appendix A3.*
- We have shortened the main text somewhat (now about 8 lines versus 11 originally), but we did want to give the context and main flavor of the regression model, and given that regression terms have now been added, some mention of this had to be made in the main text also, with most of the details in Appendix A3.

4. *pg. 16, lines 5-20, discussion of Fig. S15 could maybe be a separate Appendix.*
- We have decided to shorten this discussion instead; also, an Appendix does not really shorten the main manuscript. We now have this discussion length at 9 lines, rather than the original 16 lines.

5. *Section 6 is largely repetitive of prior sections. Can this be shortened?*
- We had to make sure we answered the calls (also from earlier reviewers) to make the aspects that are more novel/relevant to atmospheric change more clear and up front. To this end, we analyzed and included trends for results up through 2018, even if the model comparison/evaluation portion only extends to the end of the model runs (2014), as shown originally. This necessitated additional discussion, and more related text in the Conclusions, as well as in the Abstract. But it should address one of the main issues - having to do with newer results, and their separation from the model evaluation parts. Granted, there is a lot here that deals with model evaluation, and this can indeed fall in a grey region, in terms of which Journal to submit the work for. Nevertheless, we have still managed to reduce the total length of Section 6 from over 3 pages to about 2.5 pages. Given that many people often skim through a paper and read mostly the Abstract and/or Conclusions, at least initially, we feel that it is important to still have Section 6 provide a good (yet concise enough) summary of the large amount of work provided in this paper.
 - The best thing to review these changes is probably to compare the old and new versions, although we have highlighted (in bold red) the beginning text portions where either more novel atmospheric results are discussed, or where a model evaluation discussion is provided.

6. Yes, *we have changed the reference made for the "trends section"* to the appropriate Section number; we have also checked that this is not really an issue elsewhere in this manuscript.

[revised manuscript text omitted]